# Understanding the Stability-based Generalization of Personalized Federated Learning

**Yingqi Liu**[1,2]   **Qinglun Li**[3]   **Jie Tan**[4]   **Yifan Shi**   **Li Shen**[1,2*]   **Xiaochun Cao**[1]

[1]School of Cyber Science and Technology, Shenzhen Campus of Sun Yat-sen University, China
[2]Guangdong Laboratory of Artificial Intelligence and Digital Economy (SZ), China
[3]College of Systems Engineering, National University of Defense Technology, China
[4]Intelligent Game and Decision Lab, China

`liuyingqi1199@gmail.com; liqinglun@nudt.edu.cn; j.tanjie@outlook.com`
`mathshenli@gmail.com; caoxiaochun@mail.sysu.edu.cn`

## Abstract

Despite great achievements in algorithm design for Personalized Federated Learning (PFL), research on the theoretical analysis of generalization is still in its early stages. Some theoretical results have investigated the generalization performance of personalized models under the problem setting and hypothesis in convex conditions, which can not reflect the real iteration performance during non-convex training. To further understand the real performance from a generalization perspective, we propose the first algorithm-dependent generalization analysis with uniform stability for the typical PFL method, Partial Model Personalization, on smooth and non-convex objectives. Specifically, we decompose the generalization errors into aggregation errors and fine-tuning errors, then creatively establish a generalization analysis framework corresponding to the gradient estimation process of the personalized training. This framework builds up the bridge among PFL, FL and Pure Local Training for personalized aims in heterogeneous scenarios, which clearly demonstrates the effectiveness of PFL from the generalization perspective. Moreover, we demonstrate the impact of trivial factors like learning steps, stepsizes and communication topologies and obtain the excess risk analysis with optimization errors for PFL. Promising experiments on CIFAR datasets also corroborate our theoretical insights. Our code can be seen in https://github.com/YingqiLiu1999/Understanding-the-Stability-based-Generalization-of-Personalized-Federated-Learning.

## 1 Introduction

With the rapid development of data and model scales in Machine Learning, PFL effectively improves local performance via flexible client cooperation with heterogeneous data. Although the studies of algorithm design has made considerable progress in PFL, their theoretical analysis is still scarce. Due to the data and training limitations in the real world, theoretical analysis of FL/PFL usually includes two aspects: optimization properties and generalization properties. In this work, we mainly focus on the generalization properties of PFL, which come from the overfitting gap between the training and testing datasets.

Currently, the existing generalization analysis for PFL is mainly obtained in three ways: 1) high-probability generalization bounds with concentration inequalities based on the PAC hypothesis complexity like VC dimension complexity (Deng et al., 2020; Marfoq et al., 2022; Xie et al., 2024), Rademacher complexity (Mansour et al., 2020); 2) information-theoretical distances between the output hypothesis and the prior from PAC-Bayes generalization (Achituve et al., 2021; Zhang et al., 2022); 3) the privacy-preserving ability of the change in output hypothesis when the algorithm is exposed to attacks (Dai et al., 2022b). Most upper bounds above only depend on the problem setting and hypothesis in the convex condition, which can not apply to the commonly used non-convex functions such as neural networks and can not reflect the real iteration performance during personalized training. In other words, they are weak at evaluating the effectiveness of algorithm

---

*Corresponding Author

Table 1: Main results on the upper generalization bounds of PFL. $m$ is total nodes number, $T$ is training rounds, $\eta$ is local learning stepsize, $K$ is local learning step, $n$ is the selected nodes number and $\lambda$ is about communication topology, $\sigma$ is data hetegeneity and NC representation non-convex condition.

| Tpye | | Reference | Algorithm | Analysis Tools | | $m$ | $T$ | $\eta$ | $K$ | $n/\lambda$ | $\sigma$ | | NC |
|------|---|-----------|-----------|----------------|---|-----|-----|--------|-----|-------------|----------|---|----|
| SGD, FL | | Hardt et al. (2016) | SGD | Uniform Stability | | ✓ | ✓ | ✓ | ✗ | ✗ | ✗ | | ✓ |
| | | Chen et al. (2021) | FedAvg | Uniform Stability | | ✓ | ✓ | ✓ | ✓ | ✓ | ✓ | | ✗ |
| | | Sun et al. (2024b) | FedAvg | On-average Stability | | ✓ | ✓ | ✗ | ✓ | ✓ | ✓ | | ✓ |
| | | Sun et al. (2021) | D-SGD | Uniform Stability | | ✓ | ✓ | ✓ | ✗ | ✓ | ✗ | | ✓ |
| | | Zhu et al. (2022) | D-SGD | On-average Stability | | ✓ | ✓ | ✓ | ✗ | ✓ | ✗ | | ✓ |
| PFL | | Deng et al. (2020) | C-PFL | VC Dimension Complexity | | ✓ | ✗ | ✗ | ✗ | ✗ | ✗ | | ✗ |
| | | Mansour et al. (2020) | C-PFL | Rademacher Complexity | | ✓ | ✗ | ✗ | ✗ | ✗ | ✗ | | ✗ |
| | | Zhang et al. (2022) | C-PFL | PAC-Bayes Complexity | | ✓ | ✗ | ✗ | ✗ | ✗ | ✗ | | ✗ |
| | | Ours | C-PFL | Uniform Stability | | ✓ | ✓ | ✓ | ✓ | ✓ | ✓ | | ✓ |
| | | Ours | D-PFL | Uniform Stability | | ✓ | ✓ | ✓ | ✓ | ✓ | ✓ | | ✓ |

design and hyperparameter selection while building the relationship between global collaboration and local fine-tuning. Despite the optimization design above, the upper generalization bound of PFL has a great significance with both communication topology and data heterogeneity. This insight offers a deeper understanding of the fundamental nature of PFL. To this end, we aim to establish a framework for algorithm-dependent generalization analysis in PFL, which is designed to align closely with the training process and provide a clear and community-accessible foundation for further research.

Overall, we present the first algorithm-dependent generalization for the typical PFL method Partial Model Personalization with uniform stability in non-convex conditions and evaluate the excess risk for both Centralized PFL (C-PFL) and Decentralized PFL (D-PFL). Intuitively, each shared and personalized update may introduce a specific impact on the generalization errors. Therefore, we decompose the generalization errors into aggregation errors and fine-tuning errors, then establish a generalization analysis framework corresponding to the gradient estimation process of the personalized training. From our analysis, some results can be concluded that larger learning steps, larger learning rates, and denser network connections will hurt the generalization performance for both C-PFL and D-PFL, meaning that better testing performance is the trade-off between communication cost and computational efficiency. Besides, with different aggregation modes in the shared variables, we demonstrate that C-PFL generalizes better than D-PFL, which aligns with the conclusion of the generalized FL (Sun et al., 2023c). Moreover, with the analysis of data heterogeneity, we can clearly see how PFL outperforms FL and Pure Local Training in the perspective of generalization. Combined with the convergence analysis, we obtain the excess risk and find that personalized performance is the trade-off between optimization and generalization.

We list our analysis with the existing bounds of SGD/D-SGD and PFL methods in Table 1. From comparisons, our results innovatively achieve the biased gradient estimation from multi-local updates and analyze the generalization interaction between personalized and shared variables during aggregation and local training. In summary, our main contributions are as follows:

- **New framework of generalization analysis for PFL under non-convex conditions.** We build up the first algorithm-dependent generalization analysis framework for PFL with the biased gradient from multi-local updates. It is consistent with the personalized training progress and bridges PFL, FL and Pure Local Training with the clever heterogeneity analysis, which reveals the effectiveness of PFL for personalized aims. We also extend it to the decentralized scenarios with different communication topologies.

- **New results for upper generalization bounds and excess risks for PFL.** Our algorithm-dependent results achieve comparable bounds and reflect the iteration nature, effectiveness of algorithm design as well as the hyperparameters selection of PFL. Then combined with the optimization errors, we obtain the excess risk analysis and find that the better performance is the trade-off between optimization and generalization.

- **Massive experiments verify theoretical findings.** Our experiments on CIFAR datasets with different models under non-convex conditions strongly support our theoretical insights.

## 2 RELATED WORK

**Generalization for PFL.** PFL is proposed to find the greatest personalized models for each client (related work in Appendix A). Generalization analysis represents the performance in the unseen data of a well-train model, which is defined as the difference between the population risk and empirical risk. Various statistical methods have been introduced into PFL, including methods based on PAC-based analysis, Differential Privacy analysis, and PAC-Bayes analysis. For PAC-based analysis, Deng et al. (2020) derives the VC dimension complexity bound of a mixture of local and global models, and finds the optimal mixing parameter. Mansour et al. (2020) derives the Rademacher complexity bound of the clusters, data interpolation, and model interpolation. Chen et al. (2021) analyzed the stability and excess risk of both FL and local SGD under different data heterogeneity, but failed to extend them to the non-convex condition. For Differential Privacy analysis, Dai et al. (2022a) assumes that the algorithm satisfies $(\varepsilon, \delta)$-differentially private condition and proposes the lower generalization bound with the noisy perturbation. For PAC-Bayes analysis, Zhang et al. (2022) gives an upper bound of averaged generalization error on the Bayesian variational inference method and illustrates that the convergence rate of the generalization error is minimax optimal up to a logarithmic factor.

**Stability for generalization.** The stability-based methods measure the sensitivity of the data perturbation of an algorithm via uniform stability (Bousquet & Elisseeff, 2002; Hardt et al., 2016), Bayes stability (Li et al., 2019), model stability (Lei & Ying, 2020; Liu et al., 2017), on-average stability (Lei et al., 2023; Sun et al., 2024b; Kuzborskij & Lampert, 2018), and so on. More information can be seen from the introduction in Lei et al. (2023). For the generalization bounds in FL, Lei et al. (2023) develop the stability analysis for minibatch SGD and local SGD for convex, strongly convex and nonconvex problems. Sun et al. (2024b) show that the generalization performances of FedAvg, FedProx and Scaffold are closely related to the data heterogeneity and the convergence behaviors when training. Sun et al. (2023c) discuss the better generalization performance between the Central FL and Decentralized FL. In decentralized training, Zhu et al. (2022) extend the stability-based generalization to D-SGD and discuss the topology effect of it. Zhu et al. (2024) refine the stability analysis for the minimax problem in a decentralized manner.

Nowadays, almost all upper generalization bounds of PFL based on the complexity theory ignore the impact of algorithm design and the iteration nature. Therefore, we try to propose the stability-based generalization analysis to answer how algorithm design and hyperparameter selection impact the generalization capacity. Meanwhile, we extend the non-trivial analysis to D-PFL with various communication network topologies. Extensive experiments also corroborate our theoretical findings.

## 3 PROBLEM FORMULATION

In this section, we first propose the problem setup for C-PFL and D-PFL. Then we present the uniform stability for generalization error and combine it with convergence error to obtain the excess risk.

### 3.1 PROBLEM SETUP

**Personalized Federated Learning.** Compared to typical FL methods, PFL focuses on the average minimization with the personalized models rather than the consensus one. Partial Model Personalization is one of the most significant strategies in PFL, which decouples the model as the personalized variables to satisfy the individual requirements and the shared variables to leverage the collective knowledge. We consider the typical setting with $m$ clients, where each client $i$ owns the local training data $\xi_i$ and it satisfies the data distribution $\mathcal{D}_i$. For each client, the model parameter $w_i \in \mathbb{R}^d$ are partitioned into two parts: the *shared* variables $u \in \mathbb{R}^{d_u}$ and the *personalized* variables $v_i \in \mathbb{R}^{d_i}$ for $i = 1, \dots, m$. To simplify the presentation, we denote $V = (v_1, \dots, v_m) \in \mathbb{R}^{d_1 + \dots + d_m}$. So the full model on client $i$ is denoted as $w_i = (u, v_i)$. $f_i$ (such as cross-entropy function) denote the loss function for each client, $\xi_i$ is the specific data for client $i$. The Population Risk Minimization $F$ of PFL is defined as follows:

$$\min_{u,V} \quad F(u, V) := \frac{1}{m} \sum_{i=1}^{m} F_i(u, v_i), \quad where \quad F_i(u, v_i) = \mathbb{E}_{\xi_i \sim \mathcal{D}_i} f_i(u, v_i; \xi_i). \quad (1)$$

From engineering purposes, we set the feature extraction layers (close to the input) as the shared variables $u$ and the linear classification layers (close to the output) as the personalized variables $v_i$ as

Arivazhagan et al. (2019); Collins et al. (2021); Pillutla et al. (2022); Liu et al. (2024). Meanwhile, we alternately update the shared variables and personalized variables to distinguish the generalization effects between them explicitly. Algorithm 1 illustrates the specific process. We set $\nabla_u$ as stochastic gradients of the shared variables $u$ and $\nabla_v$ as stochastic gradients of the personalized variables $v_i$, respectively. Personalized variables $v_i$ first perform the local updating with the shared variables $u$ fixed in Line 2, then the shared variables $u$ update with the personalized variables $v_i$ fixed in Line 5.

**C-PFL and D-PFL.** We consider both C-PFL and D-PFL in Algorithm 2. For C-PFL, the only central server first distributes the shared variables $u^t$ to the $n$ selected clients in Line 3, then aggregates the updated shared variables $u_i^{t+1}$ to $u^{t+1}$ from the selected clients in Line 7. Different from the general FL, partial model personalization only aggregates the shared variables $u_i$ in the central server, while keeping the personal variables $v_i$ on the client side. We focus on the case of the averaged aggregation, which means $\alpha_i = 1/n$. For D-PFL, it allows clients to communicate with their neighbors in a peer-to-peer manner without the central server. The communication can be modeled as an undirected connected graph $\mathcal{G} = (\mathcal{N}, \mathcal{V}, \boldsymbol{W})$, where $\mathcal{N} = \{1, 2, \ldots, m\}$ is the set of all clients, $\mathcal{V} \subseteq \mathcal{N} \times \mathcal{N}$ is the set of communication channels, and the gossip/mixing matrix $\boldsymbol{W}$ present as below records whether the communication connects or not between any two clients. Set $\mathcal{G}_i$ as the neighbors set for each client in the undirected graph.

**Definition 1** (**The gossip/mixing matrix** (Nedic & Ozdaglar, 2009))**.** *The gossip matrix $\boldsymbol{W} = [w_{i,j}] \in [0,1]^{m \times m}$ is assumed to have these properties: (i) (Graph) If $i \neq j$ and $(i,j) \notin \mathcal{V}$, $w_{i,j} = 0$, otherwise, $w_{i,j} > 0$; (ii) (Symmetry) $\boldsymbol{W} = \boldsymbol{W}^{\top}$; (iii) (Null space property) null$\{\boldsymbol{I} - \boldsymbol{W}\} = \text{span}\{\mathbf{1}\}$; (iv) (Spectral property) $\boldsymbol{I} \succeq \boldsymbol{W} \succ -\boldsymbol{I}$.* Under these properties, the eigenvalues of $\boldsymbol{W}$ satisfies $1 = \lambda_1(\boldsymbol{W}) > \lambda_2(\boldsymbol{W}) \geq \cdots \geq \lambda_m(\boldsymbol{W}) > -1$. $\lambda := \max\{|\lambda_2(\boldsymbol{W})|, |\lambda_m(\boldsymbol{W}))|\}$ and $1 - \lambda \in (0, 1]$ is a spectral gap of $\boldsymbol{W}$ measuring the speed of communication parameters converge to their average value (Sun et al., 2022).

---

**Algorithm 1:** Local updating for PFL.

**Input** : Local steps $K$, local learning rate $\eta_u$ and $\eta_v$, initialize $u_{i,0}^t = u^t$, and $v_{i,0}^t = v_i^t$.

**Output** : For each client, locally update $u_i^{t+1}, v_i^{t+1}$.

1 **for** *local update round $k = 0, 1, ..., K_v - 1$* **do**
2 $\quad v_{i,k+1}^t \leftarrow v_{i,k}^t - \eta_v \nabla_v f_i(u_{i,0}^t, v_{i,k}^t, \xi_{i,k}^t)$.
3 **end**
4 **for** *local update round $k = 0, 1, ..., K_u - 1$* **do**
5 $\quad u_{i,k+1}^t \leftarrow u_{i,k}^t - \eta_u \nabla_u f_i(u_{i,k}^t, v_{i,K_v}^t, \xi_{i,k}^t)$.
6 **end**
7 $u_i^{t+1} \leftarrow u_{i,K_u}^t, v_i^{t+1} \leftarrow v_{i,K_v}^t$.

---

**Algorithm 2:** C-PFL and D-PFL.

**Input** : Total communication rounds $T$, number of selected clients $n$, initial the shared and personal variables $u^0, \mathbf{v}^0 = \{v_i^0\}_{i=0}^n$.

**Output** : Personal solution $u^T$ and $\mathbf{v}^T = \{v_i^T\}_{i=0}^n$.

1 C-PFL:
2 **for** communication round $t = 0$ **to** $T - 1$ **do**
3 $\quad$ Sample clients $|S^t| = n$ uniformly randomly and distribute the shared variables $u^t$.
4 $\quad$ **for** *client $i \in S^t$ in parallel* **do**
5 $\quad\quad u_i^{t+1}, v_i^{t+1} \leftarrow$ Local updating $(u_i^t, v_i^t)$
6 $\quad$ **end**
7 $\quad u^{t+1} \leftarrow \frac{1}{n} \sum_{i \in s^t} u_i^{t+1}$.
8 **end**
9 D-PFL:
10 **for** communication round $t = 0$ **to** $T - 1$ **do**
11 $\quad$ **for** *client $i \in [m]$ in parallel* **do**
12 $\quad\quad u_i^{t+1}, v_i^{t+1} \leftarrow$ Local updating $(u_i^t, v_i^t)$
13 $\quad$ **end**
14 $\quad$ Receive shared variables $u_i^{t+1}$ with matrix $W$: $u_{i,0}^{t+1} \leftarrow \sum_{l \in \mathcal{G}(i)} w_{i,l} u_i^{t+1}$.
15 **end**

---

### 3.2 Stability and Excess Risk

**Generalization Stability.** Recalling the unseen data distribution $\mathcal{D}_i$ in the population risk function in Formula (1), we select the sample $\xi_i$ from the local datasets $\mathcal{S}_i$ and estimate the expectation to represent the real distribution. The training process is rewritten as the Empirical Risk Minimization:

$$\min_{u,V} \quad f(u, V) := \frac{1}{m} \sum_{i=1}^m f_i(u, v_i), \quad where \quad f_i(u, v_i) = \frac{1}{\mathcal{S}} \sum_{\xi_i \in \mathcal{S}_i} [f_i(u, v_i; \xi_i)]. \quad (2)$$

Assuming the joint datasets of local dataset $\mathcal{S}_i$ as $\mathcal{S}$, we consider a solution $\mathcal{A}(\mathcal{S})$ of a specific algorithm $\mathcal{A}$ trained on $\mathcal{S}$, the generalization error between the population risk in (1) and empirical risk in (2) can be defined as $\varepsilon_G = \mathbb{E}_{\mathcal{S},\mathcal{A}}[F(\mathcal{A}(\mathcal{S})) - f(\mathcal{A}(\mathcal{S}))] = \mathbb{E}[F(u, V) - f(u, V)]$. This joint impact caused by both the algorithm $\mathcal{A}$ and the datasets $\mathcal{S}$ may cause a bad performance from a well-trained model on the testing dataset, which is called overfitting. Motivated by the previous studies in Hardt et al. (2016), we use the uniform stability bound to explore the generalization performance of PFL.

**Definition 2.** *(Uniform Stability) Considering a new joint dataset $\widetilde{S}$, which differs from the vanilla dataset $S$ at most one data sample $z$. The $\varepsilon$-uniformly stability for algorithm $\mathcal{A}$ is defined as below:*

$$\sup_{z_j \sim \{\mathcal{D}_i\}} \mathbb{E}[f(u, V; z_j) - f(\widetilde{u}, \widetilde{V}; z_j)] \leq \epsilon. \tag{3}$$

*The generalization error can be bound by $\epsilon_G \leq \epsilon$ (Hardt et al., 2016), if the algorithm $\mathcal{A}$ satisfies the $\varepsilon$-uniformly stability.*

**Excess Risk.** Considering $(u^*, V^*)$ as the optimal model that can be achieved by the algorithm $\mathcal{A}$ on the dataset $S$, the real test performance $\mathbb{E}[F(\mathcal{A}(S))]$ can be measured by the excess risk as follows:

$$\mathcal{E}_E = \mathbb{E}[F(\mathcal{A}(S))] - \mathbb{E}[f(u^*, V^*)]$$
$$\leq \underbrace{\mathbb{E}[F(u, V) - f(u, V)]}_{\mathcal{E}_G:\ generalization\ error} + \underbrace{\mathbb{E}[f(u, V) - f(u^*, V^*)]}_{\mathcal{E}_O:\ optimization\ error}. \tag{4}$$

Actually, if the optimal parameter $(u^*, V^*)$ could fit the personalized datasets well, the loss function $\mathbb{E}[f(u^*, V^*)]$ will tend to zero when the training time is large enough. Therefore, the real risk of the well-trained model $(u, V)$ on the test datasets can be bounded by the generalization and optimization error. $\mathcal{E}_G$ represents the performance risk of $(u, V)$ between the training datasets and testing datasets, while $\mathcal{E}_O$ represents the empirical risk between the theoretical optimum $(u^*, V^*)$ and the obtained one $(u, V)$. Utill now, most existing studies about PFL focus on the optimization error $\varepsilon_O$ of general C-PFL and D-PFL, but there is still little work to discuss their generalization nature. To further understand the optimization progress of the algorithm design and the iteration nature of personalization, we provide a comprehensive analysis of their excess risks.

## 3.3 BASIC ASSUMPTIONS

**Assumption 1 (Smoothness).** *For each client $i = \{1, \ldots, m\}$, the function $F$ is continuously differentiable. There exist constants $L_u, L_v, L_{uv}, L_{vu}$ such that for each client $i = \{1, \ldots, m\}$:*

- $\nabla_u f_i(u_i, v_i)$ *is $L_u$–Lipschitz with respect to $u_i$ and $L_{uv}$–Lipschitz with respect to $v_i$*

- $\nabla_v f_i(u_i, v_i)$ *is $L_v$–Lipschitz with respect to $v_i$ and $L_{vu}$–Lipschitz with respect to $u_i$.*

**Assumption 2 (Bounded Variance).** *The stochastic gradients in both C-PFL and D-PFL have bounded variance. That is to say, for all $u_i$ and $v_i$, there exist constants $\sigma_u$ and $\sigma_v$ such that:*

$$\mathbb{E}\left[\left\|\nabla_u f_i(u_i, v_i; \xi_i) - \nabla_u f_i(u_i, v_i)\right\|^2\right] \leq \sigma_u^2, \tag{5}$$

$$\mathbb{E}\left[\left\|\nabla_v f_i(u_i, v_i; \xi_i) - \nabla_v f_i(u_i, v_i)\right\|^2\right] \leq \sigma_v^2. \tag{6}$$

**Assumption 3 (Partial Gradient Diversity).** *There exists a constant $\delta_u^2$ that reflects the data heterogeneous degree:*

$$\left\|\nabla_u f_i(u, v_i) - \nabla_u f_i(u, V)\right\|^2 \leq \delta_u^2, \ \forall u, \ V.$$

**Assumption 4 (G-Lipschitz).** *For $\mathcal{A}(S), \mathcal{A}(\widetilde{S}) \in \mathbb{R}^d$ which are well trained by an $\epsilon$-uniformly stable algorithm $\mathcal{A}$ on dataset $S$ and $\widetilde{S}$, the personalized objective $f(u, V)$ satisfies G-Lipschitz continuity between them:*

$$\|f(\mathcal{A}(S)) - f(\mathcal{A}(\widetilde{S}))\| \leq G\|\mathcal{A}(S) - \mathcal{A}(\widetilde{S})\|. \tag{7}$$

Assumptions 1, 2 and 3 are mild and commonly used in the convergence analysis of FL and PFL (Liu et al., 2024; Chen et al., 2023; Shi et al., 2023a; Li et al., 2023a; Shi et al., 2023d; Pillutla et al., 2022; Sun et al., 2022; Reddi et al., 2021; Li et al., 2025). Assumption 4 is a variant of the vanilla Lipschitz continuity assumption, which is widely used in the uniform stability analysis (Elisseeff et al., 2005; Hardt et al., 2016; Zhou et al., 2021; Zhu et al., 2022; Sun et al., 2023c; 2024a; Li et al., 2024a;b).

## 4 THEORETICAL ANALYSIS

In this section, we present the generalization analysis and the excess risk analysis for both C-PFL and D-PFL. We state the main theoretical results and discussions as follows.

## 4.1 STABILITY AND EXCESS RISK FOR CENTRALIZED PERSONALIZATION

**Theorem 1 (Stability of C-PFL).** *Under Assumption [1]$\sim$[4], let the active ratio per communication round be $n/m$, and assume the learning rates satisfy $\eta_u = \mathcal{O}\left(\frac{1}{tK_u+k}\right) = \frac{\mu_u}{tK_u+k}$ and $\eta_v = \mathcal{O}\left(\frac{1}{tK_v+k}\right) = \frac{\mu_v}{tK_v+k}$. They decay per iteration $\tau = tK + k$, where $\mu_u$ and $\mu_v$ are the specific constants and satisfy $\mu_u \leq \frac{1}{L_u}$ and $\mu_v \leq \frac{1}{L_v}$. Let $U = sup_{u,v_i,z}f(u,v_i;z)$, then the generalization bound of C-PFL satisfies:*

$$
\begin{aligned}
&\mathbb{E}\left[\|f(u^T, V^T; z_j) - f(\widetilde{u}^T, \widetilde{V}^T; z_j)\|\right] \\
&\leq \frac{nU\tau_0}{mS} + \left(\frac{TK_u}{\tau_0}\right)^{\mu_u L_u} \frac{2G(\sigma_u + \delta_u)}{mSL_u} + \left(\frac{TK_v}{\tau_0}\right)^{\mu_v L_v}\left(1 + \frac{L_{uv}}{L_u}(\frac{TK_u}{\tau_0})^{\mu_u L_u}\right)\frac{2G\sigma_v}{mSL_v}.
\end{aligned}
\tag{8}
$$

*To simplify subsequent analysis, we assume $\mu L = \max\{\mu_u L_u, \mu_v L_v\}$ and $K = \max\{K_u, K_v\}$. By selecting $\tau_0 = \left[\frac{2G((\sigma_u+\delta_u)L_v+\sigma_v L_u)}{nUL_uL_v}\right]^{\frac{1}{1+\mu L}}(TK)^{\frac{\mu L}{1+\mu L}}$, we can minimize the bound with $\tau_0$:*

$$
\mathbb{E}\left[\|f(u^T, V^T; z_j) - f(\widetilde{u}^T, \widetilde{V}^T; z_j)\|\right] \leq \frac{4}{mS}\left[\frac{G((\sigma_u+\delta_u)L_v + \sigma_v L_u)}{L_uL_v}\right]^{\frac{1}{1+\mu L}}(nUTK)^{\frac{\mu L}{1+\mu L}}. \tag{9}
$$

**Remark 1 (Influencal factors of C-PFL).** *From the stability-based results above, severe data heterogeneity (larger gradient diversity $\delta_u$), more selected clients $n$ and more local epochs $K_u$ and $K_v$ increase the time of training on only different samples, which leads to a larger generalization gap and worse generalization performance. In contrast, the generalization gap can be alleviated with more total clients $m$ and the number of samples $S$ involved.*

**Remark 2 (Special cases of C-PFL).** *If we remove all personal variables $v_i$, the problem ([2]) degenerates to the classical FL problem FedAvg. The stability reduces to $\mathcal{O}\left((nK_uT)^{\frac{\mu_u L_u}{1+\mu_u L_u}}/m\right)$ by removing the $K_v$ and $\sigma_v$ in the result, which is compatible with the upper bound $\mathcal{O}\left((nKT)^{\frac{\mu L}{1+\mu L}}/m\right)$ of the stability of central FL algorithm FedAvg ([Sun et al., 2023c]) with multiple local update. That is to say, the upper bound of the stability is only related to the training paradigm, no matter whether training for the consensus model or the personalized models. This finding builds the bridge between the stability of FL and PFL. If we remove all shared variables $u$, the stability of C-PFL can be reduced to $\mathcal{O}\left((nK_vT)^{\frac{\mu_v L_v}{1+\mu_v L_v}}/mS\right)$, which is the stability bound of the whole FL system with partial participation ratio $n/m$ and local updates $K_v$. For further consideration, we set full participation $n/m = 1$ and only one local update $K_v = 1$, our results can degrade to $\mathcal{O}\left(T^{\frac{\mu L}{1+\mu L}}/S\right)$ on each client, which is aligned with the vanilla SGD in [Hardt et al. (2016)].*

**Personalization performs better than no personalization and Pure Local Training.** Considering the impact of data heterogeneity, we list the comparison of generalization bounds for FL, C-PFL, and Pure Local Training in Table [2]. When the data heterogeneity is more severe, PFL significantly improves generalization capabilities from no personalized methods by eliminating the influence of the gradient diversity and outperforms Pure Local Training through collaboration with other clients. Specifically, for the consensus model $w$ in FL, we assume that the global gradient diversity mainly from heterogeneity satisfies $\frac{1}{m}\sum_{i=1}^m\|\nabla F_i(w_i) - \nabla F(w)\|^2 \leq \delta_g^2$. Correspondingly, for the personalized model $(u, v_i)$, the heterogeneity from the shared variables $\delta_u$ satisfy $\frac{1}{m}\sum_{i=1}^m\|\nabla_u F_i(u, v_i) - \nabla_u F(u, V)\|^2 \leq \delta_u^2$ shown in the Assumption [3]. When the data distribution is non-iid between each client ($\delta_g, \delta_u \neq 0$), PFL performs better than FL since the gradient diversities satisfy $\delta_u^2 \leq \delta_g^2$ during personalized training. As for the generalization performance of Pure Local Training, though it is not affected by the

Table 2: Comparison with FL, PFL, Pure Local Training.

| Algorithm | Generalization Bound |
|---|---|
| FL | $\mathcal{O}\left(\frac{nU\tau_0}{mS} + \left(\frac{TK}{\tau_0}\right)^{\mu L}\frac{2G(\sigma+\delta_g)}{mSL}\right)$ |
| PFL | $\mathcal{O}\big(\frac{nU\tau_0}{mS} + \left(\frac{TK_u}{\tau_0}\right)^{\mu_u L_u}\frac{2G(\sigma_u+\delta_u)}{mSL_u}$ $+ \left(\frac{TK_v}{\tau_0}\right)^{\mu_v L_v}(1 + \frac{L_{uv}}{L_u})\frac{2G\sigma_v}{SL_v}\big)$ |
| Local | $\mathcal{O}\left(\frac{U\tau_0}{S} + \left(\frac{TK}{\tau_0}\right)^{\mu L}\frac{2G\sigma}{SL}\right)$ |

gradient diversities from data heterogeneity, the upper bound may be larger without the collaboration among $m$ clients, which is consistent with the empirical experience. This analysis builds up the bridge among FL, PFL and Pure Local Training, which first demonstrates the effectiveness and necessity of personalized design from the generalization perspective.

**Remark 3** (Comparisons with generalization bounds of C-PFL). *The generalization bounds compared in Table 4 in Appendix B calculate the complexity of the PAC problem in infinite space as the generalization error. Although considering the nature of the learning problem, they cannot analyze the impact of algorithm design and personalized iterative nature. Therefore, we highlight our contributions as follows: 1) conduct the generalization analysis in the non-convex condition, which is based on the more realistic assumptions adapted to the neural networks; 2) analyze the effectiveness of algorithm design and hyperparameters selection; 3) illustrate the error propagation between model aggregation and local training with the iteration nature.*

**Corollary 1** (**Excess risk of C-PFL.**). *Assuming that the number of dataset samples $S$ is fixed and combining with the convergence bounds of $\varepsilon_O \leq \mathbb{E}\left[f(w^T) - f(w^\star)\right] = \mathcal{O}\left(1/\sqrt{T}\right)$ proposed in Pillutla et al. (2022), the excess risk of C-PFL satisfies that $\mathcal{E}_E \leq \mathbb{E}[F(\mathcal{A}(\mathcal{S}))] - \mathbb{E}[f(w^*)] = \mathcal{O}\left(1/\sqrt{T} + (nKT)^{\frac{\mu L}{1+\mu L}}/m\right)$.*

**Remark 4** (**Influential factors of the excess risks for C-PFL**). *Assuming that the smoothness constants $L$, the gradient variance $\sigma_u$ and $\sigma_v$, the gradient diversity $\delta_u$, and the total client number $m$ are fixed in a specific analysis, the excess risk of C-PFL is decided by the number of active clients $n$, the local intervals $K_u$ and $K_v$, the total communication rounds $T$. Therefore, we can adjust the hyperparameters $n$, $K_u$, $K_v$ and $T$ to optimize the testing performance during training. The preferred choice of the number of active clients $n$ and the local intervals $K_u$ and $K_v$ are the same as that in the stability analysis, but increasing the communication rounds $T$ leads to better convergence but worse generalization performance due to the training overfitting. That is to say, the better performance for C-PFL is the trade-off between optimization and generalization.*

## 4.2 STABILITY AND EXCESS RISK FOR DECENTRALIZED PERSONALIZATION

In this section, we first provide the stability analysis of PFL with the peer-to-peer communication in the non-convex objectives. Then we combine its convergence performance to obtain the excess risk.

**Theorem 2** (**Stability for D-PFL**). *Under Assumption $1 \sim 4$, let clients communicate with each other in a peer-to-peer manner, and assume the learning rates satisfy $\eta_u = \mathcal{O}\left(\frac{1}{tK_u+k}\right) = \frac{\mu_u}{tK_u+k}$ and $\eta_v = \mathcal{O}\left(\frac{1}{tK_v+k}\right) = \frac{\mu_v}{tK_v+k}$. They decay per iteration $\tau = tK + k$, where $\mu_u$ and $\mu_v$ are the specific constants and they satisfy $\mu_u \leq \frac{1}{L_u}$ and $\mu_v \leq \frac{1}{L_v}$. Let $U = \sup_{u,v_i,z} f(u,v_i;z)$, then the generalization bound of D-PFL satisfies:*

$$
\begin{aligned}
\mathbb{E}\left[\|f(u^T, V^T; z_j) - f(\widetilde{u}^T, \widetilde{V}^T; z_j)\|\right] \leq & \frac{U\tau_0}{S} + \frac{2(\sigma_u + \delta_u)G}{SL_u}\left(\frac{1 + 6\sqrt{m}\kappa_\lambda}{m}\right)\left(\frac{TK_u}{\tau_0}\right)^{\mu_u L_u} + \\
& \frac{12\sqrt{m}\kappa_\lambda \sigma_v L_{uv}}{mSL_v L_u}\left(\frac{TK_u}{\tau_0}\right)^{\mu_v L_v} + \frac{2\sigma_v G}{SL_v}\left(\frac{TK_v}{\tau_0}\right)^{\mu_v L_v}.
\end{aligned}
\tag{10}
$$

*where $\kappa_\lambda = \left(\frac{\alpha}{e}\right)^\alpha \frac{1}{\lambda\left(\ln\frac{1}{\lambda}\right)^\alpha} + \frac{2^\alpha}{(1-\alpha)e\lambda\ln\frac{1}{\lambda}} + \frac{2^\alpha}{\lambda\ln\frac{1}{\lambda}}$ and $\lambda$ are the widely used coefficient to measure different communication connections.*

*To simplify subsequent analysis, we assume $\mu L = \max\{\mu_u L_u, \mu_v L_v\}$ and $K = \max\{K_u, K_v\}$. By selecting $\tau_0 = \left[\frac{2G(\sigma_u+\delta_u)L_v^2(1+6\sqrt{m}\kappa_\lambda)+2G\sigma_v L_u L_{uv}(m+6\sqrt{m}\kappa_\lambda)}{UmL_u L_v^2}\right]^{\frac{1}{1+\mu L}}(TK)^{\frac{\mu L}{1+\mu L}}$, we can minimize the upper generalization bound:*

$$
\begin{aligned}
& \mathbb{E}\left[\|f(u^T, V^T; z_j) - f(\widetilde{u}^T, \widetilde{V}^T; z_j)\|\right] \\
& \leq \frac{4}{S}\left[\frac{(\sigma_u + \delta_u)G}{L_u m}(1 + 6\sqrt{m}\kappa_\lambda) + \frac{\sigma_v G}{L_v}(1 + \frac{6\sqrt{m}\kappa_\lambda L_{uv}}{mL_u})\right]^{\frac{1}{1+\mu L}}(UTK)^{\frac{\mu L}{1+\mu L}}.
\end{aligned}
\tag{11}
$$

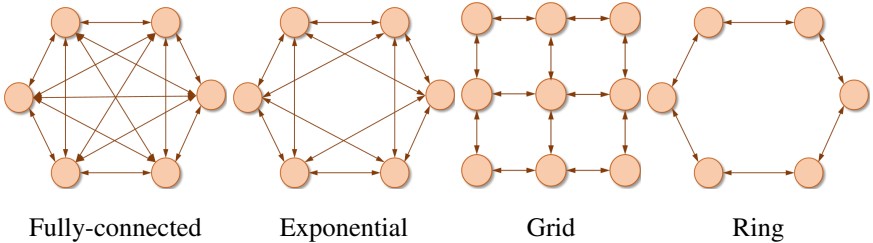

Figure 1: Illustration of various network topologies in DFL.

**Remark 5** (**Influential factors of the decentralized stability**). *The stability of D-PFL is impacted by the number of samples $S$, total clients $m$, total iterations $TK_u$ and $TK_v$, data heterogeneity $\delta_u$ as well in the analysis of C-PFL. It is worth noting that it is also decided by the communication topologies $\kappa_\lambda$ in decentralized FL.Table 3 below and Figure 1 show the different topological diagrams and their properties.*

For $\kappa_\lambda = \left(\frac{\alpha}{e}\right)^\alpha \frac{1}{\lambda(\ln\frac{1}{\lambda})^\alpha} + \frac{2^\alpha}{(1-\alpha)e\lambda\ln\frac{1}{\lambda}} + \frac{2^\alpha}{\lambda\ln\frac{1}{\lambda}}$, when $\lambda \to 1$, the upper bound for $\kappa_\lambda$ is mainly decided by $\mathcal{O}\left(1/(\lambda\left(\ln\frac{1}{\lambda}\right))\right)$. And when $\lambda \to 0$, the upper bound for $\kappa_\lambda$ is mainly decided by $\mathcal{O}\left(1/(\lambda\left(\ln\frac{1}{\lambda}\right)^\alpha)\right)$. From the analysis above, we can clearly see that denser communication topology with a smaller $\kappa_\lambda$ leads to better generalization performance in D-PFL. Therefore, the fully connected topology achieves the best gener-

Table 3: $\kappa_\lambda$ and Spectral Gap $1 - \lambda$ of communication topologies(Sun et al., 2023c; Zhu et al., 2024).

| Network Topology | $\kappa_\lambda$ | Spectral Gap $1 - \lambda$ |
|---|---|---|
| Fully-connected | 0 | 1 |
| Disconnected | 1 | 0 |
| Ring | $\mathcal{O}(m^2)$ | $\approx 16\pi^2/3m^2$ |
| Grid | $\mathcal{O}(mlnm)$ | $\mathcal{O}(1/mlog_2(m))$ |
| Exponential | $\mathcal{O}(lnm)$ | $2/(1 + log_2(m))$ |

alization performance of shared variables and is compatible with the central ones.

**Remark 6** (**Special cases of D-PFL**). *If we remove all personalized variables $v_i$, the problem (2) degenerates to the classical DFL algorithm DFedAvg. By removing all personal constants in the proof, the stability of D-PFL reduces to $\mathcal{O}\left((1 + 6\sqrt{m}\kappa_\lambda/m)^{\frac{1}{1+\mu_u L_u}}(K_u T)^{\frac{\mu_u L_u}{1+\mu_u L_u}}\right)$, which is compatible with the upper bound of the stability of decentralized federated learning DFedAvg in Sun et al. (2023c) of $\mathcal{O}\left((1 + 6\sqrt{m}\kappa_\lambda/m)^{\frac{1}{1+\mu L}}(KT)^{\frac{\mu L}{1+\mu L}}\right)$. The degradation analysis of the shared variables $u$ is the same as in C-PFL.*

**Remark 7** (**Comparisons with generalization bounds of D-PFL**). *The mere generalization analysis of D-PFL can be seen for Dis-PFL in Dai et al. (2022b), which acquires a generalization lower bound through the lens of differential privacy with the inspiration in He et al. (2021). It describes the relationship between the remaining model and generalization performance at each iteration point and suggests that a more sparse network leads to better generalization performance. However, it lacks of an understanding of the algorithm design, the impacts with different training parameters as well as the communication topologies in decentralized scenarios.*

**Corollary 2** (**Excess risk of decentralized partial model personalization**). *Assuming that the number of dataset samples $S$ is fixed and combining with the convergence rates of $\varepsilon_O \leq \mathbb{E}\left[f(w^T) - f(w^\star)\right] = \mathcal{O}\left(1/(1-\lambda)^2\sqrt{T}\right)$ provided in Shi et al. (2023a), the excess risk of D-PFL satisfies $\mathcal{E}_E \leq \mathbb{E}[F(\mathcal{A}(\mathcal{S}))] - \mathbb{E}[f(w^*)] = \mathcal{O}\left(1/(1-\lambda)^2\sqrt{T} + (1 + 6\sqrt{m}\kappa_\lambda/m)^{\frac{1}{1+\mu L}}(KT)^{\frac{\mu L}{1+\mu L}}\right)$.*

**Remark 8** (**Influential factors of the decentralized excess risks**). *Our analysis shows that the excess risk of D-PFL is highly influenced by the number of the local interval $K_u$ and $K_v$, the total communication rounds $T$, the total clients $m$, the smoothness constants $L$, the gradient variance $\sigma_{u/v}$, the data heterogeneity $\delta_u$, and the communication topologies $\lambda$ and $\kappa_\lambda$. Assuming that the total client number $m$ is fixed under the specific algorithm and data distribution (with the fixed smoothness constants $L$, the gradient variance $\sigma$ and the data heterogeneity $\delta_u$), we can adjust the communication networks $\lambda$ and $k_\lambda$, local interval $K_u$ and $K_v$ to optimize the testing performance.*

*A denser connection (smaller $\kappa_\lambda$ and smaller $\frac{1}{1-\lambda}$) means better convergence performance and generalization performance, but it brings more communication cost. The better choice for local intervals $K_u$ and $K_v$ are the same as that in stability analysis. And the better testing performance of D-PFL is a trade-off between the convergence errors and the generalization errors.*

**Remark 9** (**Comparisions between the C-PFL and D-PFL**). *From the comparison between Theorem 1 and Theorem 2, we can clearly see that C-PFL always converges and generalizes better than D-PFL in theoretical analysis. The centralized one largely reduces the propagation of the generalization error, which benefits from the regular averaging on a global server for a better consensus of the shared variables and leads to better generalization. However, to achieve a more reliable performance, the number of active clients $n$ in C-PFL must satisfy at least a polynomial order of $m$. Also, the communication burden on the central server becomes a big challenge in the training process. It means that the high communication costs are unavoidable when the whole federated system $m$ gets larger. This conclusion is also consistent with the generalization analysis of the typical FL and DFL in Sun et al. (2023c). Therefore, the suitable choice between C-PFL and D-PFL or the choice of different communication topologies in real scenarios is a trade-off among communication ability, communication cost and personalized performance.*

## 5 EXPERIMENTS

In this section, we conduct extensive experiments to verify the theoretical findings. We first introduce the typical setting for experiments, then present the empirical results and corresponding analysis.

### 5.1 EMPIRICAL SETUP

We conduct the experiments on CIFAR-10 datasets (Krizhevsky et al., 2009) in the Dirichlet distribution (Non-IID $\alpha = 0.3$) with ResNet-18 (He et al., 2016) and CIFAR-100 datasets in the Pathological distribution (Non-IID $c = 20$) with VGG-11 (Simonyan & Zisserman, 2014) for C-PFL and D-PFL. Experiments on CIFAR-100 are in Appendix C.2. To verify the impacts of the key hyperparameters, we follow Hardt et al. (2016) and study the parameter distance when disturbing only one data in Figure 2a, the generalization gap of the difference between training and testing error in Figure 2b. We explore the impact of the four factors: 1) Local Learning Epochs, 2) Local Learning Rates, 3) Client Fraction / Communication topology, 4) Total Client Number. We keep the same sets for the other factors for fairness. More implementations can be seen in Appendix C.

### 5.2 EMPIRICAL ANALYSIS

**Both less local learning epochs and lower learning rates lead to better generalization performance, but they affect the convergence speed more seriously.** We discuss this phenomenon in the first two columns for Learning Epoch and Learning Rate in Figure 2a-2b. Increasing local learning epochs and learning rates means amplifying the model distance when learning on different samples. It brings about a larger generalization error and more severe fluctuation.

**More client participation and denser network connections in each communication round enlarge the generalization gap, but they speed up the convergence rate to the same extent.** We discuss this phenomenon in the third column for Client Selection and Communication Topology in Figure 2a-2b. Increasing the fraction of client selection and choosing denser connection topologies means more frequency in learning unique samples, which enlarges the generalization gap between the two models with the disturbed datasets. This is aligned with our theoretical findings in Theorem 1, 2. Thus, there is a trade-off between communication costs and personalized performance in real life.

**A larger total participation of clients and a smaller number of local training samples increase the generalization error and reduce the convergence speed simultaneously.** We discuss this phenomenon in the fourth column in Figure 2a-2b. Since the total data number on the dataset remains the same, bigger participation clients mean fewer training samples per one. The generalization gaps get worse with the number of clients increasing in Figure 2b.

**C-PFL outperforms D-PFL in both generalization and convergence when their upper communication bandwidths are at the same level.** We discuss this difference in the comparisons of each line in Figure 2a-2b. Maintaining the maximum communication capacity of the busiest node, the central

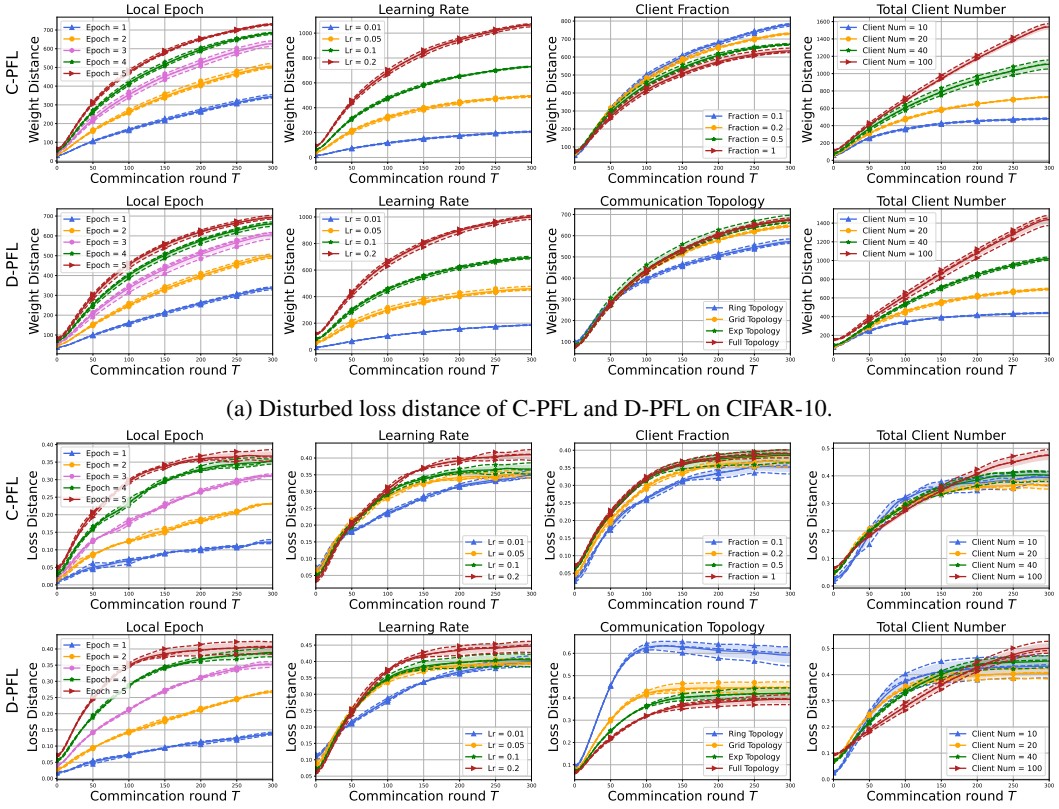

(a) Disturbed loss distance of C-PFL and D-PFL on CIFAR-10.

(b) Testing and training loss distance of C-PFL and D-PFL on CIFAR-10.

Figure 2: Empirical results of C-PFL (first line) and D-PFL (second line) on CIFAR-10.

server in C-PFL mitigates the inconsistencies driven by the updates on different samples, consistent with our theoretical results.

## 6 CONCLUSION

In this paper, we develop the first algorithm-dependent generalization analysis and the excess risk analysis for PFL in both centralized and decentralized scenarios under non-convex conditions. It builds up the bridge among PFL, FL and Pure Local Training, and demonstrates the effectiveness of the personalized design from the generalization perspective. Compared with the previous works, this analysis reveals the impact of algorithm design and hyperparameter selection on the iteration properties. Combined with the convergence errors, we obtain the excess risk analysis for PFL. Various experiments verify our theoretical findings.

**Limitation.** There are numerous avenues for future works: 1) Improve the generalization bounds for PFL with the more advanced stability methods; 2) Discuss the lower bound and tightness of the generalization of PFL to obtain the optimal training strategies for personalized training.

**Acknowledgment**: This work is supported by the STI 2030 Major Projects (No. 2021ZD0201405), Shenzhen Basic Research Project (Natural Science Foundation) Basic Research Key Project (NO. JCYJ20241202124430041), Open Research Fund from Guangdong Laboratory of Artificial Intelligence and Digital Economy (SZ) (NO. GML-KF-24-23), CCF-Baidu Open fund (NO. CCF-Baidu202413), Guangdong Basic and Applied Basic Research Foundation (2023B0303000010), Shenzhen Science and Technology Program (KJZD20240903095700001), National Natural Science Foundation of China (No. 62402499).

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

In this part, we provide the supplementary materials to prove the main theorem.

- **Appendix** A: Related Work about PFL.
- **Appendix** B: Detailed Comparisons of Generalization.
- **Appendix** C: Implementation Details and Results for Experiments.
- **Appendix** D: Generalization Bounds for C-PFL and D-PFL.

# A RELATED WORK ABOUT PFL.

**Personalized Federated Learning.** FL aims to improve model performance through collaboration among users (Sun et al., 2023b;a; Shi et al., 2023b;d). PFL aims to produce the optimal personalized models for each client via model decoupling (Arivazhagan et al., 2019; Collins et al., 2021), knowledge distillation (Li & Wang, 2019; Lin et al., 2020), multi-task learning (Huang et al., 2021; Shoham et al., 2019), model interpolation (Deng et al., 2020; Diao et al., 2020) and clustering (Ghosh et al., 2020; Sattler et al., 2020). More details can be referred to the PFL survey (Tan et al., 2022). Among them, the model decoupling method Partial Model Personalization, which divides the model into shared variables and personal variables, has proved to achieve better performance than full model personalization with fewer shared parameters. LG-FedAvg (Liang et al., 2020) relieves the data variance and device variance with jointly learning compact local representations on each device and a global model across all devices. FedPer (Arivazhagan et al., 2019), FedRep (Collins et al., 2021) and FedBABU (Oh et al., 2021) set the feature extractor as the shared variable and the linear classifiers as the personal variables. They are different from the optimization progress between the shared representation and the private linear parts. Fed-RoD (Chen & Chao, 2021) trains a global full model and many private classifiers with empirical risk minimization and balanced risk minimization. Most theoretical analyses for Partial Model Personalization mainly focus on their convergence performance. FedSim & FedAlt (Pillutla et al., 2022) provide the convergence analyses in the general non-convex setting, while FedAvg-P & Scaffold-P (Chen et al., 2023) achieve linear speedup respecting the number of the local steps. DFedSGPSM Li et al. (2023b) investigates the convergence of DFL combined with SAM under non-convex conditions. DFedPGP (Liu et al., 2024) presents the decentralized convergence bound in non-convex conditions under the directed graph and PFedDST (Fan et al., 2025) proposes the selection method for this directed cooperation, while DFedMDC & DFedSMDC (Shi et al., 2023a) focus on the convergence with the undirected network following (Sun et al., 2022; Shi et al., 2023c; Li et al., 2023a; 2024a).

# B DETAILED COMPARISON OF GENERALIZATION.

Table 4: Main results on the upper generalization bounds of PFL.

| Algorithm | Generalization Bound | $T$ | $K$ | $\eta$ | $n$ | $m$ |
|---|---|---|---|---|---|---|
| APFL, (Deng et al., 2020) | $\mathcal{O}\left(2(1-\alpha_i)^2\left(\hat{\mathcal{L}}_{\overline{\mathcal{D}}}(\bar{h}^*)+B\left\|\overline{\mathcal{D}}-\mathcal{D}_i\right\|_1+C\sqrt{(d+\log(1/\delta))/N}\right)\right)$ $+\mathcal{O}\left(2\alpha_i^2(\mathcal{L}_{\mathcal{D}_i}(h_i^*)+2C\sqrt{(d+\log(1/\delta))/S_i}+G\lambda_{\mathcal{H}}(\mathcal{S}_i))\right)$ | ✗ | ✗ | ✗ | ✗ | ✓ |
| MAPPER, (Mansour et al., 2020) | $\mathcal{O}\left(2L\left(\sqrt{\frac{d_c}{m}\log\frac{em}{d_c}}+\sqrt{\frac{d_lp}{m}\log\frac{em}{d_l}}\right)+2\sqrt{\frac{\log\frac{1}{\delta}}{m}}\right)$ | ✗ | ✗ | ✗ | ✗ | ✓ |
| pFedBayes, (Zhang et al., 2022) | $\mathcal{O}\left(C_2m^{-\frac{2\beta}{2\beta+d}}\log^{2\delta'}(m)\right)$ | ✗ | ✗ | ✗ | ✗ | ✓ |
| FedAvg & LocalTraining, (Chen et al., 2021) | $\mathcal{O}\left(\frac{1}{N}+R^2\right) \quad \& \quad \mathcal{O}(m/N)$ | ✓ | ✓ | ✓ | ✓ | ✓ |
| C-PFL (Ours) | $\mathcal{O}\left(\frac{4}{N}\left[\frac{G(\sigma_uL_v+\sigma_vL_u)}{L_uL_v}\right]^{\frac{1}{1+\mu L}}(nUTK)^{\frac{\mu L}{1+\mu L}}\right)$ | ✓ | ✓ | ✓ | ✓ | ✓ |

Compared to the above generalization bounds for PFL, the proposed analysis has made the following progress: 1) conduct the generalization analysis in the non-convex condition, which is based on the more realistic assumptions adapted to the neural networks; 2) analyze the impacts of the algorithm design and the hyperparameters selection of the number of samples $S$, the number of selected clients $n$, total clients $m$, total iterations $TK_u$ and $TK_v$ and the local learning rates $\eta_u$ and $\eta_v$; 3) illustrate

the error propagation process between model aggregation and local training with the iteration nature, which provides a reference for the choice of early stopping points when training.

**APFL** is a typical PFL method based on model interpolation, which aims to find the optimal combination of the global model and the local model with the adaptive parameter $\alpha_i$ to achieve a better client-specific model. It derives the generalization bound of a mixture of local and global models with the analysis of VC dimension complexity. $S_i, i = 1, 2, ..., n$ is the number of training data at ith user, $N = m_1 + ... + m_n$ is the total number of all data, $\mathcal{S}_i$ to be the local training set drawn from $\mathcal{D}_i$, $\left\|\overline{\mathcal{D}} - \mathcal{D}_i\right\|_1 = \int_{\Xi} \left|\mathbb{P}_{(\boldsymbol{x},y)\sim\overline{\mathcal{D}}} - \mathbb{P}_{(\boldsymbol{x},y)\sim\mathcal{D}_i}\right| d\boldsymbol{x}dy$, is the difference between distributions $\overline{\mathcal{D}} = (1/n)\sum_{i=1}^{n} \mathcal{D}_i$ and $D_i$, and $h_i^* = \arg\min_{h\in\mathcal{H}} \mathcal{L}_{\mathcal{D}_i}(h)$.

**MAPPER** is also a model interpolation method combining local and global models to pursue the better personalized results. It derives the generalization bound with the analysis of Rademacher complexity. $\mathcal{H}_c$ is the hypotheses class for the central model, and $\mathcal{H}_l$ is the hypotheses class for the local models. $d_c$ is the pseudo-dimension of $\mathcal{H}_c$ and $d_l$ is the pseudo-dimension of $\mathcal{H}_l$. This bound only depends on the average number of samples and not the minimum number of samples.

**pFedBayes** is a novel PFL method via Bayesian variational inference. Each client uses the aggregated global distribution as prior distribution and updates its personal distribution by balancing the construction error over its personal data and the KL divergence with aggregated global distribution. It derives the generalization bound with the PAC-Bayes analysis. $\delta' > \delta > 1$, and $C_1, C_2 > 0$ are constants related to Hölder smooth $\beta$, the intrinsic dimension of data $d$, the number of hidden layers $L$, the widths of neural network are equalwidth $M$, the balance parameter $\zeta$ between personalization and global aggregation, and sample size of each client $n$.

**FedAvg and LocalTraining** are the most typical methods for FL and PFL. Though the generalization analysis in (Chen et al., 2021) is not designed based on the PFL definition and not in the non-convex condition, it concludes a surprising theorem that there exists a threshold of data heterogeneity to decide whether FedAvg or LocalTraining could achieve the minimax optimal for PFL. It derives the generalization bound for LocalTraining with uniform stability and the generalization bound for FedAvg with federated stability under strongly convex conditions. $m$ represenets the client index, and $N = n_1 + + n_m$ denotes the total number of training samples. $R^2 := \min_{\boldsymbol{w}\in\mathcal{W}} \sum_{i\in[m]} n_i \|\boldsymbol{w}_\star^{(i)} - \boldsymbol{w}\|^2/N$ measures the level of heterogeneity among clients (here $\|\|$ denotes the Euclidean distance). Compared with this analysis, we demonstrate how PFL outperforms FL and Pure Local Training from the perspective of data heterogeneity, show how hyperparameter selection affects the generalization bounds and discuss the theoretical performance under the more commonly used non-convex conditions.

## C APPENDIX FOR EXPERIMENTS.

### C.1 IMPLEMENTATION DETAILS FOR EXPERIMENTS.

According to Definition 2, we construct distributed neighboring dataset $\mathcal{S} = \{\mathcal{S}_1, ..., \mathcal{S}_m\}$ and $\widetilde{\mathcal{S}} = \{\widetilde{\mathcal{S}}_1, ..., \widetilde{\mathcal{S}}_m\}$, where each corresponding local dataset pair $(\mathcal{S}_i, \widetilde{\mathcal{S}}_i)$ only differs on one randomly selected data sample. Then we deploy the same initial model $(u, V)$ with its local dataset pair $(\mathcal{S}_i, \widetilde{\mathcal{S}}_i)$ to the local client $i$. To focus on the effect of the essential factors, the regularization methods such as weight decay, data augmentations and dropout are ignored to prevent unnecessary impacts (Zhu et al., 2024; Lei et al., 2021; Wang et al.). We keep the same experiment setting for all methods and perform 300 communication rounds. The number of client sizes is 20. The client sampling radio is 0.2 in C-PFL, while each client communicates with 4 neighbors in D-PFL accordingly. The batch size is 128 and the number of local epochs is 5. We set SGD (Robbins & Monro, 1951) as the base local optimizer with a learning rate $\eta = 0.1$. We ran each experiment 3 times with different random seeds and reported the mean accuracy with standard deviation for each method.

### C.2 MORE EXPERIMENTS RESULTS OF STABILITY ON CIFAR-100.

We explore the impact of the four factors on CIFAR-100 in Figure 3 and 4: 1) Local Learning Epochs, 2) Local Learning Rates, 3) Client Fraction / Communication Topology, and 4) Total Client

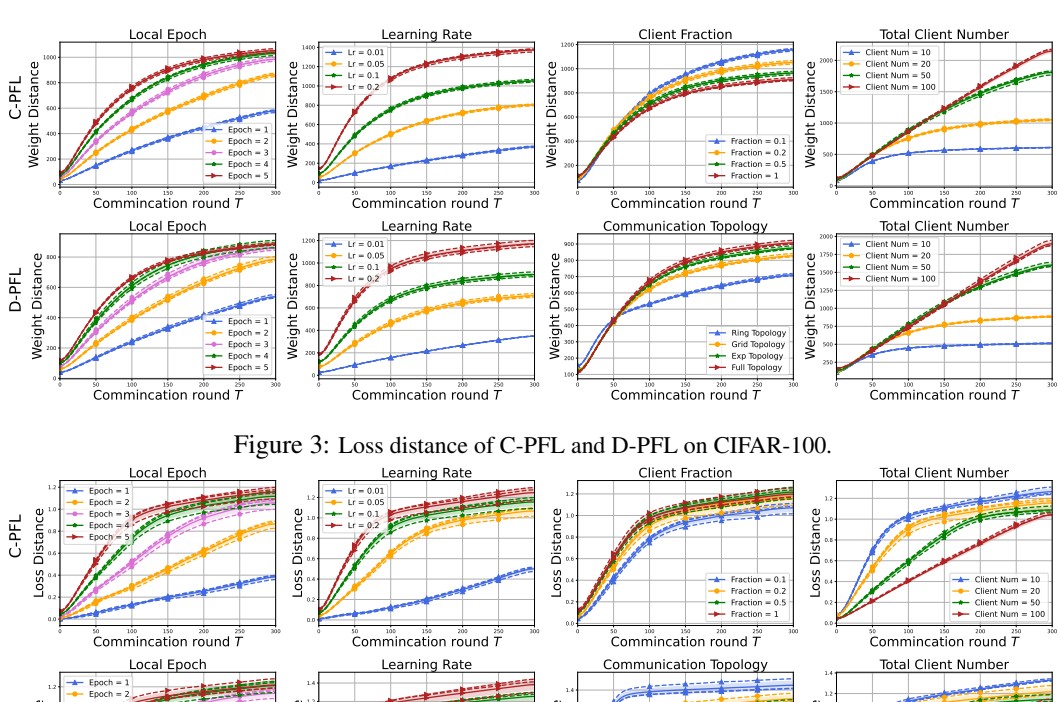

Figure 3: Loss distance of C-PFL and D-PFL on CIFAR-100.

Figure 4: Training loss distance of C-PFL and D-PFL on CIFAR-100.

Number. The empirical results on CIFAR-100 also verify that 1) Both less local learning epochs and lower learning rates lead to better generalization performance, but they affect the convergence speed more seriously; 2) More client participation and denser network connection in each communication round enlarge the generalization gap, but they speed up the convergence rate to the same extent; 3) A larger total participation clients and a smaller number of local training samples increase the generalization error and reduce the convergence speed simultaneously; 4) C-PFL outperforms D-PFL in both generalization performance and convergence performance when their upper communication bandwidths are at the same level.

### C.3 MORE EXPERIMENTS RESULTS OF TEST ACCURACY AND TRAIN LOSS ON CIFAR.

We explore the impact of the four factors in testing accuracy on CIFAR-10 in Figure 5 and CIFAR-100 in Figure 7, and training loss on CIFAR-10 in Figure 6 and CIFAR-100 in Figure 8. The four factors are the same as above. Testing accuracies correspond to the analysis of excess risk, effected by both convergence error and generalization error. Training loss corresponds to the analysis of convergence error, reflecting optimization properties.

## D GENERALIZATION BOUNDS FOR C-PFL AND D-PFL.

In this section, we introduce our proof of the generalization bounds in the main context. We first introduce the general lemmas for both C-PFL and D-PFL. Then we prove the uniform stability to measure the generalization error for them. At the beginning of our proof, we list the important variables used in the study as follows.

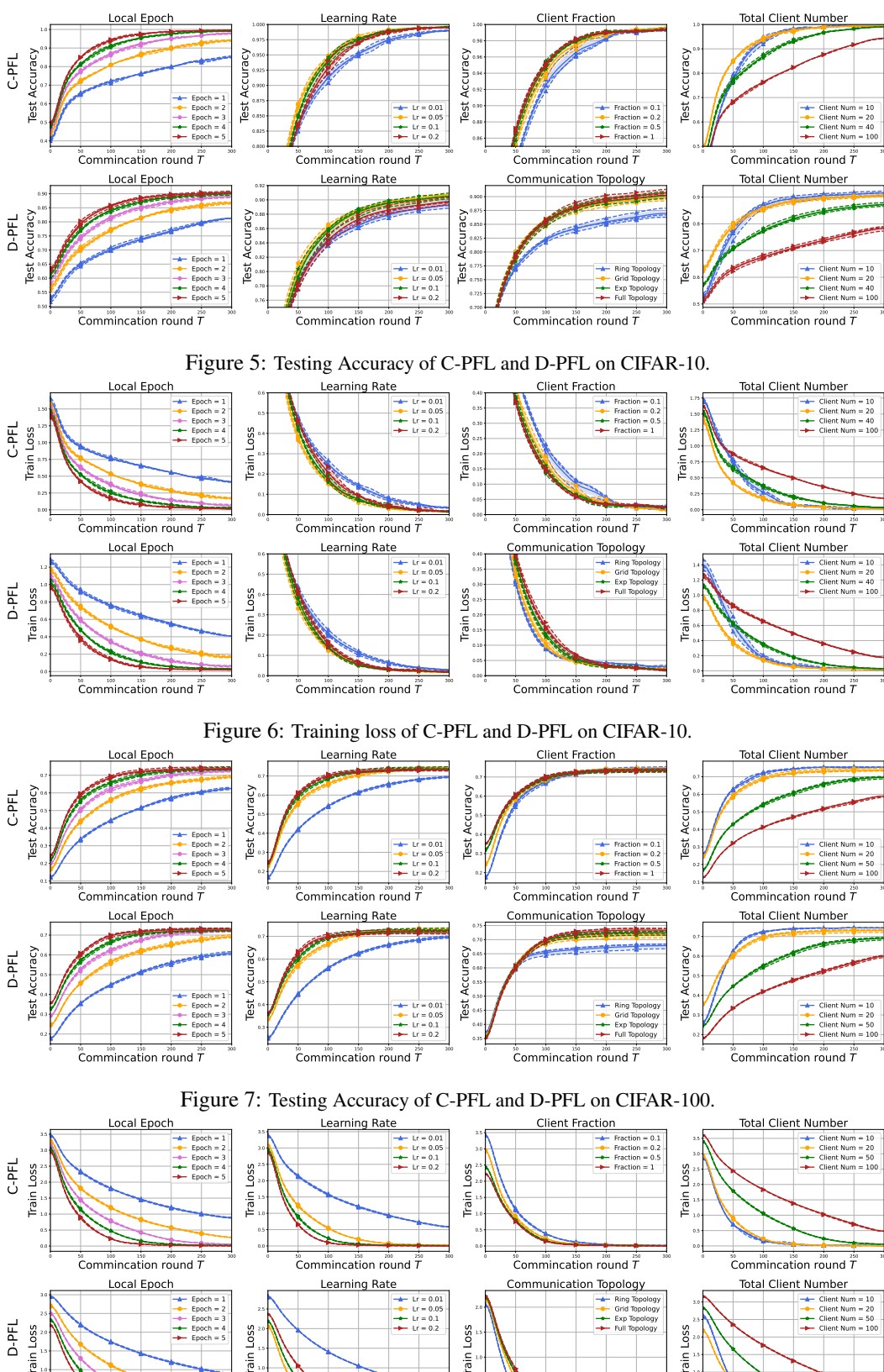

Figure 5: Testing Accuracy of C-PFL and D-PFL on CIFAR-10.

Figure 6: Training loss of C-PFL and D-PFL on CIFAR-10.

Figure 7: Testing Accuracy of C-PFL and D-PFL on CIFAR-100.

Figure 8: Training loss of C-PFL and D-PFL on CIFAR-100.

Table 5: Some abbreviations of the used terms in the proofs.

| Notation | Description |
|---|---|
| $w_{i,k}^t = (u_{i,k}^t, v_{i,k}^t)$ | parameters at $k$-th iteration |
| $w^t = (u^t, V^t)$ | parameters in round $t$ with set $\mathcal{S}$ |
| $\Delta_{u,k}^t = \sum_{i \in [m]} \mathbb{E}\|u^t - \widetilde{u}^t\|$ | stability difference of variables $u$ |
| $\Delta_{v,k}^t = \sum_{i \in [m]} \mathbb{E}\|v_i^t - \widetilde{v}_i^t\|$ | stability difference of variables $v_i$ |
| $F$ | initial function value gap |

## D.1 PRELIMINARY LEMMAS

**Lemma 1 (Mixing Matrix for Decentralized FL**, Lemma 4, Lian et al. (2017)). *For any $t \in \mathbb{Z}^+$, the mixing matrix $\mathbf{W} \in \mathbb{R}^n$ satisfies $\|\mathbf{W}^t - \mathbf{P}\|_{\mathrm{op}} \leq \lambda^t$, where $\lambda := \max\{|\lambda_2|, |\lambda_n(W)|\}$ and for a matrix $\mathbf{A}$, we denote its spectral norm as $\|\mathbf{A}\|_{\mathrm{op}}$. Furthermore, $\mathbf{1} := [1, 1, \dots, 1]^\top \in \mathbb{R}^m$ and*

$$\mathbf{P} := \frac{\mathbf{1}\mathbf{1}^\top}{n} \in \mathbb{R}^{n \times n}.$$

**Lemma 2 (Stability for C-PFL).** *We follow the definition in (Hardt et al., 2016; Zhou et al., 2021) to upper bound the uniform stability term for the shared and personalized variables $u$ and $v_i$ after round $T$ in the central FL paradigm. The updated progress of the shared variables $u$ is like the vanilla FedAvg, where the local updates and server aggregation are conducted alternately. The updated progress of the personalized variables $v_i$ is like the SGD with multiple local updates. Let function $f(w_i)$ satisfies Assumption 4, the models $w_i^T = \mathcal{A}(\mathcal{S})$ and $\widetilde{w}_i^T = \mathcal{A}(\widetilde{\mathcal{S}})$ are generated after $T$ training rounds by the centralized method, we can bound their objective difference as:*

$$
\begin{aligned}
&\mathbb{E}\|f(w_i^T; z) - f(\widetilde{w}_i^T; z)\| \\
&\leq \frac{nU\tau_0}{mS} + G\mathbb{E}\left[\|w_i^T - \widetilde{w}_i^T\| \mid \xi\right] \\
&\leq \frac{nU\tau_0}{mS} + G\mathbb{E}\left[\|u^T - \widetilde{u}^T\| \mid \xi\right] + G\mathbb{E}\left[\|v_i^T - \widetilde{v}_i^T\| \mid \xi\right]
\end{aligned}
\tag{12}
$$

*where $U = \sup_{w_i,z} f(w_i; z) = \sup_{u,v_i,z} f(u, v_i; z) < +\infty$ is the upper bound of the loss and $\tau_0 = t_0 K + k_0$ is a specific index of the total iterations.*

*Proof.* Let $I$ represent the index of the first time to sample the perturbation sample $\widetilde{z}_{i^*,j^*}$ on the dataset $\widetilde{\mathcal{S}}_{i^*}$. When $t_0 K + k_0 < I$, $\Delta_{k_0}^{t_0} = 0$. Then we define

$$P(\xi^c) = P(\Delta_{k_0}^{t_0} > 0) \leq P(I \leq t_0 K + k_0).$$

Expanding the probability we have:

$$
\begin{aligned}
&\mathbb{E}\|f(w_i^T; z) - f(\widetilde{w}_i^T; z)\| \\
&= P(\{\xi\}) \mathbb{E}\left[\|f(w_i^T; z) - f(\widetilde{w}_i^T; z)\| \mid \xi\right] + P(\{\xi^c\}) \mathbb{E}\left[\|f(w_i^T; z) - f(\widetilde{w}_i^T; z)\| \mid \xi^c\right] \\
&\leq \mathbb{E}\left[\|f(w_i^T; z) - f(\widetilde{w}_i^T; z)\| \mid \xi\right] + P(\{\xi^c\}) \sup_{w_i,z} f(w_i; z) \\
&\leq G\mathbb{E}\left[\|w_i^T - \widetilde{w}_i^T\| \mid \xi\right] + UP(\{\xi^c\}) \\
&\leq G\mathbb{E}\left[\|u^T - \widetilde{u}^T\| \mid \xi\right] + G\mathbb{E}\left[\|v_i^T - \widetilde{v}_i^T\| \mid \xi\right] + UP(\{\xi^c\}).
\end{aligned}
$$

Before the $j^*$-th data on $i^*$-th client is sampled, the iterative states are identical on both $\mathcal{S}$ and $\widetilde{\mathcal{S}}$. When the dataset $\widetilde{\mathcal{S}}_{i^*}$ is selected, the perturbation sample $\widetilde{z}_{i^*,j^*}$ can be selected with probability $1/S$. Define $\chi$ as the event sampling dataset $\mathcal{S}_{i^*}$ and the observation moment $\tau_0 = t_0 K + k_0$. Then we

have:

$$P(\{\xi^c\}) \leq P\left(I \leq t_0 K + k_0\right)$$

$$\leq \sum_{t=0}^{t_0-1} \sum_{k=0}^{K-1} P(I = tK + k; \chi) + \sum_{k=0}^{k_0} P\left(I = t_0 K + k; \chi\right)$$

$$= \sum_{t=0}^{t_0-1} \sum_{k=0}^{K-1} \sum_{\chi} P(I = tK + k \mid \chi) P(\chi) + \sum_{k=0}^{k_0} \sum_{\chi} P\left(I = t_0 K + k \mid \chi\right) P(\chi)$$

$$= \frac{n}{m} \left( \sum_{t=0}^{t_0-1} \sum_{k=0}^{K-1} P(I = tK + k) + \sum_{k=0}^{k_0} P\left(I = t_0 K + k\right) \right)$$

$$= \frac{n(t_0 K + k_0)}{mS}$$

$$= \frac{n\tau_0}{mS}.$$

The random active clients with the probability of $n/m$ in the second equality. $\qquad\square$

**Lemma 3** (**Stability for D-PFL**). *We follow the definition in (Hardt et al., 2016; Zhou et al., 2021) to upper bound the uniform stability term for the shared and personalized variables $u$ and $v_i$ after round $T$ in the decentralized FL paradigm. Let function $f(w_i)$ satisfies Assumption 4, the models $w_i^T = \mathcal{A}(\mathcal{S})$ and $\widetilde{w}_i^T = \mathcal{A}(\widetilde{\mathcal{S}})$ are generated after $T$ training rounds by the decentralized method, we can bound their objective difference as:*

$$\mathbb{E}\|f(w_i^T; z) - f(\widetilde{w}_i^T; z)\|$$
$$\leq \frac{U\tau_0}{S} + G\mathbb{E}\left[\|w_i^T - \widetilde{w}_i^T\| \mid \xi\right] \qquad (13)$$
$$\leq \frac{U\tau_0}{S} + G\mathbb{E}\left[\|u_i^T - \widetilde{u}_i^T\| \mid \xi\right] + G\mathbb{E}\left[\|v_i^T - \widetilde{v}_i^T\| \mid \xi\right]$$

*Proof.* For the D-PFL, the most part is the same as the proof for the central algorithms except the probability $P(\chi) = 1$ in a Decentralized Federated Learning setup (because all clients will participate in the training). We bound their objective difference as:

$$\mathbb{E}\left[\|f(w^T; z) - f(\widetilde{w}^T; z)\|\right] \leq G \sum_{i \in [m]} \mathbb{E}\left[\|w_{i,K}^T - \widetilde{w}_{i,K}^T\| \mid \xi\right] + \frac{U\tau_0}{S}. \qquad (14)$$

$\qquad\square$

**Lemma 4** (**Upper Bound of Aggregation Gaps**). *According to Algorithm 2, the aggregation of C-PFL is $u_{i,0}^{t+1} = u^{t+1} = \frac{1}{n} \sum_{i \in S^t} u_{i,K_u}^t$, and the aggregation of D-PFL is $u_{i,0}^{t+1} = \sum_{j \in \mathcal{A}_i} a_{ij} u_{i,K_u}^t$. On both setups, we can upper bound the aggregation gaps by:*

$$\Delta_{u,0}^{t+1} \leq \Delta_{u,K_u}^t,$$
$$\Delta_{v,0}^{t+1} = \Delta_{v,K_v}^t. \qquad (15)$$

*Proof.* For the personal variable $v_i$, they are always kept locally without aggregation, which means that $v_{i,K}^t = v_{i,0}^{t+1}$. So it is obvious to see that $v_{i,K}^t - \widetilde{v}_{i,K}^t = v_{i,0}^{t+1} - \widetilde{v}_{i,0}^{t+1}$, which proves that $\Delta_{v,0}^{t+1} = \Delta_{v,K}^t$. Then we prove the inequation for the shared variables $u$. We discuss it in central and decentralized mode respectively.

(1) C-PFL setup (Acar et al., 2021).

In centralized federated learning, we select a subset $S^t$ in each communication round $t$. Thus we have:

$$
\begin{aligned}
\Delta_{u,0}^{t+1} &= \sum_{i\in[m]} \mathbb{E}\|u_{i,0}^{t+1} - \widetilde{u}_{i,0}^{t+1}\| = \sum_{i\in[m]} \mathbb{E}\|u^{t+1} - \widetilde{u}^{t+1}\| \\
&= \sum_{i\in[m]} \mathbb{E}\|\frac{1}{n}\sum_{i\in S^t}\left(u_i^{t+1} - \widetilde{u}_i^{t+1}\right)\| = \sum_{i\in[m]} \mathbb{E}\|\frac{1}{n}\sum_{i\in S^t}\left(u_{i,K_u}^t - \widetilde{u}_{i,K_u}^t\right)\| \\
&\leq \sum_{i\in[m]} \frac{1}{n}\mathbb{E}\left[\sum_{i\in S^t}\|u_{i,K_u}^t - \widetilde{u}_{i,K_u}^t\|\right] = \sum_{i\in[m]} \frac{1}{n}\frac{n}{m}\sum_{i\in[m]}\mathbb{E}\|u_{i,K_u}^t - \widetilde{u}_{i,K_u}^t\| \\
&= \sum_{i\in[m]} \frac{1}{m}\sum_{i\in[m]}\mathbb{E}\|u_{i,K_u}^t - \widetilde{u}_{i,K_u}^t\| = \sum_{i\in[m]}\mathbb{E}\|u_{i,K_u}^t - \widetilde{u}_{i,K_u}^t\| = \Delta_{u,K_u}^t.
\end{aligned}
$$

(2) D-PFL setup (Sun et al., 2023c).

In decentralized federated learning, we aggregate the models in each neighborhood. Thus we have:

$$
\begin{aligned}
\Delta_{u,0}^{t+1} &= \sum_{i\in[m]}\mathbb{E}\|u_{i,0}^{t+1} - \widetilde{u}_{i,0}^{t+1}\| = \sum_{i\in[m]}\mathbb{E}\|\sum_{j\in W_i}w_{ij}\left(u_{j,K_u}^t - \widetilde{u}_{j,K_u}^t\right)\| \\
&\leq \sum_{i\in[m]}\sum_{j\in W_i}w_{ij}\mathbb{E}\|u_{j,K_u}^t - \widetilde{u}_{j,K_u}^t\| = \sum_{j\in[m]}\sum_{i\in W_i}w_{ji}\mathbb{E}\|u_{j,K_u}^t - \widetilde{u}_{j,K_u}^t\| \\
&\leq \sum_{j\in[m]}\mathbb{E}\|u_{j,K_u}^t - \widetilde{u}_{j,K_u}^t\| = \Delta_{u,K_u}^t.
\end{aligned}
$$

The last equality adopts the symmetry of the adjacent matrix $\mathbf{W} = \mathbf{W}^\top$. $\qquad\square$

**Lemma 5** (**Decentralized Topologies Bounds of** $\lambda$). *For $0 < \lambda < 1$ and $0 < \alpha < 1$, we have the following inequality:*

$$
\sum_{s=0}^{t-1}\frac{\lambda^{t-s-1}}{(s+1)^\alpha} \leq \frac{\kappa_\lambda}{t^\alpha}, \tag{16}
$$

*where* $\kappa_\lambda = \left(\frac{\alpha}{e}\right)^\alpha\frac{1}{\lambda\left(\ln\frac{1}{\lambda}\right)^\alpha} + \frac{2^\alpha}{(1-\alpha)e\lambda\ln\frac{1}{\lambda}} + \frac{2^\alpha}{\lambda\ln\frac{1}{\lambda}}.$

*Proof.* According to the accumulation, we have:

$$
\begin{aligned}
\sum_{s=0}^{t-1}\frac{\lambda^{t-s-1}}{(s+1)^\alpha} &= \lambda^{t-1} + \sum_{s=1}^{t-1}\frac{\lambda^{t-s-1}}{(s+1)^\alpha} \leq \lambda^{t-1} + \int_{s=1}^{s=t}\frac{\lambda^{t-s-1}}{s^\alpha}ds \\
&= \lambda^{t-1} + \int_{s=1}^{s=\frac{t}{2}}\frac{\lambda^{t-s-1}}{s^\alpha}ds + \int_{s=\frac{t}{2}}^{s=t}\frac{\lambda^{t-s-1}}{s^\alpha}ds \\
&\leq \lambda^{t-1} + \lambda^{\frac{t}{2}-1}\int_{s=1}^{s=\frac{t}{2}}\frac{1}{s^\alpha}ds + \left(\frac{2}{t}\right)^\alpha\int_{s=\frac{t}{2}}^{s=t}\lambda^{t-s-1}ds \\
&\leq \lambda^{t-1} + \lambda^{\frac{t}{2}-1}\frac{1}{1-\alpha}\left(\frac{t}{2}\right)^{1-\alpha} + \left(\frac{2}{t}\right)^\alpha\frac{\lambda^{-1}}{\ln\frac{1}{\lambda}}.
\end{aligned}
$$

Thus we have LHS $\leq \frac{1}{t^\alpha}\left(\lambda^{t-1}t^\alpha + \lambda^{\frac{t}{2}-1}\frac{t}{(1-\alpha)2^{1-\alpha}} + \frac{2^\alpha}{\lambda\ln\frac{1}{\lambda}}\right)$. The first term can be bounded as $\lambda^{t-1}t^\alpha \leq \left(\frac{\alpha}{e}\right)^\alpha\frac{1}{\lambda\left(\ln\frac{1}{\lambda}\right)^\alpha}$ and the second term can be bounded as $\lambda^{\frac{t}{2}-1}t \leq \frac{2}{e\lambda\ln\frac{1}{\lambda}}$, which indicates the selection of the constant $\kappa_\lambda = \left(\frac{\alpha}{e}\right)^\alpha\frac{1}{\lambda\left(\ln\frac{1}{\lambda}\right)^\alpha} + \frac{2^\alpha}{(1-\alpha)e\lambda\ln\frac{1}{\lambda}} + \frac{2^\alpha}{\lambda\ln\frac{1}{\lambda}}$. Furthermore, if $0 < \alpha \leq \frac{1}{2} < 1$, we have $\kappa_\lambda \leq \frac{1}{\lambda\left(\ln\frac{1}{\lambda}\right)^\alpha} + \frac{2\sqrt{2}}{e\lambda\ln\frac{1}{\lambda}} + \frac{\sqrt{2}}{\lambda\ln\frac{1}{\lambda}} \leq \max\left\{\frac{1}{\lambda}, \frac{1}{\lambda\sqrt{\ln\frac{1}{\lambda}}}\right\} + \frac{(2+e)\sqrt{2}}{e\lambda\ln\frac{1}{\lambda}} = \mathcal{O}\left(\max\left\{\frac{1}{\lambda}, \frac{1}{\lambda\sqrt{\ln\frac{1}{\lambda}}}\right\} + \frac{1}{\lambda\ln\frac{1}{\lambda}}\right)$ with respect to the constant $\lambda$. $\qquad\square$

### D.2 GENERALIZATION BOUNDS FOR C-PFL

**Lemma 6 (Selecting the Same Sample).** *Under the Assumption 1 and Assumption 4, the gradient for the shared and personalized variables satisfy $g_{u,i,k}^t = \nabla_u F_i(u_{i,k}^t, v_{i,K_v}^t; z)$ and $g_{v,i,k}^t = \nabla_v F_i(u_i^t, v_{i,k}^t; z)$, the local updates satisfy $u_{i,k+1}^t = u_{i,k}^t - \gamma g_{u,i,k}^t$ and $v_{i,k+1}^t = v_{i,k}^t - \gamma g_{v,i,k}^t$. We use $\mathbb{E}[\nabla_u F_i(u_{i,k}^t, v_{i,K_v}^t; z)] = \nabla_u f_i(u_{i,k}^t, v_{i,K_v}^t; z)$ and $\mathbb{E}[\nabla_v F_i(u_i^t, v_{i,k}^t; z)] = \nabla_v f_i(u_i^t, v_{i,k}^t; z)$. If we sample the same data $z$ (not the $z_{i^\star, j^\star}$) in dataset $\mathcal{S}$ and $\widetilde{\mathcal{S}}$ at $k$ iteration on round $t$, we have:*

$$
\begin{aligned}
\mathbb{E}\|u_{i,k+1}^t - \widetilde{u}_{i,k+1}^t\| &\le (1 + \eta_u L_u)\mathbb{E}\|u_{i,k}^t - \widetilde{u}_{i,k}^t\| + \eta_u L_{uv}\mathbb{E}\|v_{i,K_v}^t - \widetilde{v}_{i,K_v}^t\|, \\
\mathbb{E}\|v_{i,k+1}^t - \widetilde{v}_{i,k+1}^t\| &\le (1 + \eta_v L_v)\mathbb{E}\|v_{i,k}^t - \widetilde{v}_{i,k}^t\| + \eta_v L_{vu}\mathbb{E}\|u_i^t - \widetilde{u}_i^t\|.
\end{aligned}
\tag{17}
$$

*Proof.* We first conduct the local update for the personalized variables. The update progress in each round $t$ is as follows:

$$
\begin{aligned}
\mathbb{E}\|v_{i,k+1}^t - \widetilde{v}_{i,k+1}^t\| &= \mathbb{E}\|v_{i,k}^t - \widetilde{v}_{i,k}^t - \eta_v(g_{v,i,k}^t - \widetilde{g}_{v,i,k}^t)\| \\
&\le \mathbb{E}\|v_{i,k}^t - \widetilde{v}_{i,k}^t\| + \eta_v \mathbb{E}\|\nabla_v f_i(u_i^t, v_{i,k}^t; z) - \nabla_v f_i(\widetilde{u}_i^t, \widetilde{v}_{i,k}^t; z)\| \\
&\le (1 + \eta_v L_v)\mathbb{E}\|v_{i,k}^t - \widetilde{v}_{i,k}^t\| + \eta_v L_{vu}\mathbb{E}\|u_i^t - \widetilde{u}_i^t\|.
\end{aligned}
$$

The alternative update progress for the shared variables is based on the updated $v_i^{t+1} = v_{i,K_v}^{t+1}$:

$$
\begin{aligned}
&\mathbb{E}\|u_{i,k+1}^t - \widetilde{u}_{i,k+1}^t\| \\
&= \mathbb{E}\|u_{i,k}^t - \widetilde{u}_{i,k}^t - \eta_u(g_{u,i,k}^t - \widetilde{g}_{u,i,k}^t)\| \\
&\le \mathbb{E}\|u_{i,k}^t - \widetilde{u}_{i,k}^t\| + \eta_u \mathbb{E}\|\nabla_u f_i(u_{i,k}^t, v_{i,K_v}^t; z) - \nabla_u f_i(\widetilde{u}_{i,k}^t, \widetilde{v}_{i,K}^t; z)\| \\
&\le (1 + \eta_u L_u)\mathbb{E}\|u_{i,k}^t - \widetilde{u}_{i,k}^t\| + \eta_u L_{uv}\mathbb{E}\|v_{i,K_v}^t - \widetilde{v}_{i,K_v}^t\|.
\end{aligned}
$$

$\square$

**Lemma 7 (Selecting the Different Sample).** *Assume $g_{u,i,k}^t = \nabla_u F_i(u_{i,k}^t, v_{i,K_v}^t; z)$ and $\widetilde{g}_{v,i,k}^t = \nabla_v F_i(\widetilde{u}_i^t, \widetilde{v}_{i,k}^t; \widetilde{z})$, the local updates satisfy $u_{i,k+1}^t = u_{i,k}^t - \gamma g_{u,i,k}^t$ and $v_{i,k+1}^t = v_{i,k}^t - \gamma g_{v,i,k}^t$. If we sample the different data samples $z_{i^\star, j^\star}$ and $\widetilde{z}_{i^\star, j^\star}$ (simplified to $z$ and $\widetilde{z}$), we have:*

$$
\begin{aligned}
\mathbb{E}\|u_{i,k+1}^t - \widetilde{u}_{i,k+1}^t\| &\le (1 + \eta_u L_u)\mathbb{E}\|u_{i,k}^t - \widetilde{u}_{i,k}^t\| + \eta_u L_{uv}\mathbb{E}\|v_{i,K_v}^t - \widetilde{v}_{i,K_v}^t\| + 2\eta_u(\sigma_u + \delta_u), \\
\mathbb{E}\|v_{i,k+1}^t - \widetilde{v}_{i,k+1}^t\| &\le (1 + \eta_v L_v)\mathbb{E}\|v_{i,k}^t - \widetilde{v}_{i,k}^t\| + \eta_v L_{vu}\mathbb{E}\|u_i^t - \widetilde{u}_i^t\| + 2\eta_v \sigma_v.
\end{aligned}
\tag{18}
$$

*Proof.* We first conduct the local update for the personalized variables. The update progress in each round $t$ is as follows:

$$
\begin{aligned}
&\mathbb{E}\|v_{i^\star,k+1}^t - \widetilde{v}_{i^\star,k+1}^t\| \\
&= \mathbb{E}\|v_{i^\star,k}^t - \widetilde{v}_{i^\star,k}^t - \eta_v(g_{v,i^\star,k}^t - \widetilde{g}_{v,i^\star,k}^t)\| \\
&\le \mathbb{E}\|v_{i^\star,k}^t - \widetilde{v}_{i^\star,k}^t\| + \eta_v \mathbb{E}\|\nabla_v f_{i^\star}(u_{i^\star}^t, v_{i^\star,k}^t, z) - \nabla_v f_{i^\star}(\widetilde{u}_{i^\star}^t, \widetilde{v}_{i^\star,k}^t \widetilde{z})\| \\
&\le \mathbb{E}\|v_{i^\star,k}^t - \widetilde{v}_{i^\star,k}^t\| + \eta_v \mathbb{E}\|\nabla_v f_{i^\star}(u_{i^\star}^t, v_{i^\star,k}^t, z) - \nabla_v f_{i^\star}(\widetilde{u}_{i^\star}^t, \widetilde{v}_{i^\star,k}^t, z)\| \\
&\quad + \eta_v \mathbb{E}\|\nabla_v f_{i^\star}(\widetilde{u}_{i^\star}^t, \widetilde{v}_{i^\star,k}^t, z) - \nabla_v f_{i^\star}(\widetilde{u}_{i^\star}^t, \widetilde{v}_{i^\star,k}^t, \widetilde{z})\| \\
&\le (1 + \eta_v L_v)\mathbb{E}\|v_{i^\star,k}^t - \widetilde{v}_{i^\star,k}^t\| + \eta_v L_{vu}\mathbb{E}\|u_i^t - \widetilde{u}_i^t\| \\
&\quad + \eta_v \mathbb{E}\|\nabla_v F_{i^\star}(\widetilde{u}_{i^\star}^t, \widetilde{v}_{i^\star,k}^t, z) - \nabla_v f_{i^\star}(\widetilde{u}_{i^\star}^t, \widetilde{v}_{i^\star,k}^t) - \nabla_v f_{i^\star}(\widetilde{u}_{i^\star}^t, \widetilde{v}_{i^\star,k}^t, \widetilde{z}) + \nabla_v f_{i^\star}(\widetilde{u}_{i^\star}^t, \widetilde{v}_{i^\star,k}^t)\| \\
&\le (1 + \eta_v L_v)\mathbb{E}\|v_{i,k}^t - \widetilde{v}_{i,k}^t\| + \eta_v L_{vu}\mathbb{E}\|u_i^t - \widetilde{u}_i^t\| + 2\eta_v \sigma_v.
\end{aligned}
$$

The last inequality adopts $\mathbb{E}[x] = \sqrt{(\mathbb{E}[x])^2} = \sqrt{\mathbb{E}[x^2] - \mathbb{E}[x - \mathbb{E}[x]]^2} \le \sqrt{\mathbb{E}[x^2]}$.

The alternative update progress for the shared variables is based on the updated $v_i^{t+1} = v_{i,K_v}^{t+1}$:

$$
\begin{aligned}
&\mathbb{E}\|u_{i^\star,k+1}^t - \widetilde{u}_{i^\star,k+1}^t\| \\
&= \mathbb{E}\|u_{i^\star,k}^t - \widetilde{u}_{i^\star,k}^t - \eta_u(g_{u,i^\star,k}^t - \widetilde{g}_{u,i^\star,k}^t)\| \\
&\leq \mathbb{E}\|u_{i^\star,k}^t - \widetilde{u}_{i^\star,k}^t\| + \eta_u\mathbb{E}\|\nabla_u f_{i^\star}(u_{i^\star,k}^t, v_{i^\star,K_v}^t, z) - \nabla_u f_{i^\star}(\widetilde{u}_{i^\star,k}^t, \widetilde{v}_{i^\star,K_v}^t, \widetilde{z})\| \\
&\leq \mathbb{E}\|u_{i^\star,k}^t - \widetilde{u}_{i^\star,k}^t\| + \eta_u\mathbb{E}\|\nabla_u f_{i^\star}(u_{i^\star,k}^t, v_{i^\star,K_v}^t, z) - \nabla_u f_{i^\star}(\widetilde{u}_{i^\star,k}^t, \widetilde{v}_{i^\star,K_v}^t, z)\| \\
&\quad + \eta_u\mathbb{E}\|\nabla_u f_{i^\star}(\widetilde{u}_{i^\star,k}^t, \widetilde{v}_{i^\star,K_v}^t, z) - \nabla_u f_{i^\star}(\widetilde{u}_{i^\star,k}^t, \widetilde{v}_{i^\star,K_v}^t, \widetilde{z})\| \\
&\leq (1 + \eta_u L_u)\mathbb{E}\|u_{i^\star,k}^t - \widetilde{u}_{i^\star,k}^t\| + \eta_u L_{uv}\mathbb{E}\|v_{i,k}^t - \widetilde{v}_{i,K_v}^t\| \\
&\quad + \eta_u\mathbb{E}\|\nabla_u f_{i^\star}(\widetilde{u}_{i^\star,k}^t, \widetilde{v}_{i^\star,K_v}^t, z) - \nabla_u f_{i^\star}(\widetilde{u}_{i^\star,k}^t, \widetilde{v}_{i^\star,K_v}^t) - \nabla_u f_{i^\star}(\widetilde{u}_{i^\star,K_v}^t, \widetilde{v}_{i^\star,K_v}^t, \widetilde{z}) + \nabla_u f_{i^\star}(\widetilde{u}_{i^\star,k}^t, \widetilde{v}_{i^\star,K_v}^t)\| \\
&\leq (1 + \eta_u L_u)\mathbb{E}\|u_{i,k}^t - \widetilde{u}_{i,k}^t\| + \eta_u L_{uv}\mathbb{E}\|v_{i,K_v}^t - \widetilde{v}_{i,K_v}^t\| + 2\eta_u(\sigma_u + \delta_u).
\end{aligned}
$$

$\square$

**Lemma 8** (**Recursion in local update**). *Since $\Delta_k^t = \Delta_{u,k}^t + \Delta_{v,k}^t$, according to the Lemma 6 and 7, we can bound the recursion in the local training:*

$$
\Delta_{v,k+1}^t \leq (1 + \eta_v L_v)\left(\Delta_{v,k}^t + \frac{2\sigma_v}{SL_v} + \frac{L_{vu}\Delta_{u,0}^t}{L_v}\right).
$$

$$
\Delta_{u,k+1}^t \leq (1 + \eta_u L_u)\left(\Delta_{u,k}^t + \frac{2(\sigma_u + \delta_u)}{SL_u} + \frac{L_{uv}\Delta_{v,K_v}^t}{L_u}\right).
$$

*Proof.* In each iteration, the specific $j^\star$-th data sample in the $\mathcal{S}_{i^\star}$ and $\widetilde{\mathcal{S}}_{i^\star}$ is uniformly selected with the probability of $1/S$. In other datasets $\mathcal{S}_i$, all the data samples are the same. Thus we have the recursion for the personalized variables:

$$
\begin{aligned}
\Delta_{v,k+1}^t &= \sum_{i \neq i^\star} \mathbb{E}\left[\|v_{i,k+1}^t - \widetilde{v}_{i,k+1}^t\|\right] + \mathbb{E}\left[\|v_{i^\star,k+1}^t - \widetilde{v}_{i^\star,k+1}^t\|\right] \\
&\leq (1 + \eta_v L_v)\sum_{i \neq i^\star}\mathbb{E}\|v_{i,k}^t - \widetilde{v}_{i,k}^t\| + \eta_v L_{vu}\sum_{i \neq i^\star}\mathbb{E}\|u_i^t - \widetilde{u}_i^t\| \\
&\quad + \left(1 - \frac{1}{S}\right)\left[(1 + \eta_v L_v)\mathbb{E}\|v_{i,k}^t - \widetilde{v}_{i,k}^t\| + \eta_v L_{vu}\mathbb{E}\|u_i^t - \widetilde{u}_i^t\|\right] \\
&\quad + \frac{1}{S}\left[(1 + \eta_v L_v)\mathbb{E}\|v_{i,k}^t - \widetilde{v}_{i,k}^t\| + \eta_v L_{vu}\mathbb{E}\|u_i^t - \widetilde{u}_i^t\| + 2\eta_v\sigma_v\right] \\
&= (1 + \eta_v L_v)\Delta_{v,k}^t + \eta_v L_{vu}\Delta_{u,0}^t + \frac{2\eta_v\sigma_v}{S}.
\end{aligned}
$$

Similarly, for the shared variables, we have the progress in each round $t$:

$$
\begin{aligned}
\Delta_{u,k+1}^t &= \sum_{i \neq i^\star} \mathbb{E}\left[\|u_{i,k+1}^t - \widetilde{u}_{i,k+1}^t\|\right] + \mathbb{E}\left[\|u_{i^\star,k+1}^t - \widetilde{u}_{i^\star,k+1}^t\|\right] \\
&\leq (1 + \eta_u L_u)\sum_{i \neq i^\star}\mathbb{E}\|u_{i,k}^t - \widetilde{u}_{i,k}^t\| + \eta_u L_{uv}\sum_{i \neq i^\star}\mathbb{E}\|v_{i,K_v}^t - \widetilde{v}_{i,K_v}^t\| \\
&\quad + \left(1 - \frac{1}{S}\right)\left[(1 + \eta_u L_u)\mathbb{E}\|u_{i,k}^t - \widetilde{u}_{i,k}^t\| + \eta_u L_{uv}\mathbb{E}\|v_{i,K_v}^t - \widetilde{v}_{i,K_v}^t\|\right] \\
&\quad + \frac{1}{S}\left[(1 + \eta_u L_u)\mathbb{E}\|u_{i,k}^t - \widetilde{u}_{i,k}^t\| + \eta_u L_{uv}\mathbb{E}\|v_{i,K_v}^t - \widetilde{v}_{i,K_v}^t\| + 2\eta_u(\sigma_u + \delta_u)\right] \\
&= (1 + \eta_u L_u)\Delta_{u,k}^t + \eta_u L_{uv}\Delta_{v,K_v}^t + \frac{2\eta_u(\sigma_u + \delta_u)}{S}.
\end{aligned}
$$

Then we can bound the recursion formulation as:

$$\Delta_{v,k+1}^t + \frac{2\sigma_v}{SL_v} + \frac{L_{vu}\Delta_u^t}{L_v} \le (1 + \eta_v L_v)(\Delta_{v,k}^t + \frac{2\sigma_v}{SL_v} + \frac{L_{vu}\Delta_{u,0}^t}{L_v}),$$

$$\Delta_{u,k+1}^t + \frac{2(\sigma_u + \delta_u)}{SL_u} + \frac{L_{uv}\Delta_{v,K_v}^t}{L_u} \le (1 + \eta_u L_u)(\Delta_{u,k}^t + \frac{2(\sigma_u + \delta_u)}{SL_u} + \frac{L_{uv}\Delta_{v,K_v}^t}{L_u}).$$

Zoom out the variables on the left-hand side, then we finish the proof.

$\square$

**Main Proof for Theorem 1** According to the Lemma 4 and 8, it is easy to bound the local stability term. We still obverse it when the event $\xi$ happens, and we have $\Delta_{k_0}^{t_0} = 0$. Therefore, we unwind the recurrence formulation from $T, K$ to $t_0, k_0$. Let $\eta_u = \frac{\mu_u}{\tau} = \frac{\mu_u}{tK+k}$ and $\eta_v = \frac{\mu_v}{\tau} = \frac{\mu_v}{tK+k}$ are decayed as the communication round $t$ and iteration $k$ where $\mu_u \le \frac{1}{L_u}$ and $\mu_v \le \frac{1}{L_v}$ are specific constants, we have:

$$
\begin{aligned}
\Delta_{v,K_v}^T &\le \left[ \prod_{\tau=(T-1)K_v+1}^{TK_v} \left(1 + \frac{\mu_v L_v}{\tau}\right) \right] \left( \Delta_{v,0}^T + \frac{2\sigma_v}{SL_v} + \frac{L_{vu}\Delta_{u,0}^T}{L_v} \right) \\
&\le \left[ \prod_{\tau=(T-1)K_v+1}^{TK_v} \left(1 + \frac{\mu_v L_v}{\tau}\right) \right] \left( \Delta_{v,K_v}^{T-1} + \frac{2\sigma_v}{SL_v} + \frac{L_{vu}\Delta_{u,K_v}^{T-1}}{L_v} \right) \\
&\le \left[ \prod_{\tau=t_0K+k_0+1}^{TK_v} \left(1 + \frac{\mu_v L_v}{\tau}\right) \right] \left( \Delta_{v,k_0}^{t_0} + \frac{2\sigma_v}{SL_v} + \frac{L_{vu}\Delta_{u,k_0}^{t_0}}{L_v} \right) \\
&\le \left[ \prod_{\tau=t_0K+k_0+1}^{TK_v} e^{\left(\frac{\mu_v L_v}{\tau}\right)} \right] \left( \frac{2\sigma_v}{SL_v} \right) \\
&= e^{\mu_v L_v \left( \sum_{\tau=t_0K+k_0+1}^{TK_v} \frac{1}{\tau} \right)} \frac{2\sigma_v}{SL_v} \\
&\le e^{\mu_v L_v \ln\left( \frac{TK_v}{t_0K+k_0} \right)} \frac{2\sigma_v}{SL_v} \\
&\le \left( \frac{TK_v}{\tau_0} \right)^{\mu_v L_v} \frac{2\sigma_v}{SL_v}.
\end{aligned}
\tag{19}
$$

Similarly, for the shared variables, we have the progress in round $T$:

$$
\begin{aligned}
\Delta_{u,K_u}^T &\le \left[ \prod_{\tau=(T-1)K_u+1}^{TK_u} \left(1 + \frac{\mu_u L_u}{\tau}\right) \right] \left( \Delta_{u,0}^T + \frac{2(\sigma_u + \delta_u)}{SL_u} + \frac{L_{uv}\Delta_{v,K_v}^T}{L_u} \right) \\
&\le \left[ \prod_{\tau=(T-1)K_u+1}^{TK_u} \left(1 + \frac{\mu_u L_u}{\tau}\right) \right] \left( \Delta_{u,K_u}^{T-1} + \frac{2(\sigma_u + \delta_u)}{SL_u} \right) + \left[ \prod_{\tau=(T-1)K_u+1}^{TK_u} \left(1 + \frac{\mu_u L_u}{\tau}\right) \right] \frac{L_{uv}\Delta_{v,K_v}^T}{L_u} \\
&\le \left[ \prod_{\tau=t_0K+k_0+1}^{TK_u} \left(1 + \frac{\mu_u L_u}{\tau}\right) \right] \left( \Delta_{k_0}^{t_0} + \frac{2(\sigma_u + \delta_u)}{SL_u} \right) + \left[ \prod_{\tau=t_0K+k_0+1}^{TK_u} \left(1 + \frac{\mu_u L_u}{\tau}\right) \right] \frac{L_{uv}\Delta_{v,K_v}^T}{L_u} \\
&\le \left[ \prod_{\tau=t_0K+k_0+1}^{TK_u} e^{\left(\frac{\mu_u}{\tau}\right)} \right] \left( \frac{2(\sigma_u + \delta_u)}{SL_u} + \frac{L_{uv}\Delta_{v,K_v}^T}{L_u} \right)
\end{aligned}
$$

Expand the first item, then we have:

$$\Delta_{u,K_u}^T \le e^{\mu_u L_u \left(\sum_{\tau=t_0 K+k_0+1}^{TK_u} \frac{1}{\tau}\right)} \left(\frac{2(\sigma_u + \delta_u)}{SL_u} + \frac{L_{uv}\Delta_{v,K_v}^T}{L_u}\right)$$

$$\le e^{\mu_u L_u \ln\left(\frac{TK_u}{t_0 K+k_0}\right)} \left(\frac{2(\sigma_u + \delta_u)}{SL_u} + \frac{L_{uv}\Delta_{v,K_v}^T}{L_u}\right)$$

$$\le \left(\frac{TK_u}{\tau_0}\right)^{\mu_u L_u} \left(\frac{2(\sigma_u + \delta_u)}{SL_u} + \frac{L_{uv}}{L_u}\left(\frac{TK_v}{\tau_0}\right)^{\mu_v L_v} \frac{2\sigma_v}{SL_v}\right)$$

$$\le \left(\frac{TK_u}{\tau_0}\right)^{\mu_u L_u} \frac{2(\sigma_u + \delta_u)}{SL_u} + \left(\frac{TK_u}{\tau_0}\right)^{\mu_u L_u}\left(\frac{TK_v}{\tau_0}\right)^{\mu_v L_v} \frac{2L_{uv}\sigma_v}{SL_v L_u}.$$

We can see that the bound of the local stability term for the shared variables in C-PFL has an extra term $\left(\frac{TK_u}{\tau_0}\right)^{\mu_u}\left(\frac{TK_v}{\tau_0}\right)^{\mu_v} \frac{2L_{uv}\sigma_v}{SL_v L_u}$. This is the alignment error caused by the alternative update for the personalized and shared variables, which is related to the smoothness of $L_u, L_v, L_{uv}$, the local epochs $K_u, K_v$ and the variance bound $\sigma_v$.

Therefore, we get the combination of $\Delta_{u,K}^T$ and $\Delta_{v,K}^T$ as $\Delta_K^T$:

$$\Delta_K^T = \Delta_{u,K_v}^T + \Delta_{v,K_u}^T \le \left(\frac{TK_u}{\tau_0}\right)^{\mu_u L_u} \frac{2(\sigma_u + \delta_u)}{SL_u} + \left(\frac{TK_v}{\tau_0}\right)^{\mu_v L_v}\left(1 + \frac{L_{uv}}{L_u}\left(\frac{TK_u}{\tau_0}\right)^{\mu_u L_u}\right)\frac{2\sigma_v}{SL_v}.$$

According to the Lemma 2, the first term in the stability (condition is omitted for abbreviation) can be bound as:

$$\mathbb{E}\|w_i^{T+1} - \widetilde{w}_i^{T+1}\|$$
$$= \mathbb{E}\|\frac{1}{n}\sum_{i\in S^t}\left(w_{i,K}^T - \widetilde{w}_{i,K}^T\right)\| = \frac{1}{n}\mathbb{E}\|\sum_{i\in S^t}\left(w_{i,K}^T - \widetilde{w}_{i,K}^T\right)\|$$
$$\le \frac{1}{n}\mathbb{E}\sum_{i\in S^t}\|\left(w_{i,K}^T - \widetilde{w}_{i,K}^T\right)\| = \frac{1}{n}\frac{n}{m}\mathbb{E}\sum_{i\in[m]}\|\left(w_{i,K}^T - \widetilde{w}_{i,K}^T\right)\|$$
$$= \frac{1}{m}\sum_{i\in[m]}\mathbb{E}\|\left(w_{i,K}^T - \widetilde{w}_{i,K}^T\right)\| = \frac{1}{m}\Delta_K^T$$
$$\le \left(\frac{TK_u}{\tau_0}\right)^{\mu_u L_u} \frac{2(\sigma_u + \delta_u)}{mSL_u} + \left(\frac{TK_v}{\tau_0}\right)^{\mu_v L_v}\left(1 + \frac{L_{uv}}{L_u}\left(\frac{TK_u}{\tau_0}\right)^{\mu_u L_u}\right)\frac{2\sigma_v}{mSL_v}.$$

Therefore, we can upper bound the stability in C-PFL as:

$$\mathbb{E}\|f(w_i^{T+1}; z) - f(\widetilde{w}_i^{T+1}; z)\|$$
$$\le G\mathbb{E}\|w_i^{T+1} - \widetilde{w}_i^{T+1}\| + \frac{nU\tau_0}{mS}$$
$$\le \left(\frac{TK_u}{\tau_0}\right)^{\mu_u L_u} \frac{2G(\sigma_u + \delta_u)}{mSL_u} + \frac{nU\tau_0}{mS} + \left(\frac{TK_v}{\tau_0}\right)^{\mu_v L_v}\left(1 + \frac{L_{uv}}{L_u}\left(\frac{TK_u}{\tau_0}\right)^{\mu_u L_u}\right)\frac{2G\sigma_v}{mSL_v}.$$

Obviously, we can select a proper event $\xi$ with a proper $\tau_0$ to minimize the upper bound. For $\tau \in [1, TK]$, by selecting $\tau_0 = \left[\frac{2G((\sigma_u+\delta_u)L_v+\sigma_v L_u)}{nUL_u L_v}\right]^{\frac{1}{1+\mu L}} (TK)^{\frac{\mu L}{1+\mu L}}$, we can minimize the bound as:

$$\mathbb{E}\left[\|f(u^T, V^T; z_j) - f(\widetilde{u}^T, \widetilde{V}^T; z_j)\|\right] \le \frac{4}{mS}\left[\frac{G((\sigma_u + \delta_u)L_v + \sigma_v L_u)}{L_u L_v}\right]^{\frac{1}{1+\mu L}} (nUTK)^{\frac{\mu L}{1+\mu L}}.$$

### D.3 GENERALIZATION BOUNDS FOR D-PFL

**Lemma 9** (Bounded the local gradients). *When $(t, k) < (t_0, k_0)$, the sampled data is always the same between the different datasets, which shows $\Gamma_k^t = 0$. When $t = t_0$, only those updates at $k \geq k_0$ are different. When $t > t_0$, all the local gradients difference during local $K$ iterations are non-zero. Thus we can first explore the upper bound of the stages with full $K$ iterations when $t > t_0$. Let the data sample $z$ be the same random data sample and $z/\widetilde{z}$ be a different sample pair for abbreviation, when $t \geq t_0$, we have: If we sample the same data $z$ (not the $z_{i^\star, j^\star}$) in dataset $\mathcal{C}$ and $\widetilde{\mathcal{C}}$ at $k$ iteration on round $t$, we have:*

$$\mathbb{E}\|\eta_u \Gamma_{u,k}^t\| \leq \left(\frac{\tau}{\tau_0}\right)^{\mu_u L_u} \frac{2\mu_u(\sigma_u + \delta_u)}{\tau S}. \tag{20}$$

*Proof.* According to the Lemma 4 and 8, we can also bound the local stability term for the personal variables. Let the learning rate $\eta_v = \frac{\mu_v}{\tau} = \frac{\mu_v}{tK_v + k}$ is decayed as the communication round $t$ and iteration $k$ where $\mu_v$ is a specific constant, we have:

$$\Delta_{v,k}^t + \frac{2\sigma_v}{SL_v} \leq \left(\frac{\tau}{\tau_0}\right)^{\mu_v L_v} \frac{2\sigma_v}{SL_v}. \tag{21}$$

For shared variables, let $\Gamma_{u,k}^t = \left[g_{u,0,k}^t - \widetilde{g}_{u,0,k}^t, g_{u,1,k}^t - \widetilde{g}_{u,1,k}^t, \cdots, g_{u,m,k}^t - \widetilde{g}_{u,m,k}^t\right]^\top$, we have:

$$\mathbb{E}\|\eta_u \Gamma_{u,k}^t\| = \mathbb{E}\|\eta_u \left[g_{u,0,k}^t - \widetilde{g}_{u,0,k}^t, g_{u,1,k}^t - \widetilde{g}_{u,1,k}^t, \cdots, g_{u,m,k}^t - \widetilde{g}_{u,m,k}^t\right]^\top\|$$

$$\leq \eta_u \sum_{i \in [m]} \mathbb{E}\|g_{u,i,k}^t - \widetilde{g}_{u,i,k}^t\|$$

$$\leq \eta_u \sum_{i \neq i^\star} \mathbb{E}\|\nabla_u f_i(u_{i,k}^t, v_{i,K_v}^t, z) - \nabla_u f_i(\widetilde{u}_{i,k}^t, \widetilde{v}_{i,K_v}^t, z)\|$$

$$+ \frac{(S-1)\eta_u}{S} \mathbb{E}\|\nabla_u f_{i^\star}(u_{i^\star,k}^t, v_{i^\star,K_v}^t, z) - \nabla_u f_{i^\star}(\widetilde{u}_{i^\star,k}^t, \widetilde{v}_{i^\star,K_v}^t, z)\|$$

$$+ \frac{\eta_u}{S} \mathbb{E}\|\nabla_u f_{i^\star}(u_{i^\star,k}^t, v_{i^\star,K_v}^t, z) - \nabla_u f_{i^\star}(\widetilde{u}_{i^\star,k}^t, \widetilde{v}_{i^\star,K_v}^t, z) + \nabla_u f_{i^\star}(\widetilde{u}_{i^\star,k}^t, \widetilde{v}_{i^\star,K_v}^t, z) - \nabla_u f_{i^\star}(\widetilde{u}_{i^\star,k}^t, \widetilde{v}_{i^\star,K_v}^t, \widetilde{z})\|$$

$$\leq \eta_u \sum_{i \neq i^\star} \left(L_u \mathbb{E}\|u_{i,k}^t - \widetilde{u}_{i,k}^t\| + L_{uv} \mathbb{E}\|v_{i,K_v}^t - \widetilde{v}_{i,K_v}^t\|\right)$$

$$+ \frac{(S-1)\eta_u}{S}\left(L_u \mathbb{E}\|u_{i,k}^t - \widetilde{u}_{i,k}^t\| + L_{uv} \mathbb{E}\|v_{i,K_v}^t - \widetilde{v}_{i,K_v}^t\|\right)$$

$$+ \frac{\eta_u}{S}\left(L_u \mathbb{E}\|u_{i,k}^t - \widetilde{u}_{i,k}^t\| + L_{uv} \mathbb{E}\|v_{i,K_v}^t - \widetilde{v}_{i,K_v}^t\|\right)$$

$$+ \frac{\eta_u}{S} \mathbb{E}\|(\nabla f_{i^\star}(\widetilde{u}_{i^\star,k}^t, \widetilde{v}_{i^\star,K_v}^t, z) - \nabla_u f_{i^\star}(\widetilde{u}_{i^\star,k}^t, \widetilde{v}_{i^\star,K_v}^t)) - (\nabla_u f_{i^\star}(\widetilde{u}_{i^\star,k}^t, \widetilde{v}_{i^\star,K_v}^t, \widetilde{z}) - \nabla_u f_{i^\star}(\widetilde{u}_{i^\star,k}^t, \widetilde{v}_{i^\star,K_v}^t))\|$$

$$\leq \eta_u \sum_{i \in [m]} \left(L_u \mathbb{E}\|u_{i,k}^t - \widetilde{u}_{i,k}^t\| + L_{uv} \mathbb{E}\|v_{i,K}^t - \widetilde{v}_{i,K}^t\|\right) + \frac{2\eta_u \sigma_u}{S}$$

$$= \eta_u L_u(\Delta_{u,k}^t + \frac{L_{uv}\Delta_{v,K}^t}{L_u} + \frac{2\sigma_u}{SL_u}).$$

According to the Lemma 4, 8 and Eq.(21), we bound the gradient difference as:

$$\mathbb{E}\|\eta_u \Gamma_{u,k}^t\| \leq \eta_u L_u \left(\Delta_{u,k}^t + \frac{L_{uv}\Delta_{v,K}^t}{L_u} + \frac{2\sigma_u}{SL_u}\right)$$

$$\leq \left(\frac{\tau}{\tau_0}\right)^{\mu_u L_u} \frac{2\mu_u(\sigma_u + \delta_u)}{\tau S} + \left(\frac{L_{uv}}{L_u}\right)\left(\frac{\tau}{\tau_0}\right)^{\mu_v L_v} \frac{2\mu_u \sigma_v}{\tau S}.$$

where $\tau = tK + k$.

$\square$

**Lemma 10** (Bounded the local gradients). *When $(t, k) < (t_0, k_0)$, the sampled data is always the same between the different datasets, which shows $\Gamma_k^t = 0$. When $t = t_0$, only those updates at $k \geq k_0$ are different. When $t > t_0$, all the local gradients difference during local $K_u$ iterations are non-zero. Thus we can first explore the upper bound of the stages with full $K_u$ iterations when $t > t_0$. Let the data sample $z$ be the same random data sample and $z/\widetilde{z}$ be a different sample pair for abbreviation and $\Phi_{u,k}^t = \left[ u_{0,k}^t - \widetilde{u}_{0,k}^t, u_{1,k}^t - \widetilde{u}_{1,k}^t, \cdots, u_{m,k}^t - \widetilde{u}_{m,k}^t \right]^\top$. When $t \geq t_0$, we have:*

$$\mathbb{E} \| (\mathbf{I} - \mathbf{P}) \Phi_{u,K_u}^t \| \leq \frac{4\mu_u(\sigma_u + \delta_u)\kappa_\lambda}{S} \left( \frac{K_u}{\tau_0} \right)^{\mu_u L_u} \frac{1}{t^{1-\mu_u L_u}} + (\frac{L_{uv}}{L_u})\frac{4\mu_u \sigma_v \kappa_\lambda}{S} \left( \frac{K_u}{\tau_0} \right)^{\mu_v L_v} \frac{1}{t^{1-\mu_v L_v}},$$
$$(22)$$

$$\mathbb{E} \| (\mathbf{W} - \mathbf{P}) \Phi_{u,K_u}^t \| \leq \frac{2\mu_u(\sigma_u + \delta_u)\lambda\kappa_\lambda}{S} \left( \frac{K_u}{\tau_0} \right)^{\mu_u L_u} \frac{1}{t^{1-\mu_u L_u}} + (\frac{L_{uv}}{L_u})\frac{2\mu_u \sigma_v \kappa_\lambda}{S} \left( \frac{K_u}{\tau_0} \right)^{\mu_v L_v} \frac{1}{t^{1-\mu_v L_v}}.$$
$$(23)$$

*Proof.* In the decentralized method, the aggregation performs after $K$ local updates which demonstrates that the initial state of each round is $\mathbf{U}_0^t = \mathbf{W}\mathbf{U}_{K_u}^{t-1}$. It also works on their difference $\Phi_{u,0}^t = \mathbf{W}\Phi_{u,K_u}^{t-1}$. Therefore, we have:

$$\Phi_{u,K_u}^t = \Phi_{u,0}^t - \sum_{k=0}^{K_u-1} \eta_u \Gamma_{u,k}^t = \mathbf{W}\Phi_{u,K_u}^{t-1} - \sum_{k=0}^{K_u-1} \eta_u \Gamma_{u,k}^t.$$

Then we prove the recurrence between adjacent rounds. Let $\mathbf{P} = \frac{1}{m}\mathbf{1}\mathbf{1}^\top \in \mathbb{R}^{m \times m}$ and $\mathbf{I} \in \mathbb{R}^{m \times m}$ is the identity matrix, due to the double stochastic property of the adjacent matrix $\mathbf{W}$, we have:

$$\mathbf{W}\mathbf{P} = \mathbf{P}\mathbf{W} = \mathbf{P}.$$

Thus,

$$(\mathbf{I} - \mathbf{P}) \Phi_{u,K_u}^t = (\mathbf{I} - \mathbf{P}) \mathbf{W}\Phi_{u,K_u}^{t-1} - (\mathbf{I} - \mathbf{P}) \sum_{k=0}^{K_u-1} \eta_u \Gamma_{u,k}^t$$
$$= \left( \mathbf{W}\Phi_{u,K_u}^{t-1} - \sum_{k=0}^{K-1} \eta_u \Gamma_{u,k}^t \right) - \mathbf{P}\mathbf{W}\Phi_{u,K_u}^{t-1} + \mathbf{P}\mathbf{W}\Phi_{u,K_u}^{t-1} - \mathbf{P}\left( \mathbf{W}\Phi_{u,K_u}^{t-1} - \sum_{k=0}^{K_u-1} \eta_u \Gamma_{u,k}^t \right).$$

By taking the expectation of the norm on both sides, we have:

$$\mathbb{E} \| (\mathbf{I} - \mathbf{P}) \Phi_{u,K_u}^t \| \leq \mathbb{E}\|\mathbf{W}\Phi_{u,K_u}^{t-1} - \sum_{k=0}^{K_u-1} \eta_u \Gamma_{u,k}^t - \mathbf{P}\mathbf{W}\Phi_{u,K_u}^{t-1}\| + \mathbb{E}\| \sum_{k=0}^{K_u-1} \eta_u \Gamma_{u,k}^t \|$$
$$\leq \mathbb{E}\|\mathbf{W}\Phi_{u,K_u}^{t-1} - \mathbf{P}\mathbf{W}\Phi_{u,K_u}^{t-1}\| + 2\mathbb{E}\| \sum_{k=0}^{K_u-1} \eta_u \Gamma_{u,k}^t \|$$
$$= \mathbb{E}\| (\mathbf{W} - \mathbf{P}) (\mathbf{I} - \mathbf{P}) \Phi_{u,K_u}^{t-1} \| + 2\mathbb{E}\| \sum_{k=0}^{K_u-1} \eta_u \Gamma_{u,k}^t \|$$
$$\leq \lambda\mathbb{E}\| (\mathbf{I} - \mathbf{P}) \Phi_{u,K_u}^{t-1} \| + 2\mathbb{E}\| \sum_{k=0}^{K_u-1} \eta_u \Gamma_{u,k}^t \|.$$

The equality adopts $(\mathbf{W} - \mathbf{P}) (\mathbf{I} - \mathbf{P}) = \mathbf{W} - \mathbf{P} - \mathbf{W}\mathbf{P} + \mathbf{P}\mathbf{P} = \mathbf{W} - \mathbf{P}\mathbf{W}$. We know the fact that $\Phi_{u,k}^t = 0$ where $(t, k) \in (t_0, k_0)$. Thus unwinding the above inequality we have:

$$\mathbb{E} \| (\mathbf{I} - \mathbf{P}) \Phi_{u,K_u}^t \| \leq \lambda^{t-t_0+1}\mathbb{E}\| (\mathbf{I} - \mathbf{P}) \Phi_{u,K_u}^{t_0-1} \| + 2 \sum_{s=t_0}^{t} \lambda^{t-s}\mathbb{E}\| \sum_{k=0}^{K_u-1} \eta_u \Gamma_{u,k}^s \|$$
$$= 2 \sum_{s=t_0}^{t} \lambda^{t-s}\mathbb{E}\| \sum_{k=0}^{K_u-1} \eta_u \Gamma_{u,k}^s \|.$$

To maintain the term of $\mathbf{W}$, we have:

$$\left(\mathbf{W} - \mathbf{P}\right) \Phi_{u,K_u}^t = \left(\mathbf{W} - \mathbf{P}\right) \mathbf{W} \Phi_{u,K_u}^{t-1} - \left(\mathbf{W} - \mathbf{P}\right) \sum_{k=0}^{K_u-1} \eta_u \Gamma_{u,k}^t$$

$$= \left(\mathbf{W} - \mathbf{P}\right) \left(\mathbf{W} - \mathbf{P}\right) \Phi_{u,K_u}^{t-1} - \left(\mathbf{W} - \mathbf{P}\right) \sum_{k=0}^{K_u-1} \eta_u \Gamma_{u,k}^t.$$

The second equality adopts $\left(\mathbf{W} - \mathbf{P}\right)\left(\mathbf{W} - \mathbf{P}\right) = \left(\mathbf{W} - \mathbf{P}\right)\mathbf{W} - \mathbf{W}\mathbf{P} + \mathbf{P}\mathbf{P} = \left(\mathbf{W} - \mathbf{P}\right)\mathbf{W}$. Therefore we have the following recursive formula:

$$\mathbb{E}\|\left(\mathbf{W} - \mathbf{P}\right) \Phi_{u,K_u}^t \| \le \mathbb{E}\|\left(\mathbf{W} - \mathbf{P}\right)\left(\mathbf{W} - \mathbf{P}\right) \Phi_{u,K_u}^{t-1}\| + \mathbb{E}\|\left(\mathbf{W} - \mathbf{P}\right) \sum_{k=0}^{K-1} \eta_u \Gamma_{u,k}^t\|$$

$$\le \lambda \mathbb{E}\|\left(\mathbf{W} - \mathbf{P}\right) \Phi_{u,K_u}^{t-1}\| + \lambda \mathbb{E}\| \sum_{k=0}^{K-1} \eta_u \Gamma_{u,k}^t\|.$$

The same as above, we can unwind this recurrence formulation from $t$ to $t_0$ as:

$$\mathbb{E}\|\left(\mathbf{W} - \mathbf{P}\right) \Phi_{u,K_u}^t \| \le \lambda^{t-t_0+1} \mathbb{E}\|\left(\mathbf{W} - \mathbf{P}\right) \Phi_{u,K_u}^{t_0-1}\| + \sum_{s=t_0}^{t} \lambda^{t-s+1} \mathbb{E}\| \sum_{k=0}^{K_u-1} \eta_u \Gamma_{u,k}^s\|$$

$$= \sum_{s=t_0}^{t} \lambda^{t-s+1} \mathbb{E}\| \sum_{k=0}^{K_u-1} \eta_u \Gamma_{u,k}^s\|.$$

Unwinding the summation on $k$ and adopting Lemma 5, we have:

$$\sum_{s=t_0}^{t} \lambda^{t-s} \mathbb{E}\| \sum_{k=0}^{K_u-1} \eta_u \Gamma_{u,k}^s\|$$

$$\le \sum_{s=t_0}^{t} \lambda^{t-s} \sum_{k=0}^{K_u-1} \mathbb{E}\| \eta_u \Gamma_{u,k}^s\|$$

$$\le \frac{2\mu_u(\sigma_u + \delta_u)}{S\tau_0^{\mu_u L_u}} \sum_{s=t_0}^{t} \lambda^{t-s} \sum_{k=0}^{K_u-1} \frac{\tau^{\mu_u L_u}}{\tau} + \left(\frac{L_{uv}}{L_u}\right) \frac{2\mu_u\sigma_v}{S\tau_0^{\mu_v L_v}} \sum_{s=t_0}^{t} \lambda^{t-s} \sum_{k=0}^{K_u-1} \frac{\tau^{\mu_v L_v}}{\tau}$$

$$\le \frac{2\mu_u(\sigma_u + \delta_u)}{S\tau_0^{\mu_u L_u}} \sum_{s=t_0}^{t} \lambda^{t-s} \sum_{k=0}^{K_u-1} \frac{(sK_u)^{\mu_u L_u}}{sK_u} + \left(\frac{L_{uv}}{L_u}\right) \frac{2\mu_u\sigma_v}{S\tau_0^{\mu_v L_v}} \sum_{s=t_0}^{t} \lambda^{t-s} \sum_{k=0}^{K_u-1} \frac{(sK_u)^{\mu_v L_v}}{sK_u}$$

$$= \frac{2\mu_u(\sigma_u + \delta_u)}{S} \left(\frac{K_u}{\tau_0}\right)^{\mu_u L_u} \sum_{s=t_0}^{t} \frac{\lambda^{t-s}}{s^{1-\mu_u L_u}} + \left(\frac{L_{uv}}{L_u}\right) \frac{2\mu_u\sigma_v}{S} \left(\frac{K_u}{\tau_0}\right)^{\mu_v L_v} \sum_{s=t_0}^{t} \frac{\lambda^{t-s}}{s^{1-\mu_v L_v}}$$

$$\le \frac{2\mu_u(\sigma_u + \delta_u)}{S} \left(\frac{K_u}{\tau_0}\right)^{\mu_u L_u} \sum_{s=t_0-1}^{t-1} \frac{\lambda^{t-s-1}}{(s+1)^{1-\mu_u L_u}} + \left(\frac{L_{uv}}{L_u}\right) \frac{2\mu_u\sigma_v}{S} \left(\frac{K_u}{\tau_0}\right)^{\mu_v L_v} \sum_{s=t_0-1}^{t-1} \frac{\lambda^{t-s-1}}{(s+1)^{1-\mu_v L_v}}$$

$$\le \frac{2\mu_u(\sigma_u + \delta_u)\kappa_\lambda}{S} \left(\frac{K_u}{\tau_0}\right)^{\mu_u L_u} \frac{1}{t^{1-\mu_u L_u}} + \left(\frac{L_{uv}}{L_u}\right) \frac{2\mu_u\sigma_v\kappa_\lambda}{S} \left(\frac{K_u}{\tau_0}\right)^{\mu_v L_v} \frac{1}{t^{1-\mu_v L_v}}.$$

Therefore, we get an upper bound on the aggregation gap which is related to the spectrum gap:

$$\mathbb{E}\| \left( \mathbf{I} - \mathbf{P} \right) \Phi_{u,K_u}^t \| \leq 2 \sum_{s=t_0}^{t} \lambda^{t-s} \mathbb{E}\| \sum_{k=0}^{K_u-1} \eta_u \Gamma_{u,k}^s \|$$

$$\leq \frac{4\mu_u(\sigma_u + \delta_u)\kappa_\lambda}{S} \left( \frac{K_u}{\tau_0} \right)^{\mu_u L_u} \frac{1}{t^{1-\mu_u L_u}} + \left( \frac{L_{uv}}{L_u} \right) \frac{4\mu_u \sigma_v \kappa_\lambda}{S} \left( \frac{K_u}{\tau_0} \right)^{\mu_v L_v} \frac{1}{t^{1-\mu_v L_v}},$$

$$\mathbb{E}\| \left( \mathbf{W} - \mathbf{P} \right) \Phi_{u,K_u}^t \| \leq \sum_{s=t_0}^{t} \lambda^{t-s+1} \mathbb{E}\| \sum_{k=0}^{K-1} \eta_u \Gamma_{u,k}^s \|$$

$$\leq \frac{2\mu_u(\sigma_u + \delta_u)\lambda\kappa_\lambda}{S} \left( \frac{K_u}{\tau_0} \right)^{\mu_u L_u} \frac{1}{t^{1-\mu_u L_u}} + \left( \frac{L_{uv}}{L_u} \right) \frac{2\mu_u \sigma_v \kappa_\lambda}{S} \left( \frac{K_u}{\tau_0} \right)^{\mu_v L_v} \frac{1}{t^{1-\mu_v L_v}}.$$

The first inequality provides the upper bound between the difference between the averaged state and the vanilla state, and the second inequality provides the upper bound between the aggregated state and the averaged state. $\qquad \square$

**Main Proof for Theorem 2** According to the Lemma 4 and 8, it is easy to bound the local stability. We obverse it when the event $\xi$ happens, and we have $\Delta_{k_0}^{t_0} = 0$. Therefore, we unwind the recurrence formulation from $T, K$ to $t_0, k_0$. Let $\eta_u = \frac{\mu_u}{\tau} = \frac{\mu_u}{tK+k}$ and $\eta_v = \frac{\mu_v}{\tau} = \frac{\mu_v}{tK+k}$ are decayed as the communication round $t$ and iteration $k$ where $\mu_u \leq \frac{1}{L_u}$ and $\mu_v \leq \frac{1}{L_v}$ are specific constants, we have:

$$\sum_{i \in [m]} \mathbb{E}\| u_{i,K_u}^{t+1} - \widetilde{u}_{i,K_u}^{t+1} \|$$

$$= \sum_{i \in [m]} \mathbb{E}\| \left( u_{i,0}^{t+1} - \widetilde{u}_{i,0}^{t+1} \right) - \sum_{k=0}^{K-1} \eta_k^t \left( g_{u,i,k}^t - \widetilde{g}_{u,i,k}^t \right) \|$$

$$= \sum_{i \in [m]} \mathbb{E}\| \left( u_{i,0}^{t+1} - \widetilde{u}_{i,0}^{t+1} \right) - \left( u_{i,K_u}^t - \widetilde{u}_{i,K_u}^t \right) + \left( u_{i,K}^t - \widetilde{u}_{i,K_u}^t \right) - \sum_{k=0}^{K_u-1} \eta_u \left( g_{u,i,k}^t - \widetilde{g}_{u,i,k}^t \right) \|$$

$$\leq \sum_{i \in [m]} \left[ \mathbb{E}\| \left( u_{i,0}^{t+1} - \widetilde{u}_{i,0}^{t+1} \right) - \left( u_{i,K_u}^t - \widetilde{u}_{i,K_u}^t \right) \| + \mathbb{E}\| \left( u_{i,K_u}^t - \widetilde{u}_{i,K_u}^t \right) \| + \mathbb{E}\| \sum_{k=0}^{K_u-1} \eta_u \left( g_{u,i,k}^t - \widetilde{g}_{u,i,k}^t \right) \| \right]$$

$$\leq \sum_{i \in [m]} \mathbb{E}\| \left( u_{i,K_u}^t - \widetilde{u}_{i,K_u}^t \right) \| + \sum_{i \in [m]} \mathbb{E}\| \sum_{k=0}^{K_u-1} \eta_u \left( g_{u,i,k}^t - \widetilde{g}_{u,i,k}^t \right) \|$$

$$+ m\mathbb{E}\left[ \frac{1}{m} \sum_{i \in [m]} \| \left( u_{i,0}^{t+1} - \widetilde{u}_{i,0}^{t+1} \right) - \left( u_{i,K_u}^t - \widetilde{u}_{i,K_u}^t \right) \| \right]$$

$$\leq \sum_{i \in [m]} \mathbb{E}\| \left( u_{i,K_u}^t - \widetilde{u}_{i,K_u}^t \right) \| + \sum_{i \in [m]} \mathbb{E}\| \sum_{k=0}^{K_u-1} \eta_u \left( g_{u,i,k}^t - \widetilde{g}_{u,i,k}^t \right) \|$$

$$+ m\mathbb{E}\sqrt{ \frac{1}{m} \sum_{i \in [m]} \| \left( u_{i,0}^{t+1} - \widetilde{u}_{i,0}^{t+1} \right) - \left( u_{i,K_u}^t - \widetilde{u}_{i,K_u}^t \right) \|^2 }$$

$$= \sum_{i \in [m]} \mathbb{E}\| \left( u_{i,K_u}^t - \widetilde{u}_{i,K_u}^t \right) \| + \sqrt{m}\mathbb{E}\| \Phi_{u,0}^{t+1} - \Phi_{u,K_u}^t \| + \sum_{i \in [m]} \mathbb{E}\| \sum_{k=0}^{K_u-1} \eta_u \left( g_{u,i,k}^t - \widetilde{g}_{u,i,k}^t \right) \|$$

Let $\Phi_{u,0}^{t+1} = \mathbf{W}\Phi_{u,K_u}^t$, we have:

$$\sum_{i\in[m]} \mathbb{E}\|u_{i,K_u}^{t+1} - \widetilde{u}_{i,K_u}^{t+1}\|$$

$$\leq \sum_{i\in[m]} \mathbb{E}\| \left(u_{i,K_u}^t - \widetilde{u}_{i,K_u}^t\right)\| + \sqrt{m}\mathbb{E}\|\mathbf{W}\Phi_{u,K_u}^t - \Phi_{u,K_u}^t\| + \sum_{i\in[m]} \mathbb{E}\| \sum_{k=0}^{K_u-1} \eta_u \left(g_{u,i,k}^t - \widetilde{g}_{u,i,k}^t\right)\|$$

$$\leq \sum_{i\in[m]} \mathbb{E}\| \left(u_{i,K_u}^t - \widetilde{u}_{i,K_u}^t\right)\| + \sum_{i\in[m]} \mathbb{E}\| \sum_{k=0}^{K_u-1} \eta_u \left(g_{u,i,k}^t - \widetilde{g}_{u,i,k}^t\right)\|$$
$$+ \sqrt{m}\mathbb{E}\| \left(\mathbf{W} - \mathbf{P}\right)\Phi_{u,K_u}^t\| + \sqrt{m}\mathbb{E}\| \left(\mathbf{P} - \mathbf{I}\right)\Phi_{u,K_u}^t\|.$$

Since $v_{i,0}^{t+1} = v_{i,K_v}^t$ for the private variables, then we have the recursion:

$$\sum_{i\in[m]} \mathbb{E}\|v_{i,K_v}^{t+1} - \widetilde{v}_{i,K_v}^{t+1}\| = \sum_{i\in[m]} \mathbb{E}\| \left(v_{i,0}^{t+1} - \widetilde{v}_{i,0}^{t+1}\right) - \sum_{k=0}^{K_v-1} \eta_k^t \left(g_{v,i,k}^t - \widetilde{g}_{v,i,k}^t\right)\|$$

$$\leq \sum_{i\in[m]} \left[\mathbb{E}\| \left(v_{i,K_v}^t - \widetilde{v}_{i,K_v}^t\right)\| + \mathbb{E}\| \sum_{k=0}^{K_v-1} \eta_v \left(g_{v,i,k}^t - \widetilde{g}_{v,i,k}^t\right)\|\right]$$

$$\leq \sum_{i\in[m]} \mathbb{E}\| \left(v_{i,K_v}^t - \widetilde{v}_{i,K_v}^t\right)\| + \sum_{i\in[m]} \mathbb{E}\| \sum_{k=0}^{K_v-1} \eta_u \left(g_{v,i,k}^t - \widetilde{g}_{v,i,k}^t\right)\|.$$

Therefore, we can bound this by two terms in one complete communication round. One is the process of local multi-times SGD iterations, and the other is the aggregation step. For the local training process, we can continue to use Lemma 7, 8, and 9. Let $\tau = tK + k$ as above, we have:

$$\Delta_{u,K_u}^t + \frac{2(\sigma_u + \delta_u)}{SL_u}$$

$$\leq \left[\prod_{k=0}^{K_u-1} (1 + \eta_u L_u)\right] \left(\Delta_{u,0}^t + \frac{2(\sigma_u + \delta_u)}{SL_u}\right) = \left[\prod_{k=0}^{K_u-1} \left(1 + \frac{\mu_u L_u}{\tau}\right)\right] \left(\Delta_{u,0}^t + \frac{2(\sigma_u + \delta_u)}{SL_u}\right)$$

$$\leq \left[\prod_{k=0}^{K-1} e^{\frac{\mu_u L_u}{\tau}}\right] \left(\Delta_{u,0}^t + \frac{2(\sigma_u + \delta_u)}{SL_u}\right) = e^{\mu L \sum_{k=0}^{K_u-1} \frac{1}{\tau}} \left(\Delta_{u,0}^t + \frac{2(\sigma_u + \delta_u)}{SL_u}\right)$$

$$\leq e^{\mu_u L_u \ln\left(\frac{t+1}{t}\right)} \left(\Delta_{u,0}^t + \frac{2(\sigma_u + \delta_u)}{SL_u}\right) = \left(\frac{t+1}{t}\right)^{\mu_u L_u} \left(\Delta_{u,0}^t + \frac{2(\sigma_u + \delta_u)}{SL_u}\right)$$

$$\leq \left(\frac{t+1}{t}\right)^{\mu_u L_u} \left[\Delta_{u,K_u}^{t-1} + \sqrt{m}(\mathbb{E}\| \left(\mathbf{W} - \mathbf{P}\right)\Phi_{u,K_u}^t\| + \mathbb{E}\| \left(\mathbf{P} - \mathbf{I}\right)\Phi_{u,K_u}^t\|) + \frac{2(\sigma_u + \delta_u)}{SL_u}\right]$$

$$\leq \left(\frac{t+1}{t}\right)^{\mu_u L_u} \left(\Delta_{u,K_u}^{t-1} + \frac{2(\sigma_u + \delta_u)}{SL_u}\right) + \sqrt{m} \left(\frac{t+1}{t}\right)^{\mu_u L_u} (\mathbb{E}\| \left(\mathbf{W} - \mathbf{P}\right)\Phi_{u,K_u}^t\| + \mathbb{E}\| \left(\mathbf{P} - \mathbf{I}\right)\Phi_{u,K_u}^t\|)$$

$$\leq \underbrace{\left(\frac{t+1}{t}\right)^{\mu_u L_u} \left(\Delta_{u,K_u}^{t-1} + \frac{2(\sigma_u + \delta_u)}{SL_u}\right)}_{\text{local updates}} + \underbrace{\frac{6\sqrt{m}\mu_u(\sigma_u + \delta_u)\kappa_\lambda}{S} \left(\frac{K_u}{\tau_0}\right)^{\mu_u L_u} \left(\frac{t+1}{t}\right)^{\mu_u L_u} \frac{1}{t^{1-\mu_u L_u}}}_{\text{aggregation gaps}}$$

$$+ \underbrace{(\frac{L_{uv}}{L_u})\frac{6\sqrt{m}\mu_u\sigma_v\kappa_\lambda}{S} \left(\frac{K_u}{\tau_0}\right)^{\mu_v L_v} \left(\frac{t+1}{t}\right)^{\mu_v L_v} \frac{1}{t^{1-\mu_v L_v}}}_{\text{aggregation gaps}}.$$

The last adopts the Eq.(22) and (23), and the fact $\lambda \le 1$. Obviously, in the decentralized federated learning setup, the first term still comes from the updates of the local training. The second term comes from the aggregation gaps, which is related to the spectrum gap $\lambda$.

For the private variables, since we do not exchange them with neighbors, we have:

$$
\begin{aligned}
\Delta_{v,K_v}^t + \frac{2\sigma_v}{SL_v} &\le \left[ \prod_{k=0}^{K_v-1} (1 + \eta_v L_v) \right] \left( \Delta_{v,0}^t + \frac{2\sigma_v}{SL_v} \right) \\
&= \left[ \prod_{k=0}^{K_v-1} \left( 1 + \frac{\mu_v L_v}{\tau} \right) \right] \left( \Delta_{v,0}^t + \frac{2\sigma_v}{SL_v} \right) \\
&\le \left[ \prod_{k=0}^{K_v-1} e^{\frac{\mu_v L_v}{\tau}} \right] \left( \Delta_{v,0}^t + \frac{2\sigma_v}{SL_v} \right) \\
&= e^{\mu_v L_v \sum_{k=0}^{K_v-1} \frac{1}{\tau}} \left( \Delta_{v,0}^t + \frac{2\sigma_v}{SL_v} \right) \\
&\le e^{\mu_v L_v \ln\left( \frac{t+1}{t} \right)} \left( \Delta_{v,0}^t + \frac{2\sigma_v}{SL_v} \right) \\
&= \left( \frac{t+1}{t} \right)^{\mu_v L_v} \left( \Delta_{v,0}^t + \frac{2\sigma_v}{SL_v} \right).
\end{aligned}
$$

Unwinding this from $t_0$ to $T$, we have:

$$
\begin{aligned}
\Delta_{u,K_u}^T &+ \frac{2(\sigma_u + \delta_u)}{SL_u} \\
&\le \left( \frac{TK_u}{\tau_0} \right)^{\mu_u L_u} \frac{2(\sigma_u + \delta_u)}{SL_u} + \frac{6\sqrt{m}\mu_u(\sigma_u + \delta_u)\kappa_\lambda}{S} \left( \frac{K_u}{\tau_0} \right)^{\mu_u L_u} \sum_{t=t_0+1}^{T} \left( \frac{t+1}{t} \right)^{\mu_u L_u} \frac{1}{t^{1-\mu_u L_u}} \\
&\quad + \frac{6\sqrt{m}\mu_u\sigma_v\kappa_\lambda L_{uv}}{SL_v} \left( \frac{K_u}{\tau_0} \right)^{\mu_v L_v} \sum_{t=t_0+1}^{T} \left( \frac{t+1}{t} \right)^{\mu_v L_v} \frac{1}{t^{1-\mu_v L_v}} \\
&\le \left( \frac{TK_u}{\tau_0} \right)^{\mu_u L_u} \frac{2(\sigma_u + \delta_u)}{SL_u} + \frac{12\sqrt{m}\mu_u(\sigma_u + \delta_u\kappa_\lambda}{S} \left( \frac{K_u}{\tau_0} \right)^{\mu_u L_u} \sum_{t=t_0+1}^{T} \frac{1}{t^{1-\mu_u L_u}} \\
&\quad + (\frac{L_{uv}}{L_u}) \frac{12\sqrt{m}\mu_u\sigma_v\kappa_\lambda}{S} \left( \frac{K_u}{\tau_0} \right)^{\mu_v L_v} \sum_{t=t_0+1}^{T} \frac{1}{t^{1-\mu_v L_v}} \\
&\le \left( \frac{TK_u}{\tau_0} \right)^{\mu_u L_u} \frac{2(\sigma_u + \delta_u)}{SL_u} + \frac{12\sqrt{m}\mu_u(\sigma_u + \delta_u)\kappa_\lambda}{S} \left( \frac{K_u}{\tau_0} \right)^{\mu_u L_u} \frac{t^{\mu_u L_u}}{\mu_u L_u} \Bigg|_{t=t_0+1}^{t=T} \\
&\quad + (\frac{L_{uv}}{L_u}) \frac{12\sqrt{m}\mu_u\sigma_v\kappa_\lambda}{S} \left( \frac{K_u}{\tau_0} \right)^{\mu_v L_v} \frac{t^{\mu_v L_v}}{\mu_v L_v} \Bigg|_{t=t_0+1}^{t=T} \\
&\le \left( \frac{TK_u}{\tau_0} \right)^{\mu_u L_u} \frac{2(1 + 6\sqrt{m}\kappa_\lambda)(\sigma_u + \delta_u)}{SL_u} + \left( \frac{L_{uv}}{L_u} \right) \left( \frac{TK_u}{\tau_0} \right)^{\mu_v L_v} \frac{12\sqrt{m}\kappa_\lambda\sigma_v}{SL_v}.
\end{aligned}
$$

The second inequality adopts the fact that $1 < \frac{t+1}{t} \le 2$ when $t > 1$ and the fact of $0 < \mu < \frac{1}{L}$.

For the personalized variables, unwinding this from $t_0$ to $T$, we have:

$$
\Delta_{v,K_v}^T + \frac{2\sigma_v}{SL_v} \le \left( \frac{TK_v}{\tau_0} \right)^{\mu_v L_v} \frac{2\sigma_v}{SL_v}.
$$

Then the first term in the stability (conditions is omitted for abbreviation) can be bounded as:

$$\mathbb{E}\|u^{T+1} - \widetilde{u}^{T+1}\| \leq \frac{1}{m} \sum_{i \in [m]} \mathbb{E}\| \left( u_{i,K_u}^T - \widetilde{u}_{i,K_u}^T \right) \|$$

$$\leq \left( \frac{TK_u}{\tau_0} \right)^{\mu_u L_u} \frac{2\left(1 + 6\sqrt{m}\kappa_\lambda\right)\left(\sigma_u + \delta_u\right)}{mSL_u} + \left( \frac{L_{uv}}{L_u} \right) \left( \frac{TK_u}{\tau_0} \right)^{\mu_v L_v} \frac{12\sqrt{m}\kappa_\lambda\sigma_v}{mSL_v},$$

$$\mathbb{E}\|v^{T+1} - \widetilde{v}^{T+1}\| \leq \frac{1}{m} \sum_{i \in [m]} \mathbb{E}\| \left( v_{i,K_u}^T - \widetilde{v}_{i,K_u}^T \right) \| \leq \left( \frac{TK_v}{\tau_0} \right)^{\mu_v L_v} \frac{2\sigma_v}{SL_v}.$$

Therefore, we can upper bound the stability in decentralized federated learning as:

$$\mathbb{E}\left[ \|f(w_i^{T+1}; z) - f(\widetilde{w}_i^{T+1}; z)\| \right]$$

$$\leq G\mathbb{E}\left[ \|w_i^{T+1} - \widetilde{w}_i^{T+1}\| \mid \xi \right] + \frac{U\tau_0}{S}$$

$$\leq G\mathbb{E}\left[ \|u^{T+1} - \widetilde{u}^{T+1}\| \mid \xi \right] + G\mathbb{E}\left[ \|v^{T+1} - \widetilde{v}^{T+1}\| \mid \xi \right] + \frac{U\tau_0}{S}$$

$$\leq \frac{2(\sigma_u + \delta_u)G}{SL_u} \left( \frac{1 + 6\sqrt{m}\kappa_\lambda}{m} \right) \left( \frac{TK_u}{\tau_0} \right)^{\mu_u L_u} + \frac{U\tau_0}{S}$$

$$+ \left( \frac{L_{uv}}{L_u} \right) \left( \frac{TK_v}{\tau_0} \right)^{\mu_v L_v} \frac{12\sqrt{m}\kappa_\lambda\sigma_v G}{SL_v} + \frac{2\sigma_v G}{SL_v} \left( \frac{TK_v}{\tau_0} \right)^{\mu_v L_v}$$

$$\leq \frac{2(\sigma_u + \delta_u)G}{SL_u} \left( \frac{1 + 6\sqrt{m}\kappa_\lambda}{m} \right) \left( \frac{TK_u}{\tau_0} \right)^{\mu_u L_u} + \frac{2\sigma_v G}{SL_v} \left( \frac{TK_v}{\tau_0} \right)^{\mu_v L_v}$$

$$+ \frac{12\sqrt{m}\kappa_\lambda\sigma_v L_{uv}}{mSL_v L_u} \left( \frac{TK_u}{\tau_0} \right)^{\mu_v L_v} + \frac{U\tau_0}{S}.$$

The same as the centralized setup, we can select a proper event $\xi$ with a proper $\tau_0$ to minimize the error of the stability. To simplify subsequent analysis, we assume $\mu L = \max\{\mu_u L_u, \mu_v L_v\}$ and $K = \max\{K_u, K_v\}$. For $\tau \in [1, TK]$, by selecting $\tau_0 = (TK)^{\frac{\mu L}{1+\mu L}} \left[ \frac{2G(\sigma_u + \delta_u)L_v^2(1 + 6\sqrt{m}\kappa_\lambda) + 2G\sigma_v L_u L_{uv}(m + 6\sqrt{m}\kappa_\lambda)}{UmL_u L_v^2} \right]^{\frac{1}{1+\mu L}}$, we get the minimal generalization bound for D-PFL:

$$\mathbb{E}\left[ \|f(u^T, V^T; z_j) - f(\widetilde{u}^T, \widetilde{V}^T; z_j)\| \right]$$

$$\leq \frac{4}{S} (UTK)^{\frac{\mu L}{1+\mu L}} \left[ \frac{(\sigma_u + \delta_u)G}{L_u m}(1 + 6\sqrt{m}\kappa_\lambda) + \frac{\sigma_v G}{L_v}(1 + \frac{6\sqrt{m}\kappa_\lambda L_{uv}}{mL_u}) \right]^{\frac{1}{1+\mu L}}.$$

