# OpenReview forum: "Understanding the Stability-based Generalization of Personalized Federated Learning"
_ICLR.cc/2025/Conference — ICLR 2025 Poster_

### Official Review · Reviewer_45Sj · 2024-10-29

**Soundness:** 4
**Presentation:** 3
**Contribution:** 3
**Rating:** 8
**Confidence:** 4

**Summary:**

The paper addresses an important theoretical gap in Personalized Federated Learning (PFL) by focusing on generalization performance beyond convex conditions, specifically analyzing non-convex settings. The authors introduce a generalization analysis that incorporates algorithm-dependent factors through uniform stability. The paper investigates the effect of hyperparameters, learning rates, and communication modes (Centralized vs. Decentralized PFL) on generalization performance. Additionally, they provide excess risk bounds and propose early stopping criteria for optimal risk in PFL, supported by experiments on the CIFAR dataset.

**Strengths:**

The major contribution of this paper is from the theoretical aspect in generalization.

1. Generalization in (personalized) federated learning is critically important yet understudied, largely due to its complexity compared to the extensive body of work on convergence analysis in FL.

2. This paper offers a rigorous analysis, with well-rounded discussions and comparisons that cover various specific cases.

2. By focusing on the “algorithm-matter” generalization through uniform stability, the authors provide a practical framework that incorporates the effects of stepsize, learning steps, and communication structure (C-PFL vs. D-PFL). This analysis fills a significant gap in the field by moving beyond static theoretical assumptions.

3. The detailed examination of hyperparameters like learning steps and stepsizes, along with communication modes, is valuable for both theoretical insights and practical implications. The finding that C-PFL generalizes better than D-PFL is intriguing and has practical relevance.

**Weaknesses:**

I don’t see any major weaknesses in the paper. However, I suggest: 1) incorporating additional datasets and models in the experiments to strengthen the findings; and 2) clarifying the notation throughout, e.g., 'U' in the theorem.

**Questions:**

see the weakness part.

---

> ### Author Response · Authors · 2024-11-23
> **To Reviewer 45Sj （1）**
>
> We greatly appreciate your suggestions and comments, which will make our work better than this version. Thanks for your positive comments, and we address the weaknesses and minor comments below. All the suggestions have been modified in the revision paper in purple from you and in blue for all reviewers.
>
> **W1:**
>  We add more experiments on CIFAR-100 with VGG11 in the appendix D in the revision. The additional results can further strongly support our theoretical findings.
> - **Empirical setup:** We conduct experiments on CIFAR-100 in the Pathological distribution (Non-IID c = 20) with VGG-11 for both C-PFL and D-PFL. To verify the impacts of the key hyperparameters, we study the parameter distance when disturbing only one data, the generalization gap of the difference between training and testing error, and testing performance when personalized training. We explore the impact of the four factors: 1) Local Learning Epochs, 2) Local Learning Rates, 3) Client Fraction / Communication topology, 4) Total Client Number. The detailed setup can be seen in Appendix D.1.
> - **Empirical analysis:** From the empirical results, we draw the similar conclusions as CIFAR-10: 1)Less local learning epochs and lower learning rates lead to better generalization performance, but they affect the convergence speed more seriously; 2) More client participation and denser network connection enlarge the generalization gap, but they speed up the convergence to the same extent; 3) A larger total participation clients and a smaller number of local training samples increase the generalization error and reduce the convergence speed simultaneously; 4) C-PFL outperforms D-PFL in both generalization performance and convergence performance when their upper communication bandwidths are at the same level. Compared to the easier dataset CIFAR-10, the generalization error on CIFAR-100 is larger.
>
> **W2:** We have polished it in the revision.  $U= sup_{u,v_i,z}f(u,v_i;z)$ is the upper bound of the personalzied loss function.
>
> Thanks for your comments and your suggestions are very meaningful to us. All the suggestions have been modified in the revision paper in purple for you and in blue for all reviewers. lf you have any further questions, please do not hesitate to contact us. Thanks again!

---

> ### Author Response · Authors · 2024-11-25
> **Look forward to further discussions**
>
> Dear reviewer,
>
> We hope this message finds you well. We greatly appreciate your time and effort in reviewing our paper. We have carefully answered all the questions you raised about our paper and look forward to further discussions with you.

---

### Official Review · Reviewer_Nkbg · 2024-11-01

**Soundness:** 3
**Presentation:** 3
**Contribution:** 3
**Rating:** 6
**Confidence:** 4

**Summary:**

This paper establishes a generalization stability analysis for a widely-used personalized algorithm, partial federated learning (PFL), in both centralized and decentralized federated learning (FL) settings. The authors also discuss the relationship between the derived stability results and previously established generalization stability upper bounds, and provide insights into  how these findings can inform and guide the training process.

**Strengths:**

The paper presents a well-organized structure and provides a clear comparison between its results and prior findings. Additionally, it offers an in-depth discussion on the impact of hyperparameters on generalization performance.

**Weaknesses:**

1. Inconsistent Notation: The notation used throughout the paper lacks alignment, which hinders readability. For instance, in Algorithm 2, the aggregation step update is defined as $u^{t+1}=\frac{1}{n}\sum u_i^{t+1}$, but in line 866, the update formula appears as ${\frac{1}{n}}\sum_{i\in \mathcal{N}}u_{i, K_u}^t$. It is unclear what $\mathcal{N}$  represents in this context. Additionally, in line 812, the notation $I$ is introduced without definition. This inconsistency in notation complicates understanding and interpretation.
2. Unclear Experimental Validation of the Proposed Theorem: It is difficult to discern how the experimental results support the theoretical findings. The authors establish generalization stability, yet in the experiments, they only measure the gap between training accuracy and test accuracy. There is no clear implementation of perturbations as discussed in the theoretical framework, making it unclear how the experiments substantiate the proposed stability theorem.

**Questions:**

1. In line 803, could you clarify how the inequality is derived? Specifically, how is $\Vert f(w;z) - f(\tilde{w};z)\Vert^2$ bounded given that $U=sup_{w, z}f(w;z)$.
2. In line 806, it appears there should be an inequality rather than an equality, since $\Vert a + b \Vert \leq \Vert a \Vert + \Vert b\Vert$.
3. The notation $I$ requires further explanation, as it is unclear how to derive the bound for $P(\{\xi^c\})$;
4. Since the authors discuss generalization stability, I recommend comparing the results with similar studies focused on generalization stability rather than generalization bounds as shown in Table 1.

---

> ### Author Response · Authors · 2024-11-23
> **To Reviewer Nkbg (1)**
>
> We greatly appreciate your suggestions and comments, which will make our work better than this version. Thanks for your positive comments, and we address the weaknesses and minor comments below. All the suggestions have been modified in the revision paper in violet for you and in blue for all reviewers.
>
> **Weaknesses:**
>
> **W1:** Thank you for your kind advice and we have corrected the typos in the revision.
> - $u^{t+1}=\frac{1}{n} \sum u_i^{t+1}$ in Algorithm 2 should be $u^{t+1}=\frac{1}{n} \sum_{i\in S^t} u_i^{t+1}$. $\frac{1}{n} \sum_{i \in \mathcal{N}} u_{i, K_u}^t$. Line 1002 should be $\frac{1}{n} \sum_{i \in S^t } u_{i}^{t+1} = \frac{1}{n} \sum_{i \in S^t } u_{i, K_u}^t$ in the revision.
> - $\mathcal{N}$ represents the client set in D-PFL defined in Line 207 in the revision. We check all the use of  $\mathcal{N}$ in the revision.
> - $I$ is the index of the first time to sample the perturbation sample $z_{i^*,j^*}$. When $t_0K+k_0 < I$, $\Delta_{k_0}^{t_0} = 0$. We add this explanation in Line 931 in the revision.
>
> **W2:**
> - Thank you for your kind advice and we added the perturbations experiments on CIFAR10/100 in Figure 1(a) and Figure 3(a) in the revision. We measure the model parameter difference between the perturbations datasets as [1-2] to further establish the nature of generalization stability. The empirical results also strongly support our theoretical insights that: 1) Fewer local learning epochs and lower learning rates lead to better generalization performance; 2) More client participation and denser network connection enlarge the generalization gap; 3) Larger total participation clients and a smaller number of local training samples increase the generalization error; 4) C-PFL outperforms D-PFL in generalization performance when their upper communication bandwidths are at the same level.
> - We present the gap between training loss and test loss followed [2-5] rather than the gap between training accuracy and test accuracy in the last version. These results are reserved in Figure 1(b) and Figure 3(b) in the revision. In realistic experiments, we want to improve the accuracy in the test dataset with a well-trained model in the training dataset with the same data distribution. Therefore, we also reserve the curves of test accuracy in Figure 1(c) and Figure 3(c), which is the final aim in PFL.
>
> [1] Miaoxi Zhu, Li Shen, Bo Du, and Dacheng Tao. Stability and generalization of the decentralized stochastic gradient descent ascent algorithm. Advances in Neural Information Processing Systems, 36, 2024.
>
> [2] Moritz Hardt, Ben Recht, and Yoram Singer. Train faster, generalize better: Stability of stochastic gradient descent. In International conference on machine learning, pp. 1225–1234. PMLR, 2016.
>
> [3] Zhenyu Sun, Xiaochun Niu, and Ermin Wei. Understanding generalization of federated learning via stability: Heterogeneity matters. In International Conference on Artificial Intelligence and Statistics, pp. 676–684. PMLR, 2024b.
>
> [4] Tongtian Zhu, Fengxiang He, Lan Zhang, Zhengyang Niu, Mingli Song, and Dacheng Tao. Topology aware generalization of decentralized sgd. In International Conference on Machine Learning, ICML, pp. 27479–27503. PMLR, 2022.
>
> [5] Tao Sun, Dongsheng Li, and Bao Wang. Stability and generalization of decentralized stochastic gradient descent. In Proceedings of the AAAI Conference on Artificial Intelligence, volume 35, pp.9756–9764, 2021.
>
> **Questions:**
>
> **Q1:**  We define $U=\sup_{w_i,z}f(w_i;z)$ so $\|f(w_i; z)-f(\tilde{w}_i ; z)\| \leq \sup_{w_i,z}f(w_i;z)$ in Line 286 and 927 in the revision.
>
> **Q2:** Thank you and we have polished it in the revision.
>
> **Q3:**  $I$ in Line 812 (last version) is the index of the first time to sample the perturbation sample $z_{i^*,j^*}$. When $t_0K+k_0 < I$, $\Delta_{k_0}^{t_0} = 0$. And $P\left(\xi^c\right)=P\left(\Delta_{k_0}^{t_0}>0\right) \leq P\left(I \leq t_0 K+k_0\right)$. We add this explanation in Line 931 in the revision.
>
> **Q4:** We polish the comparisons with generalization stability of central and decentralized learning in Table 1 in the revision. And the comparisons with generalization bound with PFL methods are removed to Appendix B.
>
> Thanks for your comments and your suggestions are very meaningful to us. All the suggestions have been modified in the revision paper in violet for you and in blue for all reviewers..  lf you have any further questions, please do not hesitate to contact us. Thanks again!

---

> ### Author Response · Authors · 2024-11-25
> **Look forward to further discussions**
>
> Dear reviewer,
>
> We hope this message finds you well. We greatly appreciate your time and effort in reviewing our paper. We have carefully answered all the questions you raised about our paper and look forward to further discussions with you.

---

### Official Review · Reviewer_tNFV · 2024-11-03

**Soundness:** 2
**Presentation:** 1
**Contribution:** 2
**Rating:** 3
**Confidence:** 4

**Summary:**

This paper derives uniform-stability-based generalization bounds (as defined in [1]) for personalized federated learning (FL). Bounds are derived in both the centralized setting (where there is a central server) and the decentralized setting. By combining the derived bounds with existing bounds on the optimization error, the paper provides an expression for the number of communication rounds which they claim is the optimal number of rounds to minimize the total (= optimization + generalization) error. Some experiments are performed to validate some of the theoretical insights.

[1]: Moritz Hardt, Ben Recht, and Yoram Singer. Train faster, generalize better: Stability of stochastic gradient descent. In *International conference on machine learning*, pp. 1225–1234. PMLR, 2016.

**Strengths:**

This paper considers an important problem of trying to quantify the generalization properties of personalized FL. An important by-product of such an analysis is quantifying the optimal number of communication rounds that minimizes the overall error (= optimization error + generalization error); papers on convergence guarantees do not consider the generalization error.

**Weaknesses:**

**Technical issues:**

**1.** In Remark 4/Corollary 1 and Remark 8/Corollary 2, where the authors are deriving the "optimal" number of communication rounds $T^{*}$, how do the authors know that the derived generalization bounds are order-optimal w.r.t. $T$, $K$ and other important quantities. If the tightness of the derived generalization bounds is not shown, then it is unfair to call it "optimal" number of communication rounds. Moreover, in Corollary 1 and 2, the dependence on other problem-specific parameters (such as smoothness constants, stochastic gradient variance, etc.) shouldn't be ignored if they are being used to derive the "optimal" number of communication rounds.

**2.** Can the authors please summarize the technical challenges compared to the seminal analysis in Hardt et al., (2016) for SGD? Is there any particular challenge due to local updates in the clients?

**3.** Overall I'm not too impressed/surprised by the derived results -- a more satisfactory result for me would have been something that shows the generalization benefits of personalization compared to *no* personalization.

**Presentation issues:**

**1.** The presentation of Theorem 1 and Theorem 2 is rather poor and complicated. What is the meaning of "*They decay per iteration $\tau = t K + k$,...*"? And the equations below have $\tau_0$ instead of $\tau$. I'd have directly presented eq. (8) and (10) with the optimal value of $\tau_0$ rather than presenting the intermediate results (eq. (7) and (9)). And what is $\kappa_\lambda$ in the context of D-PFL? Also, it is very hard to parse Table 1.

**2.** Definition 2 is not clear to me -- what is the index $j$ (subscript of $z$)? In eq. (4), a bracket is missing at the end in $F(\mathcal{A}(S)$ and $f(\mathcal{A}(S)$. In Assumption 2, what is $F_i$? Only $f_i$ was introduced earlier.

**3.** Paper writing needs to be improved. For instance, I'd suggest using "algorithm-dependent" instead of "algorithm-matter". Also, it should be "influential" in Remarks 4, 5 and 8.

**Questions:**

Please address the questions in Weaknesses.

---

> ### Author Response · Authors · 2024-11-23
> **To Reviewer tNFV  (1)**
>
> Thanks for your comments and your suggestions are very meaningful to us. We reply to them below and all the suggestions have been modified in the revision paper in cyan for you and  in blue for all reviewers. We sincerely hope that you could reconsider the contributions of our work.
>
> **Technical issues:**
>
> **W1:**
> - Thank you for your kind comments. In the non-convex conditions, we can only obtain the upper bound of excess risk without the lower bound, so it is unsuitable to call it "optimal" number of communication rounds. We polish our analysis about the recommended stop point $T$ and check all the "optimal" in the revision.
> - In our analysis, the excess risk is based on a specific algorithm and setting, so we assume that the problem-specific parameters of smoothness, stochastic gradient variance are fixed to discuss the training hyper-parameters like $T$ and $K$.
>
> **W2:**
> Our technical contributions are as follows:
>
> (1) *We analyze the effect of the **multiple local iterations** and **partial partition**, which bridges centralized SGD and the partial partition FL*. The technical difficulty is that local updates fail to be an unbiased gradient estimation after multiple local iterates, thereby merging the multiple local iterations is non-trivial.
>
> (2) *We analyze the effect of the **personalized training** and **federated aggregation**, which is the first generalization analysis to bridges FL and PFL*. The technical difficulty is that the alignment errors exist in local training of the personalized variables after combining the two parts, thereby understanding the relationship between FL and PFL is non-trivial.
>
> (3) *We analyze the effect of **communication topologies** in Decentralized PFL, which is the first generalization analysis to bridges C-PFL and D-PFL.* Lack of the central server, we need more complicated analysis of the aggregation error introduced by peer-to-peer communication with different gossip matrix $W$. Therefore, understanding the relationship between C-PFL and D-PFL is non-trivial.
>
> (4) *We analyze the effect of **model decoupling** and **alternative training** in both FL \& DFL \& PFL, which is the first generalization analysis to bridges full model personalization and parital model personalization.* We analyze the generalization performance of different model parts separately and obtain the alternative errors between them, which provides interesting and useful suggestions for theoretical analysis of model decoupling and alternative training.
>
> **W3:**
>
> - We add more comparisons between the personalization method and the no personalization method in Table 1 in the revision. Compared to the reference, we are the first stability analysis in both C-PFL and D-PFL with the analysis of multi local updates and the whole training hyperparameters. The contributions in our work can be summarized in the four-holds: (1) extend centralized SGD to the partial partition FL; (2) extend FL to PFL; (3)extend C-PFL to D-PFL; (4)extend full model personalization and parital model personalization.
>
> - Intuitively, whether the personalization generalizes better than no personalization is uncertain，which is related to the data heterogeneity in [1]. Personalization methods  generalizes better than no personalization with more heterogeneity, while no personalization is bettern with more homogeneity. But [1] only proposed the analysis for FL and pure local training rather than PFL and only conducted the analysis under strongly convex conditions. Therefore, it is an interesting problem to compare the PFL methods and the FL methods with the analysis of data heterogeneity in the future.
>
> [1] Shuxiao Chen, Qinqing Zheng, Qi Long, and Weijie J Su. A theorem of the alternative for personalized federated learning. arXiv preprint arXiv:2103.01901, 2021.

---

> ### Author Response · Authors · 2024-11-23
> **To Reviewer tNFV (2)**
>
> **Presentation issues:**
>
> **W1:**
> - "They decay per iteration $\tau = tK+k$" means "The learning rates decay per iteration $\tau = tK+k$". And $\tau_0$ is a specific observation point in the equations below.
> - We present eq. (7) and (9) to extablish the changing progress of the stability, which proves that the generalization gaps in eq. (8) and (10) are available and describe the detailed effect of the hyperparameters.
> - $\kappa_\lambda=\left(\frac{\alpha}{e}\right)^\alpha\frac{1}{\lambda\left(\ln\frac{1}{\lambda}\right)^\alpha} + \frac{2^\alpha}{\left(1-\alpha\right)e\lambda\ln\frac{1}{\lambda}} + \frac{2^\alpha}{\lambda\ln\frac{1}{\lambda}}$ is a widely used coefficient to measure different communication connections. When $\lambda\rightarrow 1$, it is bounded by $\mathcal{O}(\frac{1}{\lambda ln \lambda})$ and when $\lambda\rightarrow 0$, it is bounded by $\mathcal{O}(\frac{1}{\lambda\left(\ln\frac{1}{\lambda}\right)^\alpha})$. This detailed information can be found in Remark 5 in Line 383. We also add more explanation after Eq.(10) in the revision.
> - We refresh the comparisons with the generalization bounds with PFL and add more details of the comparisons methods in Table 2 in Appendix B in the revision.
>
> **W2:** Thank you and we have polished it in the revision.
>
> - $j$ is the $j$ sample in the local training dataset of client $i$. And the $z_j$ in eq.(4) should be $\xi_j$.
> - We have added the bracket in eq. (4) and changed $F_i$ to $F$ in Assumption 2.
>
> **W3:** Thank you and we have polished it in the revision.
>
> Thanks for your comments and your suggestions are very meaningful to us. All the suggestions above have been modified in cyan and  in blue for all reviewers in the revision paper. Please reconsider our contributions to the FL community. If you have any further questions, please do not hesitate to contact us. Thanks again!

---

> ### Author Response · Authors · 2024-11-25
> **Look forward to further discussions**
>
> Dear reviewer,
>
> We hope this message finds you well. We greatly appreciate your time and effort in reviewing our paper. We have carefully answered all the questions you raised about our paper and look forward to further discussions with you.

---

> > ### Comment · Reviewer_tNFV · 2024-11-26
> > **Reply**
> >
> > I thank the authors for their rebuttal and revisions. Unfortunately, I'm not satisfied enough to change my score.
> >
> > **1.** *Regarding $T^{\ast}$ in Remark 4:* Given that the authors are deriving $T^{\ast}$ based on an upper bound and they can't establish a matching lower bound, the result is not strong enough IMO. Moreover, the derived value seems vacuous in the full-device participation case, i.e. $n = m$, or even when $n \geq m^{\frac{1+\mu L}{1 + 2 \mu L}}$. In this case, $T^{\ast} \leq \mathcal{O}(1/K)$, which is bizarre as it indicates there should be no global updates at all (here I have ignored constant factors just like the authors have done).
> >
> > **2.** I understand the analysis of FL requires one to handle the bias due to multiple local updates, but every FL paper does that. To me, the analysis appears to be an extension of the results of Hardt et al., (2016) when there are multiple local updates with personalization. Sure, there are some algebraic challenges in the extension but is there any part of the proof that introduces some new/interesting proof techniques/insights? The authors haven't pointed out anything specific in the rebuttal and their response was very generic. Moreover, there are no matching lower bounds, so it is hard to gauge how tight these bounds are.
> >
> > **3.** What I was looking for when I asked about personalization vs. no personalization was something like the following: in eq. (1), if we have a single $v$ for all $i$ instead of a $v_i$ for each $i$, can we quantify how much worse will one $v$ be as a function of the system heterogeneity? This kind of result would be very interesting to me. The results in this paper look like an extension of Hardt et al. to FL with personalization. And as I explained in point #2, I don't find this very interesting.
> >
> > Due to these reasons, I will keep my score.

---

> > > ### Author Response · Authors · 2024-11-29
> > > **Reply to Reviewer (3)**
> > >
> > > **Reference for Reply (2)**
> > >
> > > [1] Karimireddy, Sai Praneeth, Satyen Kale, Mehryar Mohri, Sashank Reddi, Sebastian Stich, and Ananda Theertha Suresh. Scaffold: Stochastic controlled averaging for federated learning. In International conference on machine learning, pp. 5132-5143. PMLR, 2020.
> > >
> > > [2] Tao Sun, Dongsheng Li, and Bao Wang. Decentralized federated averaging. IEEE Transactions on Pattern Analysis and Machine Intelligence, 2022.
> > >
> > > [3] Krishna Pillutla, Kshitiz Malik, Abdel-Rahman Mohamed, Mike Rabbat, Maziar Sanjabi, and Lin Xiao. Federated learning with partial model personalization. In International Conference on Machine Learning, pp. 17716–17758. PMLR, 2022.
> > >
> > > [4] Yingqi Liu, Yifan Shi, Qinglun Li, Baoyuan Wu, Xueqian Wang, and Li Shen. Decentralized directed collaboration for personalized federated learning. In Proceedings of the IEEE/CVF Conference on Computer Vision and Pattern Recognition, pp. 23168–23178, 2024.
> > >
> > > [5] Chen, Yiming, Liyuan Cao, Kun Yuan, and Zaiwen Wen. Sharper convergence guarantees for federated learning with partial model personalization. arXiv preprint arXiv:2309.17409 (2023).
> > >
> > > [6] Shuxiao Chen, Qinqing Zheng, Qi Long, and Weijie J Su. A theorem of the alternative for personalized federated learning. arXiv preprint arXiv:2103.01901, 2021.
> > >
> > > [7] Tao Sun, Dongsheng Li, and Bao Wang. Stability and generalization of decentralized stochastic gradient descent. In Proceedings of the AAAI Conference on Artificial Intelligence, volume 35, pp. 9756–9764, 2021.
> > >
> > > [8] Xin Zhang, Zhuqing Liu, Jia Liu, Zhengyuan Zhu, and Songtao Lu. Taming communication and sample complexities in decentralized policy evaluation for cooperative multi-agent reinforcement learning. Advances in Neural Information Processing Systems, 34:18825–18838,2021.
> > >
> > > [9]Alexander Rogozin, Aleksandr Beznosikov, Darina Dvinskikh, Dmitry Kovalev, Pavel Dvurechensky, and Alexander Gasnikov. Decentralized distributed optimization for saddle point problems. arXiv preprint arXiv:2102.07758, 2021.
> > >
> > > Thank you for your kind advice! We agree and appreciate the three questions summarized above. These are three very interesting research directions in this field, and they also effectively helped us polish the paper. We must admit that the current work cannot completely solve all the open questions mentioned above in the short rebuttal time, but we still tried our best to optimize the technical innovations and interesting results on personalized analysis, decentralized analysis, and heterogeneous analysis. We sincerely hope that you could reconsider our contributions as the first algorithm-dependent generalization analysis of C-PFL and D-PFL with model decoupling and alternative training. If you have any further questions, please do not hesitate to contact us. Thanks again!

---

> > > ### Author Response · Authors · 2024-12-02
> > > **Look forward to further discussions**
> > >
> > > Dear reviewer,
> > >
> > > We hope this message finds you well.
> > >
> > > Thank you for your earlier follow-up comment. We have provided further clarification on the new question you raised and would like to know if our reply has satisfactorily addressed your concern.
> > >
> > > As the rebuttal deadline is approaching, we kindly ask that if you have any remaining questions or concerns regarding our paper, please let us know at your earliest convenience so that we can address them and get back to you before the deadline.
> > >
> > > If our reply has resolved your concerns, we sincerely hope that you consider raising your rating of our paper. We would be deeply grateful for your recognition.
> > >
> > > Thank you once again for your time and kindness.

---

> > > > ### Comment · Reviewer_tNFV · 2024-12-02
> > > > **Follow up**
> > > >
> > > > Thank you for the detailed response! The response A3 seems like a step in the right direction (although, I'm not sure I fully understand it). However, I'm not convinced enough to change my score.
> > > >
> > > > First, in the stated "global variance assumption", do you need different $u_i$'s or just one $u$ (i.e., $\nabla_u F_i(u, v_i)$ and $\nabla_u F(u, V)$)? Is the bound in Remark 1 of the rebuttal in the no-personalization case? It's hard for me to know for sure from the way it's written ("*Assuming that $v_i$ attends to the aggregation, the generalization error for $v_i$ will add the heterogeneity impact and eq. (1) changes into...*"). Also, I don't understand Remark 2. What is the meaning of "*our analysis with data distribution could obtain a non-iid bound...*"? Moreover, the authors are merely deriving upper bounds, and so it's not fair to claim that their bounds can help to choose between FedAvg and PureLocalTraining (just like the issue in the previous claims about $T^{*}$ in the paper). Further, looking at the bounds in Remark 2, the bound of FedAvg is always better (due to the $1/m$ factor in the second term). So per the author's bounds, FedAvg is always better and there is not really a decision to be made (unless I missed something). Besides Remark 2 seems a bit orthogonal to the comparison of personalized and regular FL (pure local training is not FL).
> > > >
> > > > Overall, the responses about the centralized setting haven't changed my mind to endorse this paper (I'm not very knowledgeable about decentralized learning, so I can't judge how significant the results are). Moreover, poor writing and presentation is a recurring theme in this paper and discussions -- to the extent that it is hard to understand several things. This is another reason I feel this paper is not ready to be published at this time.

---

> > > > > ### Author Response · Authors · 2024-12-03
> > > > > **Reply to Reviewer**
> > > > >
> > > > > Thank you for the kind response! We reply to your questions as follows.
> > > > >
> > > > > 1) We introduce the Assumption 3 (Partial Gradient Diversity) in [1] with just one $u$. We correct this in the reply box above.
> > > > >
> > > > > 2) The bound in Remark 1 of the rebuttal is for no personalization.  Taking C-PFL as an example, the difference between no personalization and partial model personalization is the aggregation for $v_i$, which determines whether the linear classification layer will be affected by heterogeneity.  We add the detailed proof as follows.
> > > > > - For the no personalized model $w$, we assume global heterogeneity:  $\frac{1}{m} \sum_{i=1}^m\left\||\nabla F_i\left(w_i\right)-\nabla_u F\left(w\right)\right\||^2 \leq \delta^2$.  For the partial personalized model $(u,v_i）$, we assume shared feature extractor heterogeneity $\delta_u$ and personalized linear classifier $\delta_v$:  $\frac{1}{m} \sum_{i=1}^m\left\||\nabla_u F_i\left(u, v_i\right)-\nabla_u F\left(u, V\right)\right\||^2 \leq \delta_u^2$ and $\frac{1}{m} \sum_{i=1}^m\left\||\nabla_v F_i\left(u, v_i\right)-\nabla_v F\left(u, V\right)\right\|^2 \leq \delta_v^2$.  In training progress, $ \delta_u^2  \leq \delta^2 \leq \delta_u^2 + \delta_v^2$, since $\|| a \||^2 \leq \|| a+ b \||^2 \leq \||a\||^2 +\||b\||^2$.
> > > > > - Considering the impact of data heterogeneity, we list the generalization bound below for the comparison of FL, PFL, and PureLocalTraining.   In the case of severe heterogeneity, $\delta$ mainly depends on $\delta_v$, which is why the linear classification layer is used as personalized variables. Partial personalized methods significantly improve generalization capabilities and experimental performance compared to FL by eliminating the influence of $\delta_v$. We refresh the presentation of symbols to provide a clearer reading in the reply box above.
> > > > > | Algorithm | Generalization Bound | Remark                                                            |
> > > > > |--|---|--|
> > > > > | FL  | $\mathcal{O}\left(\frac{U\tau_0}{S} + \left(\frac{TK}{\tau_0}\right)^{\mu L}\frac{2G(\sigma+\delta)}{mSL}\right)$  | with the global heterogeneity $\delta$        |
> > > > > | PFL  |  $\mathcal{O}\left(\frac{nU\tau_0}{mS}+\left(\frac{TK_u}{\tau_0}\right)^{\mu_u L_u}\frac{2G(\sigma_u+\delta_u)}{mSL_u} +\left(\frac{TK_v}{\tau_0}\right)^{\mu_v L_v} \left( 1 + \frac{L_{uv}}{L_u} (\frac{TK_u}{\tau_0})^{\mu_u L_u} \right) \frac{2G\sigma_v}{SL_v}\right)$  | with the shared parts heterogeneity $\delta_u$                    |
> > > > > | PureLocal Traing |  $\mathcal{O}\left(\frac{U\tau_0}{S} + \left(\frac{TK}{\tau_0}\right)^{\mu L}\frac{2G\sigma}{SL}\right)$ | without heterogeneity but affected by the small number of samples |
> > > > >
> > > > > 3. *"Our analysis with data distribution could obtain a non-iid bound to show that the presumably difficult (infinite dimensional) problem can be reduced to a simple binary decision problem"* means that *"We can derive a generalization bound that takes into account non-iid data distribution"*.
> > > > >
> > > > > 4. There is little work about the lower bound of SGD or bi-level problems [2-5]. They either need strong assumptions such as Lipschitz Hessian or are only conducted under convex conditions.  Also, the results may exceed the upper bound under non-convex conditions with classical smooth assumptions. Moreover, the lower bound for PFL even FL is still blank. Our most significant contributions in this work are to find the algorithm-dependent generalization with iteration property for PFL and obtain
> > > > >  the bound with different communication topologies in C-PFL and D-PFL, which have not been studied before.
> > > > >
> > > > > 5. The bound in Remark 2 for PureLocalTraining does not contain heterogeneous properties. We fixed the typos in the previous rebuttal box and updated it in the table above in answer 2.
> > > > >
> > > > > 6. It is a byproduct of our analysis in Remark 2 in the rebuttal. The comparison between PFL and FL needs to be further discussed in combination with the Lipschitz properties of global and local functions. Considering that the lack of lower bounds makes these discussions imprecise, we will remove these inappropriate remarks and only emphasize the upper bound.
> > > > >
> > > > > We polish our discussions and detailed information above. If you have any further questions, please do not hesitate to contact us. Thanks again!
> > > > >
> > > > > [1] Federated learning with partial model personalization. In International Conference on Machine Learning, pp. 17716–17758. PMLR, 2022.
> > > > >
> > > > > [2]  Stability of sgd: Tightness analysis and improved bounds. In Uncertainty in artificial intelligence, pp. 2364-2373. PMLR, 2022.
> > > > >
> > > > > [3] Stability of stochastic gradient descent on nonsmooth convex losses. Advances in Neural Information Processing Systems 33 (2020): 4381-4391.
> > > > >
> > > > > [4] Lower Bounds of Uniform Stability in Gradient-Based Bilevel Algorithms for Hyperparameter Optimization. In The Thirty-eighth Annual Conference on Neural Information Processing Systems.
> > > > >
> > > > > [5] A theorem of the alternative for personalized federated learning. arXiv preprint arXiv:2103.01901, 2021.

---

> ### Author Response · Authors · 2024-11-29
> **Reply to reviewer (1)**
>
> Thank you for your kind suggestions. We have answered your questions in the comment below.
>
> **A1:** Thank you for your constructive suggestions! We have not discussed the tightness of the generalization bound in our paper due to the fact that we are the first work to analyze the stability and generalization of PFL and we can not make comparisons. And We have to admit that we do not analyze the lower bound of our generalization gap, which is still a valuable and open problem for generalization analysis. We are thinking about this problem and will continue on this as future work inspired by [1]. Moreover, we will delete the discussions of $T^*$ in our work.
>
> **A2：** Thank you for your constructive suggestions!  We illustrate the technical contributions from the perspective of personalized and decentralized analysis as below.
>
> **New techniques about Personalized FL:**
> -  Compared to the analysis of FL, the analysis of Partial Model Personalization is more difficult to construct the alignment error between the shared aggregation variables and the local update. Therefore, we decompose the generalization errors into aggregation errors from shared variables and fine-tuning errors from both shared and personalized variables, then we build the bridge between $\Delta_v$ and $\Delta_u$ according to the gradient estimation process of the personalized training. It is the first work to analyze the generalization impact from personalized variables to shared variables, which uncovers the interaction mechanism between these two updating processes and provides valuable guidance for alternating personalized optimization.
> -  Also, it will be an interesting and open problem to propose the unique stability for PFL. We are thinking about this problem and will continue on this as future work.
>
> **New techniques about Decentralized PFL:**
> - We achieve the tight bound without the strong assumption of bounded gradient compared with D-SGD [2]. We transform the bounded gradient assumption into a functional representation related to T, which constructed a more flexible matrix operation in Lemma 5 and Lemma 10 to establish the relationship between the communication topology and the aggregation error. This technique is non-trivial.
>
> [1] Zhang, Yikai, Wenjia Zhang, Sammy Bald, Vamsi Pingali, Chao Chen, and Mayank Goswami. Stability of sgd: Tightness analysis and improved bounds. In Uncertainty in artificial intelligence, pp. 2364-2373. PMLR, 2022.
>
> [2] Tao Sun, Dongsheng Li, and Bao Wang. Stability and generalization of decentralized stochastic gradient descent. In Proceedings of the AAAI Conference on Artificial Intelligence, volume 35, pp. 9756–9764, 2021.

---

> ### Author Response · Authors · 2024-11-29
> **Reply to Reviewer (2)**
>
> **A3：**  We show the innovative results from the perspective of personalized and decentralized analysis.
>
> **Innovative Results about Personalized Training:** The most basic difference between FL and PFL of the generalization bound comes from the testing dataset. That is to say, the global aggregation does not bring extra generalization errors when the training dataset keeps the same during local training. So the core difference between FL and PFL of generalization bound is the distribution of test dataset. Following the framework of this paper, we can creatively analyze the impact of data heterogeneity on PFL and FL. Here we make a preliminary attempt to analyze the effects of data heterogeneity.
> - **New assumption:** We define the bounded global variance assumption:  $\frac{1}{m} \sum_{i=1}^m\left\|| \nabla_u F_i\left(u, v_i\right)-\nabla_u F\left(u, V\right)\right\| |^2 \leq \delta_u^2$, where that bound $\delta$ could be the indicator of the data heterogeneity. The bigger the $\delta$, the more heterogeneous the distribution is. This assumption is commonly used in the convergence analysis for both FL [1-2] and PFL [3-5].
> - **Analysis technology:** We use this assumption to due with $\nabla_u F_i\left(u_i,v_i,z\right)$ and get
>
> $ \quad\quad\mathbb{E}||u_{i,k+1}^t$-$\tilde{u}_{i,k+1}^t||$
>
> $ \quad\quad\leq(1+\eta_uL_u)\mathbb{E}||u_{i,k}^t-\tilde{u}_{i,k}^t|| $
>
>  $ \quad\quad\quad+\eta_uL_{uv}\mathbb{E}||v_{i,K_v}^t-\tilde{v}_{i,K_v}^t||+2\eta_u(\sigma_u+\delta_u)$
>
>  when selecting the different sample in Lemma 7 in Line 1117.
> Advancing analysis of heterogeneity, we finally deduce a rough generalization bound $\frac{nU\tau_0}{mS}+\left(\frac{TK_u}{\tau_0}\right)^{\mu_u L_u}\frac{2G(\sigma_u+\delta_u)}{mSL_u} $+$\left(\frac{TK_v}{\tau_0}\right)^{\mu_v L_v}$ $\left( 1 + \frac{L_{uv}}{L_u} (\frac{TK_u}{\tau_0})^{\mu_u L_u} \right)$$ \frac{2G\sigma_v}{SL_v}$ for C-PFL (eq.(1)).
> And $\frac{U\tau_0}{S}+\frac{2G(\sigma_u+\delta_u)}{SL_u}\left(\frac{1+6\sqrt{m}\kappa_\lambda}{m}\right)\left(\frac{T K_u}{\tau_0}\right)^{\mu_uL_u}+\frac{2 \sigma_v G}{S L_v}\left(1+\frac{6\sqrt{m} \kappa_\lambda}{m}\left(\frac{L_{u v}}{L_v}\right)\right)\left(\frac{TK_v}{\tau_0}\right)^{\mu_vL_v}$ for D-PFL (eq.(2)) .
> - **Remark1:**  From that bound we can see how data distribution affects the generalization bound of *personalization* and *no personalization*. Assuming that $v_i$ attends to the aggregation, the generalization error for $v_i$ will add the heterogeneity impact and eq. (1) changes into $\frac{nU\tau_0}{mS} + \left(\frac{TK_u}{\tau_0}\right)^{\mu_uL_u}\frac{2G(\sigma_u+\delta_u)}{mSL_u} +\left(\frac{TK_v}{\tau_0}\right)^{\mu_vL_v} \left(1+\frac{L_{uv}}{L_u} (\frac{TK_u}{\tau_0})^{\mu_u L_u} \right) \frac{2G(\sigma_v+\delta_v)}{SL_v}$. It can clearly be seen that *personalization* without the system heterogeneity impact on $v_i$ performs better than *no personalization*.
> - **（Delete）Remark 2:** As an implication, our analysis with data distribution could obtain a non-iid bound to show that the presumably difficult (infinite dimensional) problem can be reduced to a simple binary decision problem of choosing between classical FedAvg and PureLocalTraining as in [6]. For full participation FedAvg, the generalization bound can be reduced to $\mathcal{O}\left(\frac{U\tau_0}{S} + \left(\frac{TK}{\tau_0}\right)^{\mu L}\frac{2G(\sigma+\delta)}{mSL}\right)$ removing the personalized parameters $v_i$. For PureLocalTraining, the generalization bound can be reduced to $\mathcal{O}\left(\frac{U\tau_0}{S} + \left(\frac{TK}{\tau_0}\right)^{\mu L}\frac{2G\sigma}{SL}\right)$ removing the shared parameters $u$. From the comparisons, when $\delta \rightarrow 0$ with iid distribution, FedAvg generalizes better than PureLocalTraining with m-client data augmentation. With the heterogeneity getting larger, the generalization bounds of these two algorithms will meet at $\delta = (m-1)\sigma$. So it can be an indicator for the binary decision problem.
>
> **Innovative Results about Decentralized PFL:** Compared to the results of Hardt et al, the analysis of D-PFL demonstrates the effect of communication topologies for generalization error. We introduce the topological spectral $(1-\lambda)$ and $\kappa_\lambda$ to show how communication topologies affect the generalization analysis.
> - We conclude that 1) C-PFL generalized better than D-PFL;2)Denser communication topology leads to better generalization performance, which means that the fully connected topology achieves the best generalization performance of shared variables and is compatible with the central ones.
> - Compared to the existing decentralized analysis, we obtain the tightest boundaries without the assumptions of bounded gradient [7] or the strong assumption that the weight difference obeys Gaussian distribution [8]. And [9] contains an extra constant term concerned with $\lambda$, which means the upper bound suffers from a nonvanishing influence of the communication network.

---

### Official Review · Reviewer_wmwb · 2024-11-04

**Soundness:** 2
**Presentation:** 2
**Contribution:** 2
**Rating:** 6
**Confidence:** 4

**Summary:**

The paper studies the generalization gaps of personalized federated learning (PFL) under centralized and decentralized cases. Uniform stability tools are used to derive generalization upper bounds that reflect the influences of graph topologies.

**Strengths:**

The paper provides generalization bounds leveraging uniform stability of the algorithm for PFL under both centralized and decentralized settings. The bounds are topology-related, which provides new insight in how graph structures affect the generalization.

**Weaknesses:**

1. The main weakness of the paper is that the bounds do not recover the centralized training with SGD. To be specific, PFL reduces to centralized SGD when $K=1$ and $v_i$ is constant $\forall i \in [m]$. However, the proposed bounds in Theorem 1 indicate a worse performance than SGD (Hardt et al., 2016).

2. The analysis of the paper is standard as literature, which means there is limited technical contribution of this paper.

3. Shown by (Sun, Niu, Wei, 2024), the generalization of FL is affected by different data heterogeneity levels, while this paper does not capture such phenomenon.

I am willing to raise the score if my concerns are addressed.

Reference:

Sun, Z., Niu, X., & Wei, E. (2024, April). Understanding generalization of federated learning via stability: Heterogeneity matters. In International Conference on Artificial Intelligence and Statistics (pp. 676-684). PMLR.

**Questions:**

1. Why does not the bound Theorem 1 match the centralized SGD if $K=1$ without personalization? Is it due to technical analysis? Could the authors solve such technical issue?

2. What are the main technical difficulties in the analysis, comparing to existing literature?

---

> ### Author Response · Authors · 2024-11-23
> **To Reviewer wmwb （1）**
>
> Thanks for your comments and your suggestions are very meaningful to us. We reply to them below and all the suggestions have been modified in the revision paper  in orange for you and in blue for all reviewers. We sincerely hope that you could reconsider the contributions of our work.
>
> **Weakness:**
>
> **W1:** Thank you for your kind comment. Our bound in Theorem 1 can further recover the analysis of SGD in（Hardt et al., 2016) and we polish the discussion in Remark 2 in the revision.
> - In the last version, we performed only one step of regression analysis. In the physical sense, the first step of the degradation process is to degrade from partial personalization to partial participation in local training, but the result is still for the entire partially participating system, rather than the pure local update for each node. Therefore, our stability for PFL reduce to $\mathcal{O}\left( (nK_vT)^{\frac{\mu_v L_v}{1+\mu_v L_v}}/(m S)\right)$ by removing the $K_u$ and $\sigma_u$ in eq. (7) .
> - In the revision, we further degrade the stability on the basis of the first step. When $K_v = 1$ and all participation ($\frac{n}{m} = 1$), $\mathcal{O}\left( (nK_vT)^{\frac{\mu_v L_v}{1+\mu_v L_v}}/(m S)\right)$ can reduce to $\mathcal{O}\left( T^{\frac{\mu_u L_u}{1+\mu_v L_v}}/S\right)$, which is align with the analysis in（Hardt et al., 2016). In the physical sense, the second step of degradation is to assume global participation, and the stability analysis degrades to the pure local training for each node, which is the same as that of SGD in（Hardt et al., 2016).
>
> **W2:** Our technical contributions are as follows:
>
> (1) *We analyze the effect of the **multiple local iterations** and **partial partition**, which bridges centralized SGD and the partial partition FL*. The technical difficulty is that local updates fail to be an unbiased gradient estimation after multiple local iterates, thereby merging the multiple local iterations is non-trivial.
>
> (2) *We analyze the effect of the **personalized training** and **federated aggregation**, which is the first generalization analysis to bridges FL and PFL*. The technical difficulty is that the alignment errors exist in local training of the personalized variables after combining the two parts, thereby understanding the relationship between FL and PFL is non-trivial.
>
> (3) *We analyze the effect of **communication topologies** in Decentralized PFL, which is the first generalization analysis to bridge C-PFL and D-PFL.* Lack of the central server, we need more complicated analysis of the aggregation error introduced by peer-to-peer communication with different gossip matrix $W$. Therefore, understanding the relationship between C-PFL and D-PFL is non-trivial.
>
> (4) *We analyze the effect of **model decoupling** and **alternative training** in both FL \& DFL \& PFL, which is the first generalization analysis to bridge full model personalization and partial model personalization.* We analyze the generalization performance of different model parts separately and obtain the alternative errors between them, which provides interesting and useful suggestions for theoretical analysis of model decoupling and alternative training.

---

> ### Author Response · Authors · 2024-11-23
> **To Reviewer wmwb （2）**
>
> **W3:**
> - Thank you for your kind advise and it is our aim to bridge the generalization bound and data heterogeneity analysis for PFL in the future work. We also point out this meaningful idea in the CONCLUSION in Line 539. With the analysis of data heterogeneity, people can get more ideas to choose the suitable personalized strategies under different data distribution, which will be another significant work for PFL \& FL community.
>
> - That analysis on data heterogeneity in the reference work is interesting and inspires us a lot. Carefully considered the heterogeneity analysis, we think the key to consider the data distribution is how to measure the local gradient $g_i(\theta)$ ($\nabla_u F_i\left(u_i, v_i,z \right)$ in our paper), which can be dueled with $\| \nabla R_i(\theta) - \nabla R(\theta) \|$ contained the heterogeneity information $D_i$ in Lemma 1 in (Sun, Niu, Wei, 2024). The similar idea can be extended to our analysis in Lemma 7 and Lemma 9.  Thank you and we add the discussion about it in Table 1 and Related Work in our revision.
>
>
> - Another intuition to consider the data distribution in generalization analysis is to bridge the local gradient $\nabla_u F_i\left(u_i, v_i,z \right)$ and global variance $\frac{1}{m} \sum_{i=1}^m\left\|\nabla_u F_i\left(u_i, v_i\right)-\nabla_u F\left(u_i, V\right)\right\|^2 \leq \delta^2$, which is a commenly used assumption in the convergence analysis in PFL [1-3]. The bigger $\delta$ means the further distance between the global distribution and any local distribution.
>
>
> - The data heterogeneity analysis exists from the convergence performance in our excess risk analysis. Referred to the heterogeneity $\delta$ in Eq.(7) in [1] for C_PFL and Eq.(3) in [2] for D_PFL, the data heterogeneity exactly effects our excess risk.
>
>
> [1] Krishna Pillutla, Kshitiz Malik, Abdel-Rahman Mohamed, Mike Rabbat, Maziar Sanjabi, and Lin Xiao. Federated learning with partial model personalization. In International Conference on Machine Learning, pp. 17716–17758. PMLR, 2022.
>
> [2] Yifan Shi, Yingqi Liu, Yan Sun, Zihao Lin, Li Shen, Xueqian Wang, and Dacheng Tao. Towards more suitable personalization in federated learning via decentralized partial model training. arXiv preprint arXiv:2305.15157, 2023b.
>
> [3] Chen, Y., Cao, L., Yuan, K., & Wen, Z. (2023). Sharper convergence guarantees for federated learning with partial model personalization. arXiv preprint arXiv:2309.17409.
>
> **Questions:**
>
> **Q1:** See W1.
>
> **Q2:** See W2.
>
> All the suggestions above have been modified in orange in the revision paper for you and  in blue for all reviewers. Please reconsider our contributions to the FL community. lf you have any further questions, please do not hesitate to contact us. Thanks again!

---

> > ### Comment · Reviewer_wmwb · 2024-11-26
> >
> > Thank authors for the response. I think my concerns have been addressed, so I raised my score to 6.

---

> ### Author Response · Authors · 2024-11-25
> **Look forward to further discussions**
>
> Dear reviewer,
>
> We hope this message finds you well. We greatly appreciate your time and effort in reviewing our paper. We have carefully answered all the questions you raised about our paper and look forward to further discussions with you.

---

### Author Response · Authors · 2024-12-02
**Summary to Reviewers, AC and SACs**

Dear ACs, SACs,

Thank you for taking the time to thoroughly review our paper and for engaging in thoughtful discussions with us over the past two weeks. We greatly appreciate the reviewers' valuable comments, which have helped us refine and improve our work. We hereby reiterate the core contributions of our work and summarize this rebuttal to facilitate subsequent review.

**Core Contributions:**

- **First work on the algorithm-dependent generalization for both C-PFL and D-PFL under non-convex conditions.** We build up the first uniform stability-based
generalization for PFL with the biased gradient from multi-local updates. Our analysis establishes the interaction mechanism between global aggregation and local fine-tuning, then provides valuable guidance for alternating personalized optimization. We also extend this analysis to decentralized scenarios with different communication topologies, which have a huge impact on generalization bounds. The analysis achieves the tightest bound in decentralized training without strong assumptions.

- **New theoretical results for upper generalization bounds and excess risks for PFL.** Our theoretical results reveal the impacts of the trivial factors on generalization performance. Following the framework, we can creatively analyze the advantages of PFL over FL with data heterogeneity. Moreover, we analyze how communication topologies influence the upper generalization bounds of D-PFL and demonstrate that C-PFL generalizes better than D-PFL. Combined with the convergence errors, we finally obtain the excess risk of both C-PFL and D-PFL.

- **Massive experiments to verify the theoretical findings of PFL.** We evaluate important factors to verify our theoretical findings on CIFAR10/100 with different VGG/ResNet under non-convex conditions. The empirical results strongly support our theoretical insights.

**Rebuttal Summary:**

- **Adding comparisons with stability analysis and highlighting our advantages over the current PFL analysis.** We add the comparisons of stability analysis of SGD、FL and D-SGD in Table 1. We are the first to analyze the generalization impact from personalized variables to shared variables, which uncovers the interaction mechanism between these two updating processes and provides valuable guidance for alternating personalized optimization. Compared to the existing analysis of PFL in Table 2, we are the first to conduct the generalization analysis in the non-convex condition in both CFL and DFL, which includes the impacts of the algorithm design and the hyperparameters selection for personalized training.
- **More detailed verification experiments with different datasets and models.** We follow the reviewers' advice to add the experiments of model parameter difference between the perturbations datasets and add more experiments on CIFAR-100 with VGG11 in Appendix D.2. The additional results can further strongly support our theoretical findings.
- **Correct typos and polish the paper writing.** We polish the contributions and technical innovations of this paper. We add more details in the analysis and further optimize the comparison with SGD in the remark. In the appendix, we add more detailed information about the existing generalization analysis of PFL.
- **Explore the future work.** Based on the reviewers' comments, we discuss three interesting and open problems in future work: 1) the lower bound and tightness of PFL; 2) the unique stability design for PFL; 3) the impact of data heterogeneity on PFL and FL.  Each of these three issues deserves careful discussion in separate works.  In the rebuttal, We preliminarily present the generalization advantage of PFL over FL with data heterogeneity and compare the generalization upper bounds of FL and PureLocalTraining. The above three open problems are of great significance to the FL community.


We would like to thank the reviewers, ACs, and SACs again for your hard work and valuable suggestions for this paper. Please reconsider our contributions to the FL community.

Best wishes,

Authors.

---

### Meta-Review · Area_Chair_fQNZ · 2024-12-21

**Metareview:**

a) Summary

This work analyzes the generalization properties of federated learning with personalization by studying the stability of models trained under non-convex loss functions. It provides bounds on generalization error, decomposing the contributions of global aggregation and local updates, and examines the effects of hyperparameters like learning rate and number of communication rounds. The results show that centralized training schemes achieve better generalization than decentralized ones, and the analysis offers guidance on selecting training parameters to minimize excess risk.

b) Strengths
- Non trivial generalization bounds in FL: The paper tackles the underexplored area of generalization analysis in personalized federated learning (PFL), particularly under non-convex settings, providing insights beyond the commonly studied convergence guarantees.
- Dependence on algorithms, topology, and hyperparameter: the analysis based on uniform stability can be used to study the effects of the algorithm choice, communication topology, and hyperparameters on model performance.
- Empirical Validation: Experimental results on CIFAR datasets support the theoretical findings and show the utility of the bounds.
- Interesting finding: both theory and empirical evidence suggests that current that centralized schemes generalize better than decentralized ones. Understanding this better and overcoming this promises to be an interesting research agenda for the FL community.

c) Weaknesses
- Lack of Tight Bounds: The theoretical results do not include lower bounds, making it difficult to assess the tightness or optimality of the derived generalization error bounds.

- Limited Technical Novelty: The analysis primarily extends existing stability results (e.g., Hardt et al., 2016) without introducing significantly new proof techniques or theoretical insights.

- Incomplete Treatment of Data Heterogeneity: While data heterogeneity is acknowledged as important, the generalization bounds fail to fully quantify its impact, leaving a key aspect of federated learning underexplored.

- Effect of personalization is not properly captured by the current analysis as raised in the discussion by Reviewer tNFV.

d) Reasons to recommend **accept**

This paper provides a first step in understanding the generalization properties of personalized federated learning (PFL), an area that remains underexplored compared to convergence analysis. Though far from optimal and despite the lack of novel theoretical progress, the work derives bounds that seem to offering actionable guidance for practical FL settings. I believe that this work will spark interesting discussions among the FL community and lead to further interesting research.

This is not to say that the concerns raised by the reviewers, especially in the lengthy discussions with Reviewer tNFV who remained unconvinced by the authors, can be dismissed. This paper really **needs additional polishing**. I strongly recommend the authors to incorporate the comments, especially **flesh out the generalization advantage of PFL over FL with data heterogeneity** which was developed over the discussion with tNFV.

**Additional Comments On Reviewer Discussion:**

During the rebuttal period, reviewers raised concerns about the lack of tightness in the generalization bounds, limited novelty in theoretical contributions, incomplete treatment of data heterogeneity, and poor presentation, particularly in the context of comparing personalization vs. no personalization. The authors addressed some of these issues by revising claims about "optimal" bounds, adding a preliminary analysis on data heterogeneity, improving notation, and expanding experimental validation to include perturbation-based experiments. However, key concerns, such as the absence of lower bounds and a deeper exploration of personalization’s advantages under heterogeneity, remained unresolved, and presentation issues persisted.

In my final decision, I considered that while the work has limitations, it introduces a valuable framework for understanding generalization in PFL and provides empirical evidence supporting its theoretical findings. The paper offers actionable insights and has the potential to inspire further research in this underexplored area, outweighing its weaknesses, provided the authors commit to addressing the identified gaps.

---

### Decision · Program_Chairs · 2025-01-22

Accept (Poster)